# Robust Simulation-Based Inference under Missing Data via Neural Processes

**Yogesh Verma, Ayush Bharti**
Department of Computer Science, Aalto University
{yogesh.verma, ayush.bharti}@aalto.fi

**Vikas Garg**
YaiYai Ltd and Aalto University
vgarg@csail.mit.edu

## Abstract

Simulation-based inference (SBI) methods typically require fully observed data to infer parameters of models with intractable likelihood functions. However, datasets often contain missing values due to incomplete observations, data corruptions (common in astrophysics), or instrument limitations (e.g., in high-energy physics applications). In such scenarios, missing data must be imputed before applying any SBI method. We formalize the problem of missing data in SBI and demonstrate that naive imputation methods can introduce bias in the estimation of SBI posterior. We also introduce a novel amortized method that addresses this issue by jointly learning the imputation model and the inference network within a neural posterior estimation (NPE) framework. Extensive empirical results on SBI benchmarks show that our approach provides robust inference outcomes compared to standard baselines for varying levels of missing data. Moreover, we demonstrate the merits of our imputation model on two real-world bioactivity datasets (Adrenergic and Kinase assays). Code is available at https://github.com/Aalto-QuML/RISE.

## 1 Introduction

Mechanistic models for studying complex physical or biological phenomena have become indispensable tools in research fields as diverse as genetics (Riesselman et al., 2018), epidemiology (Kypraios et al., 2017), gravitational wave astronomy (Dax et al., 2021), and radio propagation (Bharti et al., 2022a). However, fitting such models to observational data can be challenging due to the intractability of their likelihood functions, which renders standard Bayesian inference methods inapplicable. Simulation-based inference (SBI) methods (Cranmer et al., 2020) tackle this issue by relying on forward simulations from the model instead of evaluating the likelihood. These simulations are then either used to train a conditional density estimator (Papamakarios and Murray, 2016; Lueckmann et al., 2017b; Greenberg et al., 2019; Papamakarios et al., 2019; Radev et al., 2020), or to measure distance with the observed data (Sisson, 2018; Briol et al., 2019; Pesonen et al., 2023), to approximately estimate the posterior distribution of the model parameters of interest.

SBI methods implicitly assume that the observed data distribution belongs to the family of distributions induced by the model; i.e., the model is *well-specified*. However, this assumption is often violated in practice where models tend to be *misspecified* since the complex real-world phenomena under study are not accurately represented. Even if the model is well-specified, the data collection mechanism might hinder the applicability of SBI methods since it can induce missing data due to, for instance, incomplete observations (Luken et al., 2021), instrument limitations (Kasak et al., 2024), or unfavorable experimental conditions.

Although the former problem of model misspecification has been studied in a number of works (Frazier et al., 2020; Dellaporta et al., 2022; Fujisawa et al., 2021; Bharti et al., 2022b; Ward et al., 2022; Schmitt et al., 2023; Gloeckler et al., 2023; Huang et al., 2023; Gao et al., 2023; Kelly et al., 2024), the latter problem of missing data in SBI has received relatively less attention. A notable exception is the work of Wang et al. (2024), which attempts to handle missing data by augmenting and imputing constant values (e.g., zero or sample mean) and performing inference with a binary mask indicator. However, this approach can lead to biased estimates, reduced variability, and distorted relationships between variables (Graham et al., 2007). This is exemplified in Figure 1 where we investigate the impact of missing data on neural pos-

terior estimation (NPE, Papamakarios and Murray (2016))—a popular SBI method—on a population genetics model. We observe that simply incorporating missing values and their corresponding masks in NPE methods as in Wang et al. (2024) leads to biased posterior estimates.

Other SBI works that address missing data include Lueckmann et al. (2017a) and Gloeckler et al. (2024), however, they fail to account for the underlying mechanism that leads to missing values in the data.

Outside of SBI, the problem of missing data has been extensively studied (Van Buuren and Groothuis-Oudshoorn, 2011), with Rubin (1976) categorizing it into three types: missing completely at random (MCAR), missing at random (MAR), and missing not at random (MNAR). Recent advances in machine learning have led to the development of novel methods for addressing this problem using generative adversarial networks (GANs, Luo et al. (2018); Yoon et al. (2018); Li et al. (2019); Yoon and Sull (2020)), variational autoencoders (VAEs, Nazabal et al. (2020); Collier et al. (2020); Mattei and Frellsen (2019); Ipsen et al. (2020); Ghalebikesabi et al. (2021b)), Gaussian processes (Casale et al., 2018; Fortuin et al., 2020; Ramchandran et al., 2021; Ong et al., 2024), and optimal transport (Muzellec et al., 2020; Zhao et al., 2023; Vo et al., 2024). These methods offer new perspectives on the problem of missing data imputation, but their application has been primarily limited to predicting missing values. Notably, they have not been developed for inference over missing values, which remains a significant challenge for SBI.

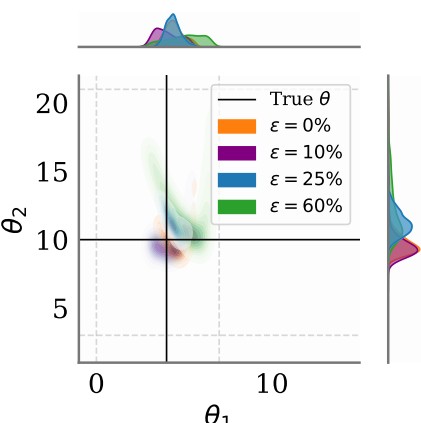

Figure 1: **Effect of missing data on SBI.** NPE posterior for the two-parameter Ricker model (Wood, 2010) where the method of Wang et al. (2024) (with zero augmentation) is used to handle $\varepsilon\%$ of values missing in the data. As $\varepsilon$ increases, the NPE posteriors become biased and drift away from the true parameter value, denoted by the black lines.

**Contributions.** In this paper, we introduce a novel SBI method that is robust to shift in the posterior distribution due to missing data. Our method, named RISE (short for "Robust Inference under imputed SimulatEd data"), jointly performs imputation and inference by combining NPE with latent neural processes (Foong et al., 2020). Doing so allows us to learn an *amortized* model unlike other robust SBI methods in the literature, and to handle missing data under different assumptions (Little and Rubin, 2019). We summarize our main contributions below:

- we motivate the problem of missing data in SBI, arguing how it can induce bias in posterior estimation;
- we propose RISE, an amortized method, that jointly learns an imputation and inference network to deal with missing data;
- RISE outperforms competing baselines in inference and imputation tasks across varying levels of missingness, demonstrating robust performance in settings entailing missing data.

## 2 PRELIMINARIES

Consider a simulator-based model $p(\cdot \,|\, \theta)$ that takes in a parameter vector $\theta \in \Theta \subseteq \mathbb{R}^p$ and maps it to a point $\mathbf{x} = [x_1, \dots, x_d]^\top$ in some data space $\mathcal{X} \subseteq \mathbb{R}^d$. We assume that $p(\cdot \,|\, \theta)$ is intractable, meaning that its associated likelihood function is unavailable and cannot be evaluated point-wise. However, in our setting, generating independent and identically distributed (iid) realisations $\mathbf{x} \sim p(\cdot \,|\, \theta)$ for a fixed $\theta$ is straightforward. Given a dataset $\tilde{\mathbf{x}}$ collected via real-world experiments from some true data-generating process and a prior distribution on the parameters $p(\theta)$, we are interested in approximating the posterior distribution $p(\theta \,|\, \tilde{\mathbf{x}}) \propto p(\tilde{\mathbf{x}} \,|\, \theta)p(\theta)$. This can be achieved, for instance, using the popular neural posterior estimation (NPE) method, which we now introduce.

**Neural posterior estimation.** NPE (Papamakarios and Murray, 2016) involves training conditional density estimators, such as normalizing flows (Papamakarios et al., 2021), to learn a mapping from each datum $\mathbf{x}$ to the posterior distribution $p(\theta \,|\, \mathbf{x})$. Specifically, we can approximate the posterior

distribution with $q_\phi(\theta \,|\, \mathbf{x})$ using learnable parameters $\phi$. In particular, we can train $q_\phi$ by minimizing an empirical loss

$$\ell_{\text{NPE}}(\phi) \triangleq -\frac{1}{n}\sum_{i=1}^{n} \log q_\phi(\theta_i \,|\, \mathbf{x}_i) \approx -\mathbb{E}_{\theta \sim p(\theta)}[\mathbb{E}_{\mathbf{x} \sim p(\mathbf{x} \,|\, \theta)}[\log q_\phi(\theta \,|\, \mathbf{x})]], \tag{1}$$

using the dataset $\{(\theta_i, \mathbf{x}_i)\}_{i=1}^{n}$ simulated from the joint distribution $p(\theta, \mathbf{x}) = p(\mathbf{x} \,|\, \theta)p(\theta)$. When the data space $\mathcal{X}$ is high-dimensional, or there are multiple observations $\mathbf{X} = [\mathbf{x}^{(1)}, \ldots, \mathbf{x}^{(m)}]$ for each $\theta$, we can use a summary function $\eta : \mathcal{X} \to \mathcal{S}$ (such as a deep set (Zaheer et al., 2017)) to enable a condensed representation. Assuming that the summary function is parameterized by $\kappa$, the joint NPE loss with respect to both $\phi$ and $\kappa$ can be defined as $\ell_{\text{NPE}}(\phi, \kappa) \triangleq -\frac{1}{n}\sum_{i=1}^{n} \log q_\phi(\theta_i \,|\, \eta_\kappa(\mathbf{x}_i))$. Once both $q_\phi$ and $\eta_\kappa$ are trained, the NPE posterior estimate $q_{\hat\phi}(\theta \,|\, \eta_{\hat\kappa}(\tilde{\mathbf{x}}))$ for any given real data $\tilde{\mathbf{x}}$ is obtained by a simple forward pass of $\tilde{\mathbf{x}}$ through the trained networks, making NPEs amortized. We now provide a brief background on the missing data problem, which is the focus of this work.

**Missing data background.** In the context of missing data, each data sample is composed of an observed part $\mathbf{x}_{\text{obs}}$ and a missing (or unobserved) part $\mathbf{x}_{\text{mis}}$ such that $\mathbf{x} = (\mathbf{x}_{\text{obs}}, \mathbf{x}_{\text{mis}})$. The missingness pattern for each $\mathbf{x}$ is described by a binary mask variable $\mathbf{s} \in \{0, 1\}^d$, where $s_i = 1$ if the element $x_i$ is observed and $s_i = 0$ if $x_i$ is missing, $i \in \{1, \ldots, d\}$. The joint distribution of $\mathbf{x}$ and $\mathbf{s}$ can be factorized as $p(\mathbf{x}, \mathbf{s}) = p(\mathbf{s} \,|\, \mathbf{x})p(\mathbf{x})$. Based on specific assumptions about what the conditional distribution of the mask (or the missingness mechanism) depends on, three different scenarios arise (Little and Rubin, 2019): (i) missing-completely-at-random (MCAR), where $p(\mathbf{s} \,|\, \mathbf{x}) = p(\mathbf{s})$; (ii) missing-at-random (MAR), where $p(\mathbf{s} \,|\, \mathbf{x}) = p(\mathbf{s} \,|\, \mathbf{x}_{\text{obs}})$; and (iii) missing-not-at-random (MNAR), where $p(\mathbf{s} \,|\, \mathbf{x}) = p(\mathbf{s} \,|\, \mathbf{x}_{\text{obs}}, \mathbf{x}_{\text{mis}})$.

The missingness mechanism can be ignored for both MCAR and MAR when learning $p(\mathbf{x}_{\text{obs}}, \mathbf{s})$, but not for MNAR where it depends on $\mathbf{x}_{\text{mis}}$ (Ipsen et al., 2020). We aim to handle all the three cases when performing SBI.

## 3 METHOD

We begin by analyzing the issue of missing data in SBI settings in Section 3.1. We then present RISE — our method for handling missing data in SBI. Section 3.2 outlines our learning objective, and Section 3.3 describes how we parameterize the imputation model in RISE using neural processes.

### 3.1 MISSING DATA PROBLEM IN SBI

We assume that the simulator can faithfully replicate the true data-generating process (i.e., the simulator is well-specified), however, the data collection mechanism induces missing values in each data point $\mathbf{x}$. As a result, $\mathbf{x}$ contains both observed and missing values,[1] represented as $\mathbf{x} = (\mathbf{x}_{\text{obs}}, \mathbf{x}_{\text{mis}})$. For instance, $\mathbf{x} = (0.1\ 1.2\ -\ 0.9)$ exemplifies a scenario where a specific coordinate $x_i$ is missing (indicated by '$-$'). Naturally, SBI methods cannot operate on missing values, and so imputing $\mathbf{x}_{\text{mis}}$ is necessary before proceeding to inference. However, if the missing values are not imputed accurately, then the corresponding SBI posterior becomes biased (e.g., as observed in Figure 1 due to constant imputation). We now describe this problem mathematically.

**Definition 1** (SBI posterior under true imputation). *Let $p_{true}(\mathbf{x}_{mis} \,|\, \mathbf{x}_{obs})$ be the true predictive distribution of the missing values given the observed data. Then, the SBI posterior can be written as*

$$p_{SBI}(\theta \,|\, \mathbf{x}_{obs}) = \int \underbrace{p_{SBI}(\theta \,|\, \mathbf{x}_{obs}, \mathbf{x}_{mis})}_{Inference} \underbrace{p_{true}(\mathbf{x}_{mis} \,|\, \mathbf{x}_{obs})}_{Imputation} d\mathbf{x}_{mis}. \tag{2}$$

We thus have a distribution over the missing values given $\mathbf{x}_{\text{obs}}$, and the problem of SBI under missing data is formulated as an expectation of the SBI posterior $p_{\text{SBI}}(\theta \,|\, \mathbf{x}_{\text{obs}}, \mathbf{x}_{\text{mis}})$ with respect to $p_{\text{true}}(\mathbf{x}_{\text{mis}} \,|\, \mathbf{x}_{\text{obs}})$, analogous to traditional (likelihood-based) Bayesian inference methods (Schafer and Schenker, 2000; Zhou and Reiter, 2010). Therefore, estimating the above expectation requires

---

[1]Note that during training, $\mathbf{x}_{\text{obs}}$ and $\mathbf{x}_{\text{mis}}$ are partitions of the simulated data $\mathbf{x}$, while during inference we only observe $\tilde{\mathbf{x}}_{\text{obs}}$ from the real world.

access to $p_{\text{true}}(\mathbf{x}_{\text{mis}} \mid \mathbf{x}_{\text{obs}})$ (Raghunathan et al., 2001; Gelman et al., 1995), which is infeasible in most practical cases.

**Definition 2** (SBI posterior under estimated imputation). *Let $\hat{p}(\mathbf{x}_{mis} \mid \mathbf{x}_{obs})$ denote an estimate of the true imputation model $p_{true}(\mathbf{x}_{mis} \mid \mathbf{x}_{obs})$. Then, the corresponding SBI posterior can be written as*

$$\hat{p}_{SBI}(\theta \mid \mathbf{x}_{obs}) = \int p_{SBI}(\theta \mid \mathbf{x}_{obs}, \mathbf{x}_{mis}) \hat{p}(\mathbf{x}_{mis} \mid \mathbf{x}_{obs}) d\mathbf{x}_{mis}. \tag{3}$$

**Proposition 1.** *If $\hat{p}(\mathbf{x}_{mis} \mid \mathbf{x}_{obs})$ is misaligned with $p_{true}(\mathbf{x}_{mis} \mid \mathbf{x}_{obs})$, then the estimated SBI posterior $\hat{p}_{SBI}(\theta \mid \mathbf{x}_{obs})$ will be biased (in general), i.e., $\mathbb{E}_{\theta \sim p_{SBI}(\cdot \mid \mathbf{x}_{obs})}[\theta] \neq \mathbb{E}_{\theta \sim \hat{p}_{SBI}(\cdot \mid \mathbf{x}_{obs})}[\theta]$.*

The proof, which follows straightforwardly using Definition 1 and Definition 2, is given in Appendix A.2.1 for completeness. Proposition 1 says that the bias in the SBI posterior directly comes from the discrepancy between the true imputation model $p_{\text{true}}(\mathbf{x}_{\text{mis}} \mid \mathbf{x}_{\text{obs}})$ and the estimated one $\hat{p}(\mathbf{x}_{\text{mis}} \mid \mathbf{x}_{\text{obs}})$. This applies irrespective of the inference method used, and therefore, rather unsurprisingly, the key to reducing this bias is to learn the imputation model as accurately as possible. The rest of this section presents our method, named RISE, which combines the imputation task with SBI to reduce this bias.

### 3.2 ROBUST SBI UNDER MISSING DATA

Let $p_{\text{true}}(\theta \mid \mathbf{x}_{\text{obs}}, \mathbf{x}_{\text{mis}})$ be the true posterior given both the observed data and the missing values, i.e., given $\mathbf{x} = (\mathbf{x}_{\text{obs}}, \mathbf{x}_{\text{mis}})$. Our objective is to estimate the true posterior given only $\mathbf{x}_{\text{obs}}$. That is, we seek to approximate

$$p_{\text{true}}(\theta \mid \mathbf{x}_{\text{obs}}) \triangleq \int p_{\text{true}}(\theta \mid \mathbf{x}_{\text{obs}}, \mathbf{x}_{\text{mis}}) p_{\text{true}}(\mathbf{x}_{\text{mis}} \mid \mathbf{x}_{\text{obs}}) d\mathbf{x}_{\text{mis}} = \int p_{\text{true}}(\theta, \mathbf{x}_{\text{mis}} \mid \mathbf{x}_{\text{obs}}) d\mathbf{x}_{\text{mis}}.$$

We therefore introduce a family of distributions $r_\psi(\theta, \mathbf{x}_{\text{mis}} \mid \mathbf{x}_{\text{obs}})$ parameterized by $\psi$, and propose to solve the following optimization problem

$$\underset{\psi}{\arg\min} \ \mathbb{E}_{\mathbf{x}_{\text{obs}} \sim p_{\text{true}}} \text{KL} \left[ p_{\text{true}}(\theta, \mathbf{x}_{\text{mis}} \mid \mathbf{x}_{\text{obs}}) \ || \ \underbrace{r_\psi(\theta, \mathbf{x}_{\text{mis}} \mid \mathbf{x}_{\text{obs}})}_{\textbf{(joint imputation and inference)}} \right]. \tag{4}$$

Solving this problem requires access to $p_{\text{true}}(\mathbf{x}_{\text{mis}} \mid \mathbf{x}_{\text{obs}})$, which in most real-world scenarios, we do not have. Since samples for $\mathbf{x}_{\text{mis}}$ are required during training, we need to resort to methods such as variational approximation or expectation maximization. Here, we adopt a variational approach, treating $\mathbf{x}_{\text{mis}}$ as latent variables in a probabilistic imputation setting. Specifically, the imputation network needs to estimate these latents for the inference network to map them to the output space. Both networks are tightly coupled since the distribution induced by the imputation network shapes the input of the inference network.

Mathematically, assuming access to only data samples $(\mathbf{x}_{\text{obs}}, \theta) \sim p_{\text{true}}$, we proceed to solving

$$\underset{\psi}{\arg\min} \ \mathbb{E}_{\mathbf{x}_{\text{obs}} \sim p_{\text{true}}} \text{KL}[p_{\text{true}}(\theta \mid \mathbf{x}_{\text{obs}}) \ || \ r_\psi(\theta \mid \mathbf{x}_{\text{obs}})]. \tag{5}$$

Our next proposition computes a variational lower bound for this objective, which we can maximize efficiently using an encoder-decoder architecture resembling variational autoencoders (VAEs).

**Proposition 2** (Training objective). *The objective in Equation (5) admits a variational lower bound, resulting in the following optimization problem.*

$$\hat{\phi}, \hat{\varphi} = \underset{\phi, \varphi}{\arg\min} - \mathbb{E}_{(\mathbf{x}_{obs}, \theta) \sim p_{true}} \mathbb{E}_{\mathbf{x}_{mis} \sim p(\mathbf{x}_{mis} \mid \mathbf{x}_{obs})} \left[ \log \underbrace{\hat{p}_\varphi(\mathbf{x}_{mis} \mid \mathbf{x}_{obs})}_{\textit{(imputation)}} + \log \underbrace{q_\phi(\theta \mid \mathbf{x}_{obs}, \mathbf{x}_{mis})}_{\textit{(inference)}} \right] \tag{6}$$

$$= \underset{\phi, \varphi}{\arg\min} \ \ell_{RISE}(\phi, \varphi),$$

*where $\ell_{RISE}(\phi, \varphi)$ denotes the loss function for RISE.*

Therefore, we can approximate the true imputation model $p_{\text{true}}(\mathbf{x}_{\text{mis}} \mid \mathbf{x}_{\text{obs}})$ using a parametric neural network $\hat{p}_\varphi$, parameterized by its vector of weights and biases $\varphi$, and the SBI posterior given the full dataset $p_{\text{SBI}}(\theta \mid \mathbf{x}_{\text{mis}}, \mathbf{x}_{\text{obs}})$ using the conditional density $q_\phi$ as in NPE.

The proof of Proposition 2 is outlined in Appendix A.2.2. Note that $\ell_{\text{RISE}}$ is a general loss which reduces to $\ell_{\text{NPE}}$ when there is no missing data, i.e., $\mathbf{x} = \mathbf{x}_{\text{obs}}$. In case a summary network $\eta_\kappa$ is required before passing the data to $q_\phi$, the joint loss function for RISE can be simply defined as

$$\ell_{\text{RISE}}(\phi, \varphi, \kappa) \triangleq -\mathbb{E}_{(\mathbf{x}_{\text{obs}}, \theta) \sim p_{\text{true}}, \mathbf{x}_{\text{mis}} \sim p(\mathbf{x}_{\text{mis}} \mid \mathbf{x}_{\text{obs}})} \left[ \log q_\phi(\theta \mid \eta_\kappa(\mathbf{x}_{\text{obs}}, \mathbf{x}_{\text{mis}})) + \log \hat{p}_\varphi(\mathbf{x}_{\text{mis}} \mid \mathbf{x}_{\text{obs}}) \right].$$

The expectation in Equation (6) is taken with respect to the joint distribution of the simulator and the prior (as is standard for SBI methods), and the variational imputation distribution $p(\mathbf{x}_{\text{mis}} \mid \mathbf{x}_{\text{obs}})$. Note that for simulations in our controlled experiments, we do not need to resort to the variational distribution $p(\mathbf{x}_{\text{mis}} \mid \mathbf{x}_{\text{obs}})$, and can instead generate samples from $p_{\text{true}}(\mathbf{x}_{\text{mis}} \mid \mathbf{x}_{\text{obs}})$ directly by first sampling $\mathbf{x}$ using the simulator, and then partitioning it into $\mathbf{x}_{\text{obs}}$ and $\mathbf{x}_{\text{mis}}$ based on the missingness assumption (i.e. creating the mask $\mathbf{s}$ under MCAR or MAR or MNAR assumption) such that $\varepsilon\%$ portion of the data is missing. The $\mathbf{x}_{\text{mis}}$ values are then used as true labels when comparing against the output of the imputation model $\hat{p}_\varphi$ during training. This allows us to amortize over instances of real data. In Section 3.3, we discuss how RISE can be used to amortize over the proportion of missing values $\varepsilon$ in the data.

Using a latent variable representation (Kingma, 2013) for the imputation model, we factorize $\hat{p}_\varphi(\mathbf{x}_{\text{mis}} \mid \mathbf{x}_{\text{obs}})$, similarly to Mattei and Frellsen (2019), as

$$\hat{p}_\varphi(\mathbf{x}_{\text{mis}} \mid \mathbf{x}_{\text{obs}}) = \int \hat{p}_\alpha(\mathbf{x}_{\text{mis}} \mid \tilde{\mathbf{z}}) \, \hat{p}_\beta(\tilde{\mathbf{z}} \mid \mathbf{x}_{\text{obs}}) d\tilde{\mathbf{z}},$$

where $\varphi = (\alpha, \beta)$ are parameters of the imputation model, and $\tilde{\mathbf{z}} = (\mathbf{z}, \mathbf{s})$ represents both the latent variable $\mathbf{z}$ and the masking variable $\mathbf{s}$, which we can utilize to simulate various missingness environments. The conditional distribution of the latent $\hat{p}_\beta(\tilde{\mathbf{z}} \mid \mathbf{x}_{\text{obs}})$ may depend on both the observed and the missing data depending on the different missingness assumptions (Little and Rubin, 2019):

- MCAR: $\hat{p}_\beta(\tilde{\mathbf{z}} \mid \mathbf{x}_{\text{obs}}) = p_{\beta_1}(\mathbf{z} \mid \mathbf{x}_{\text{obs}})p_{\beta_2}(\mathbf{s})$
- MAR: $\hat{p}_\beta(\tilde{\mathbf{z}} \mid \mathbf{x}_{\text{obs}}) = p_{\beta_1}(\mathbf{z} \mid \mathbf{x}_{\text{obs}})p_{\beta_2}(\mathbf{s} \mid \mathbf{x}_{\text{obs}})$
- MNAR: $\hat{p}_\beta(\tilde{\mathbf{z}} \mid \mathbf{x}_{\text{obs}}) = p_{\beta_1}(\mathbf{z} \mid \mathbf{x}_{\text{obs}}) \int p_{\beta_2}(\mathbf{s} \mid \mathbf{x}_{\text{mis}}, \mathbf{x}_{\text{obs}})p(\mathbf{x}_{\text{mis}} \mid \mathbf{x}_{\text{obs}})d\mathbf{x}_{\text{mis}}$.

Note that for the MCAR and MAR cases, we only need the latent $\mathbf{z}$ in order to impute $\mathbf{x}_{\text{mis}}$ (Mattei and Frellsen, 2019), in which case $\tilde{\mathbf{z}} = \mathbf{z}$. However, in the MNAR case, $\tilde{\mathbf{z}} = (\mathbf{z}, \mathbf{s})$ as we will explicitly need to account for the missingness mechanism (Ipsen et al., 2020). Hereafter, we continue to denote the latent variable with $\tilde{\mathbf{z}}$ for a general formulation encompassing all the three cases. The pseudocode for training RISE is outlined in Algorithm 1.

### 3.3 LEARNING THE IMPUTATION MODEL USING NEURAL PROCESS

We utilize neural processes (NPs, Garnelo et al. (2018)) for parameterizing the imputation model $\hat{p}_\varphi(\mathbf{x}_{\text{mis}} \mid \mathbf{x}_{\text{obs}})$. NPs represent a family of neural network-based meta-learning models that combine the flexibility of deep learning with well-calibrated uncertainty estimates and a tractable training objective. These models learn a distribution over predictors given their target positions or locations. We refer the interested reader to Appendix A.3 for a detailed background. We employ neural processes to model the predictive density over missing values at their specific locations.

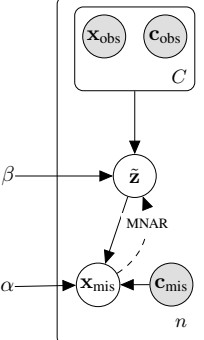

Let $\mathbf{c}_{\text{mis}} = (c_{\text{mis},1}, \ldots, c_{\text{mis},k})$ and $\mathbf{c}_{\text{obs}} = (c_{\text{obs},1}, \ldots, c_{\text{obs},d-k})$ denote the locations pertaining to $\mathbf{x}_{\text{mis}}$ and $\mathbf{x}_{\text{obs}}$, respectively, where $k$ denotes the number of missing values (or the dimensionality of $\mathbf{x}_{\text{mis}}$). Furthermore, let $C = \{\mathbf{x}_{\text{obs}}, \mathbf{c}_{\text{obs}}\}$ be the observed context set. Then, following latent neural processes (Foong et al., 2020), we obtain

$$\hat{p}_\varphi(\mathbf{x}_{\text{mis}} \mid \mathbf{c}_{\text{mis}}, C) = \int \hat{p}_\alpha(\mathbf{x}_{\text{mis}} \mid \mathbf{c}_{\text{mis}}, \tilde{\mathbf{z}})\hat{p}_\beta(\tilde{\mathbf{z}} \mid C)d\tilde{\mathbf{z}}$$

$$= \int \hat{p}_\beta(\tilde{\mathbf{z}} \mid C) \prod_{i=1}^{k} \hat{p}_\alpha(x_{\text{mis},i} \mid c_{\text{mis},i}, \tilde{\mathbf{z}})d\tilde{\mathbf{z}}. \tag{7}$$

Figure 2: Plate diagram

Here we have assumed conditional independence of each $x_{\text{mis},i}$ given $c_{\text{mis},i}$ and $\tilde{\mathbf{z}}$, which allows for the joint distribution to factorize into a product of its marginals. Note that this factorization directly inherits the consistency properties from neural processes, as established by Garnelo et al. (2018) and Dubois et al. (2020), ensuring a consistent distribution representation. The associated plate diagram is given in Figure 2. To fully specify the model, we utilize the following:

- **Encoder** $\hat{p}_\beta(\tilde{\mathbf{z}} \mid C)$, which provides a distribution over the latent variables $\tilde{\mathbf{z}}$ having observed the context set $C$. The encoder is parameterized to be permutation invariant to correctly treat $C$ as a set (as required by NPs).

- **Decoder** $\hat{p}_\alpha(x_{\text{mis},i} \mid c_{\text{mis},i}, \tilde{\mathbf{z}})$, which provides a predictive distribution over each missing value $x_{\text{mis},i}$ conditioned on $\tilde{\mathbf{z}}$ and the missing location $c_{\text{mis},i}$. In practice, this distribution is assumed to be a Gaussian, and the parameters $\alpha$ denote the predicted mean and variance.

The likelihood given in Equation (7) is not analytically tractable. Therefore, following Foong et al. (2020), we estimate $\hat{p}_\varphi(\mathbf{x}_{\text{mis}} \mid \mathbf{c}_{\text{mis}}, C)$ using $m$ Monte Carlo samples $\tilde{\mathbf{z}}_1, \ldots, \tilde{\mathbf{z}}_m \sim \hat{p}_\beta(\tilde{\mathbf{z}} \mid C)$ as

$$\log \hat{p}_\varphi(\mathbf{x}_{\text{mis}} \mid \mathbf{c}_{\text{mis}}, C) \approx \log \left( \frac{1}{m} \sum_{j=1}^{m} \prod_{i=1}^{k} \hat{p}_\alpha\big(x_{\text{mis},i} \mid c_{\text{mis},i}, \tilde{\mathbf{z}}_j\big) \right). \tag{8}$$

This can directly be used with standard optimizers (Kingma, 2014) to learn the model parameters.

As NPs are meta-learning models, we can utilize them to amortize over the proportion of missing values $\varepsilon$. Doing so is beneficial in cases where inference is required on multiple datasets with varying proportions of missing values, so as to avoid re-training for each $\varepsilon$. Assuming $p(\varepsilon)$ to be the distribution of the missingness proportion, we can consider each sample from $p(\varepsilon)$ to be one task when training RISE. Specifically, this can be done by first initializing the parameters of RISE, and then repeating the following: (i) Sample $\varepsilon \sim p(\varepsilon)$, and (ii) Perform Steps 2-7 from Algorithm 1. We name this variant of our method as RISE-Meta. For each sample from the imputation model, we obtain a posterior distribution via the inference network, thus resulting in an ensemble of posterior distributions across all samples. In Section 5.3, we test the ability of RISE-Meta to generalize to unknown levels of missing values.

---

**Algorithm 1** RISE (training)

---

**Require:** Simulator $p(\cdot \mid \theta)$, prior $p(\theta)$, iterations $n_{\text{iter}}$, missingness degree $\varepsilon$
1: Initialize parameters $\phi, \varphi$ of RISE
2: **for** $k = 1, \ldots, n_{\text{iter}}$ **do**
3:     Sample $(\mathbf{x}, \theta) \sim p(\cdot \mid \theta)p(\theta)$
4:     Create mask $\mathbf{s}$ wrt $\varepsilon$ and MCAR/-MAR/MNAR
5:     Compute $\ell_{\text{RISE}}$ using Equation (6)
6:     $\phi, \varphi \leftarrow \texttt{optimize}(\ell_{\text{RISE}}; \phi, \varphi)$
7: **end for**

---

## 4 RELATED WORK

**Missing data in SBI.** Wang et al. (2024) attempt to handle missing data by augmenting the missing values with, e.g. zeros or sample mean, and subsequently training NPE with a binary mask indicator, but this approach can lead to biased posterior estimates, as we saw in Figure 1 and Section 3.1. Wang et al. (2022; 2023) propose imputing missing values by sampling from a kernel density estimate of the training data or using a nearest-neighbor search, and training the NPE model using augmented simulations. However, these approaches neglect the missingness mechanisms, which can distort the relationships between variables (Graham et al., 2007) and are not scalable to higher dimensions. Lueckmann et al. (2017a) learn an imputation model agnostic of the missingness mechanism. More recently, Gloeckler et al. (2024) have proposed a transformer-based architecture for SBI that can potentially handle conditioning on data with missing values. This method can perform arbitrary conditioning and evaluation, i.e. for a given $\mathbf{x} = [\mathbf{x}_{\text{obs}}, \mathbf{x}_{\text{mis}}]$, it first estimates the imputation distribution, i.e. $p(\mathbf{x}_{\text{mis}} \mid \mathbf{x}_{\text{obs}})$, and then estimates the posterior distribution $p(\theta \mid \mathbf{x}_{\text{obs}}, \mathbf{x}_{\text{mis}})$. However, it does not model the mechanism underlying the missing data and is thus not equipped to handle the MNAR settings. In contrast, RISE incorporates the missingness mechanism during its training and is therefore able to estimate the full posterior distribution, accounting for all variables.

**Deep imputation methods.** There is a growing body of work on imputing missing data using deep generative models. These include using GANs for missing data under MCAR assumption (Yoon et al.,

2018; Li et al., 2019), and VAEs under MAR assumption (Mattei and Frellsen, 2019; Nazabal et al., 2020). Deep generative models have also been studied under MNAR assumption (Ghalebikesabi et al., 2021a; Gong et al., 2021; Ipsen et al., 2020; Ma and Zhang, 2021). We contribute to this line of work by using latent NPs to handle missing data under all the three missingness assumptions. Instead of learning an imputation model, Smieja et al. (2018) propose replacing a typical neuron's response in the first hidden layer by its expected value to process missing data in neural networks.

## 5 EXPERIMENTS

In this section, we assess the significance of RISE via detailed empirical investigations. Our first objective is to demonstrate that RISE yields posteriors that are robust to missing values in the data compared to baseline methods (see Section 5.1 and Section 5.2). Secondly, we aim to test the generalization capability of RISE-Meta in cases where the proportion of missing values in the data is not known *a priori* (Section 5.3). Thirdly, as learning the imputation model accurately is central to RISE's performance, we aim to validate that employing a NP-based imputation model in RISE yields state-of-the-art results when imputing real-world datasets. Finally, we intend to provide some experimental evidence where learning the inference and imputation components jointly, as is done in RISE, performs better than learning them separately.

Our experiments are organized as follows. We first provide results on SBI benchmarks in Sections 5.1, 5.2, and 5.3. In Section 5.4, we report our ablation studies to evaluate the imputation performance of RISE on real-world bioactivity datasets.

**Performance metrics.** We evaluate the accuracy of the posterior using the following metrics: (i) the nominal log posterior probability of true parameters (NLPP), (ii) the classifier two-sample test (C2ST) score (Lopez-Paz and Oquab, 2017), and (iii) the maximum mean discrepancy (MMD) (Gretton et al., 2012). The MMD and C2ST metrics are computed between the posterior samples obtained under missing data (either using RISE or the baseline methods) and samples from a reference NPE posterior under no missing data. We use a radial basis function kernel for computing the MMD, and set its lengthscale using the median heuristic (Gretton et al., 2012) on the reference posterior samples.

**Baselines.** We evaluate RISE's performance against baselines derived from NPE (Greenberg et al., 2019). These include the mask-based method proposed by Wang et al. (2024), and NPE-NN that combines NPE with a feed-forward neural network for joint training and imputation (Lueckmann et al., 2017a). While NPE-NN shares RISE's joint training paradigm, it performs single imputation rather than the multiple imputation approach used in RISE. We also compare against Simformer (Gloeckler et al., 2024), a recent diffusion and transformer-based approach for posterior estimation.

**Implementation.** RISE is implemented in PyTorch (Paszke et al., 2019) and utilizes the same training configuration as the competing baselines (see Appendix A.4.4 for details). We take $\varepsilon \in \{10\%, 25\%, 60\%\}$ to test performance from low to high missingness scenarios. We adopt the masking approach as described in Mattei and Frellsen (2019) and Ipsen et al. (2020) for MCAR and MNAR respectively. Specifically, for MCAR we randomly mask $\varepsilon\%$ of the data, and for MNAR we use $\varepsilon$ to compute a masking probability, which is then used to mask data according to their values. This *self-censoring* approach is described in Appendix A.4.3, and leads to a missingness proportion less than (or equal to) $\varepsilon$. We set a simulation budget of $n = 1000$ for all the SBI experiments, and take 1000 samples from the posterior distributions to compute the MMD, C2ST and NLPP. The performance is evaluated over 10 random runs. For further details, see Appendix A.4.

### 5.1 PERFORMANCE ON SBI BENCHMARKS

We evaluate the performance of RISE in settings with missing data using four common benchmark models from the SBI literature, namely, (i) Ricker model: a two parameter simulator from population genetics (Wood, 2010); (ii) Ornstein-Uhlenbeck process (OUP): a two parameter stochastic differential equation model (Chen et al., 2021); (iii) Generalized Linear Model (GLM): a 10 parameter model with Bernoulli observations; and (iv) Gaussian Linear Uniform (GLU): a 10-dimensional Gaussian model with the mean vector as the parameter and a fixed covariance. The models are described in Appendix A.4.1, and the prior distributions we used are reported in Appendix A.4.2.

Table 1: NLPP and C2ST metrics under MCAR and MNAR scenarios, with missing value proportion $\varepsilon$. RISE demonstrates superior posterior estimation performance. For MNAR scenarios, the proportion of missing values averages below $\varepsilon$ due to self-censoring (details in Appendix A.4.3). Note that Simformer results are unavailable for Ricker and OUP due to the lack of official implementation.

| | Dataset | $\epsilon$ | NLPP | | | | C2ST | | | |
|---|---|---|---|---|---|---|---|---|---|---|
| | | | NPE-NN | Wang et al. | Simformer | RISE | NPE-NN | Wang et al. | Simformer | RISE |
| MCAR | GLU | 10% | $-2.51 \pm 0.11$ | $-2.50 \pm 0.10$ | $-2.45 \pm 0.12$ | $\mathbf{-2.31} \pm 0.10$ | $0.87 \pm 0.01$ | $0.87 \pm 0.01$ | $0.85 \pm 0.01$ | $\mathbf{0.83} \pm 0.01$ |
| | | 25% | $-3.92 \pm 0.11$ | $\mathbf{-3.54} \pm 0.17$ | $-3.65 \pm 0.17$ | $-3.71 \pm 0.11$ | $0.90 \pm 0.01$ | $0.92 \pm 0.01$ | $0.91 \pm 0.01$ | $\mathbf{0.89} \pm 0.01$ |
| | | 60% | $-6.37 \pm 0.12$ | $-6.52 \pm 0.17$ | $-6.62 \pm 0.27$ | $\mathbf{-6.21} \pm 0.11$ | $0.98 \pm 0.01$ | $0.97 \pm 0.01$ | $0.96 \pm 0.01$ | $\mathbf{0.93} \pm 0.01$ |
| | GLM | 10% | $-6.57 \pm 0.13$ | $-7.10 \pm 0.11$ | $-6.47 \pm 0.16$ | $\mathbf{-6.32} \pm 0.15$ | $0.84 \pm 0.01$ | $0.86 \pm 0.01$ | $0.84 \pm 0.01$ | $\mathbf{0.80} \pm 0.01$ |
| | | 25% | $-7.72 \pm 0.16$ | $-7.84 \pm 0.17$ | $-7.37 \pm 0.13$ | $\mathbf{-7.22} \pm 0.17$ | $0.93 \pm 0.01$ | $0.94 \pm 0.01$ | $0.92 \pm 0.01$ | $\mathbf{0.91} \pm 0.01$ |
| | | 60% | $-9.02 \pm 0.17$ | $-8.97 \pm 0.15$ | $-8.93 \pm 0.18$ | $\mathbf{-8.71} \pm 0.14$ | $0.99 \pm 0.01$ | $0.99 \pm 0.01$ | $0.98 \pm 0.01$ | $\mathbf{0.97} \pm 0.01$ |
| | Ricker | 10% | $-4.90 \pm 0.16$ | $-4.74 \pm 0.31$ | - | $\mathbf{-4.20} \pm 0.09$ | $0.94 \pm 0.01$ | $0.93 \pm 0.01$ | - | $\mathbf{0.90} \pm 0.01$ |
| | | 25% | $-4.94 \pm 0.17$ | $-5.14 \pm 0.27$ | - | $\mathbf{-4.64} \pm 0.15$ | $0.96 \pm 0.01$ | $0.95 \pm 0.01$ | - | $\mathbf{0.92} \pm 0.01$ |
| | | 60% | $-4.97 \pm 0.17$ | $-5.24 \pm 0.11$ | - | $\mathbf{-4.72} \pm 0.17$ | $0.97 \pm 0.01$ | $0.99 \pm 0.01$ | - | $\mathbf{0.94} \pm 0.01$ |
| | OUP | 10% | $-2.25 \pm 0.18$ | $-2.37 \pm 0.18$ | - | $\mathbf{-2.09} \pm 0.11$ | $0.89 \pm 0.01$ | $0.88 \pm 0.01$ | - | $\mathbf{0.87} \pm 0.01$ |
| | | 25% | $-2.74 \pm 0.18$ | $-2.55 \pm 0.13$ | - | $\mathbf{-2.43} \pm 0.15$ | $0.90 \pm 0.01$ | $0.91 \pm 0.01$ | - | $\mathbf{0.89} \pm 0.01$ |
| | | 60% | $-2.87 \pm 0.19$ | $-2.75 \pm 0.17$ | - | $\mathbf{-2.52} \pm 0.11$ | $0.95 \pm 0.01$ | $0.94 \pm 0.01$ | - | $\mathbf{0.93} \pm 0.01$ |
| MNAR | GLU | 10% | $-2.35 \pm 0.10$ | $-2.42 \pm 0.17$ | $-2.15 \pm 0.10$ | $\mathbf{-1.90} \pm 0.09$ | $0.89 \pm 0.01$ | $0.88 \pm 0.01$ | $0.87 \pm 0.01$ | $\mathbf{0.85} \pm 0.01$ |
| | | 25% | $-3.31 \pm 0.17$ | $-3.67 \pm 0.12$ | $\mathbf{-3.12} \pm 0.12$ | $-3.26 \pm 0.10$ | $0.92 \pm 0.01$ | $0.93 \pm 0.01$ | $0.91 \pm 0.01$ | $\mathbf{0.88} \pm 0.01$ |
| | | 60% | $-5.97 \pm 0.19$ | $-6.03 \pm 0.11$ | $-6.02 \pm 0.12$ | $\mathbf{-5.80} \pm 0.27$ | $0.96 \pm 0.01$ | $0.95 \pm 0.01$ | $0.93 \pm 0.01$ | $\mathbf{0.92} \pm 0.01$ |
| | GLM | 10% | $-6.05 \pm 0.27$ | $-5.98 \pm 0.22$ | $-6.17 \pm 0.18$ | $\mathbf{-5.82} \pm 0.11$ | $0.89 \pm 0.01$ | $0.90 \pm 0.01$ | $0.87 \pm 0.01$ | $\mathbf{0.85} \pm 0.01$ |
| | | 25% | $-6.47 \pm 0.14$ | $-6.51 \pm 0.32$ | $-6.57 \pm 0.14$ | $\mathbf{-6.12} \pm 0.15$ | $0.94 \pm 0.01$ | $0.95 \pm 0.01$ | $0.92 \pm 0.01$ | $\mathbf{0.89} \pm 0.01$ |
| | | 60% | $-7.78 \pm 0.37$ | $-8.38 \pm 0.12$ | $-7.56 \pm 0.15$ | $\mathbf{-7.11} \pm 0.17$ | $0.97 \pm 0.01$ | $0.98 \pm 0.01$ | $0.96 \pm 0.01$ | $\mathbf{0.95} \pm 0.01$ |
| | Ricker | 10% | $-4.67 \pm 0.24$ | $-4.35 \pm 0.13$ | - | $\mathbf{-4.10} \pm 0.18$ | $0.94 \pm 0.01$ | $0.94 \pm 0.01$ | - | $\mathbf{0.92} \pm 0.01$ |
| | | 25% | $-4.91 \pm 0.20$ | $-5.05 \pm 0.18$ | - | $\mathbf{-4.75} \pm 0.23$ | $0.95 \pm 0.01$ | $0.96 \pm 0.01$ | - | $\mathbf{0.93} \pm 0.01$ |
| | | 60% | $-5.25 \pm 0.21$ | $-5.12 \pm 0.16$ | - | $\mathbf{-4.82} \pm 0.26$ | $0.97 \pm 0.01$ | $0.99 \pm 0.01$ | - | $\mathbf{0.95} \pm 0.01$ |
| | OUP | 10% | $-2.21 \pm 0.13$ | $-2.32 \pm 0.18$ | - | $\mathbf{-2.10} \pm 0.12$ | $0.93 \pm 0.01$ | $0.92 \pm 0.01$ | - | $\mathbf{0.88} \pm 0.01$ |
| | | 25% | $-2.42 \pm 0.17$ | $-2.57 \pm 0.11$ | - | $\mathbf{-2.24} \pm 0.17$ | $0.97 \pm 0.01$ | $0.95 \pm 0.01$ | - | $\mathbf{0.93} \pm 0.01$ |
| | | 60% | $-2.92 \pm 0.15$ | $-2.79 \pm 0.21$ | - | $\mathbf{-2.47} \pm 0.21$ | $0.99 \pm 0.01$ | $0.99 \pm 0.01$ | - | $\mathbf{0.97} \pm 0.01$ |

The results for NLPP, C2ST are shown in Table 1 and MMD in Table 5, comparing performance across varying missingness levels $\varepsilon$ under both MCAR and MNAR conditions. We observe that RISE achieves the lowest values of C2ST across missingness types, thus outperforming the baselines in estimating the posterior distributions. As $\varepsilon$ increases, the gap between RISE and the baselines increases, indicating that RISE is able to better handle high missingness levels in the data. As a sanity check, we also investigate the imputation capability of RISE . Figure 5 shows that RISE achieves better imputation, which then naturally translates to robust posterior estimation. The difference in performance is more stark in the MNAR case, as expected, since the baseline methods do not explicitly model the missingness mechanism.

## 5.2 HODGKIN-HUXLEY MODEL

We now apply RISE on a real-world computational neuroscience simulator (Hodgkin and Huxley, 1952), namely the Hodgkin-Huxley model, which is a popular example in the SBI literature (Lueckmann et al., 2017b; Gao et al., 2023; Gloeckler et al., 2023). The aim is to infer the posterior over two parameters given the data of dimension 1200 (see Appendix A.4.1 for the model description).

We set uniform priors and perform inference under different values of $\varepsilon$ and missingness assumptions, similar to Section 5.1. Figure 3 shows that RISE's posteriors are robust to increasing proportions of missing values as they stay around the true parameter value as compared to NPE-NN. We also evaluate the expected coverage of the posterior in Appendix B.2, which demonstrates that RISE produces conservative posterior approximations and achieves better calibration than NPE-NN.

## 5.3 GENERALIZING ACROSS UNKNOWN LEVELS OF MISSINGNESS

Next, we test the generalization capability of our method to unknown levels of missing values. We perform meta-learning over different proportions of missing values $\varepsilon$ in the dataset (termed RISE-Meta). For training RISE-Meta, we take the distribution of $p(\varepsilon)$ to be an equiprobable discrete distribution on the set $\{10\%, 25\%, 60\%\}$. We also train NPE-NN with a missingness degree of $60\%$ as a baseline. We evaluate all the methods over 100 samples of varying missingness proportion $\varepsilon \sim \mathcal{U}([0,1])$. Figure 4 shows the MMD results on GLM, GLU, Ricker and OUP tasks. We observe

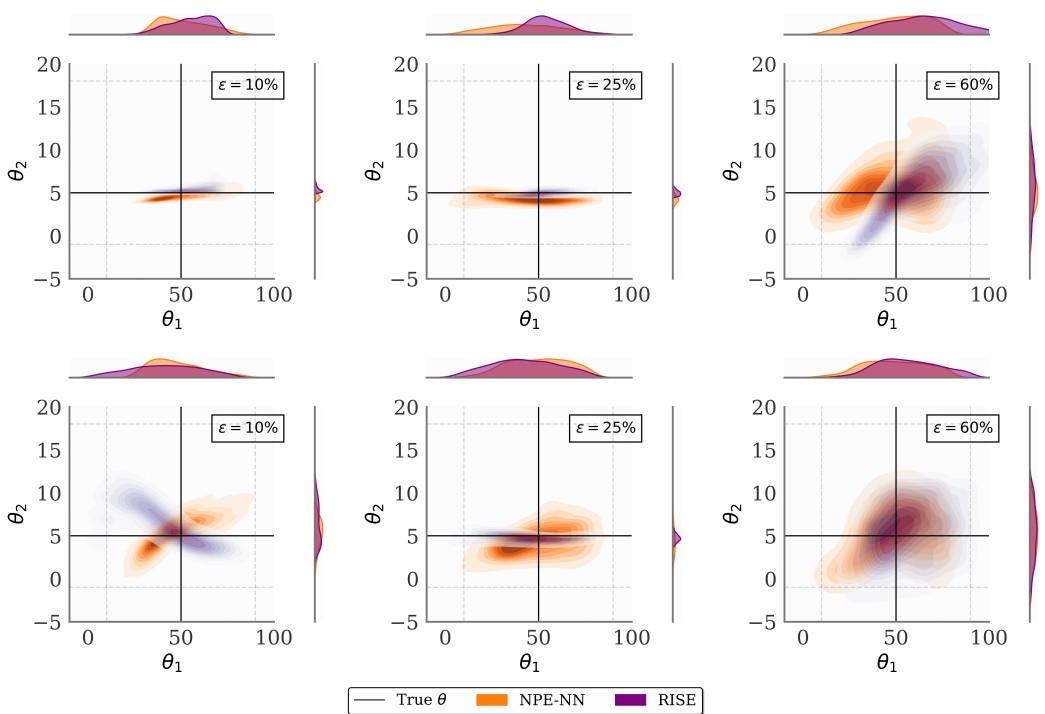

Figure 3: Posterior estimates for the Hodgkin-Huxley model under MCAR (*top row*) and MNAR (*bottom row*) with varying proportions of missing values in the data (denoted by $\varepsilon$). The posteriors obtained from RISE stay close to the true parameter (denoted by the black lines) for all values of $\varepsilon$, while those from the baseline methods move further away as $\varepsilon$ increases.

that RISE-Meta achieves the lowest MMD values for both the tasks, thus demonstrating its ability to better generalize to unknown levels of missing values in the data.

## 5.4 ABLATION STUDIES

**Imputation performance on real-world datasets.** We now look at how the neural process-based imputation model in RISE performs on real-world datasets. The task is to predict and impute bioactivity data on Adrenergic receptor assays (Whitehead et al., 2019) and Kinase assays (Martin et al., 2017) from the field of drug discovery. The Kinase test data consists of outliers, unlike the training data, which makes imputation challenging. We can therefore use such data to assess the generalization capabilities of RISE. We compare the RISE imputation method to other methods from this field such as QSAR (Cherkasov et al., 2014), Conduit[2] (Whitehead et al., 2019), and Collective Matrix Factorization (CMF) (Singh and Gordon, 2008). We also include a standard deep neural network (DNN) and a vanilla neural process as baselines. Table 2 (left) reports the coefficient of determination $R^2$ (Wright, 1921) between the true and the predicted assays. We observe that RISE achieves state-of-the-art results in these tasks, demonstrating the efficacy of the neural processes-based imputation model.

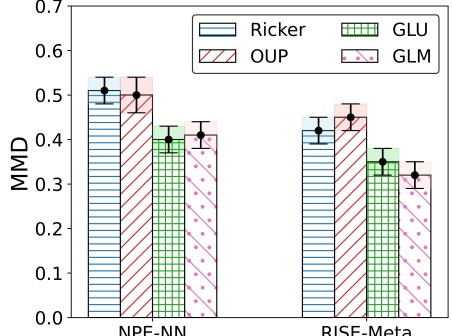

Figure 4: Generalizing over missingness.

---

[2]Since, the official implementation is unavailable, we use the re-implementation provided here: https://github.com/PenelopeJones/neural_processes.

Table 2: **Ablation studies.** (*left*) $R^2$ scores (↑) on the bioactivity datasets. (*right*) RMSE (↓) across different proportions of missingness ($\varepsilon$) for comparing the effect of joint versus separate learning.

| Method | Adrenergic | Kinase |
|---|---|---|
| QSAR | (N/A) | $-0.19 \pm 0.01$ |
| CMF | $0.59 \pm 0.02$ | $-0.11 \pm 0.01$ |
| DNN | $0.60 \pm 0.05$ | $0.11 \pm 0.01$ |
| NP | $0.61 \pm 0.03$ | $0.17 \pm 0.04$ |
| Conduilt | $0.62 \pm 0.04$ | $0.22 \pm 0.03$ |
| CNP | $0.65 \pm 0.04$ | $0.24 \pm 0.02$ |
| RISE | $\mathbf{0.67} \pm 0.03$ | $\mathbf{0.26} \pm 0.03$ |

| Missingness ($\epsilon$) | Method | GLM | GLU |
|---|---|---|---|
| 10% | NPE-RF-Sep | $0.69 \pm 0.03$ | $0.44 \pm 0.02$ |
| | RISE-Sep | $0.67 \pm 0.03$ | $0.43 \pm 0.02$ |
| | RISE | $\mathbf{0.65} \pm 0.04$ | $\mathbf{0.41} \pm 0.01$ |
| 25% | NPE-RF-Sep | $1.02 \pm 0.05$ | $0.48 \pm 0.02$ |
| | RISE-Sep | $0.99 \pm 0.03$ | $0.45 \pm 0.02$ |
| | RISE | $\mathbf{0.93} \pm 0.06$ | $\mathbf{0.43} \pm 0.02$ |
| 60% | NPE-RF-Sep | $1.34 \pm 0.10$ | $0.64 \pm 0.02$ |
| | RISE-Sep | $1.31 \pm 0.03$ | $0.58 \pm 0.03$ |
| | RISE | $\mathbf{1.27} \pm 0.01$ | $\mathbf{0.56} \pm 0.03$ |

**Joint vs separate learning.** This experiment involves investigating the impact of training the imputation and the inference model in RISE jointly (as we proposed) versus separately (termed RISE-Sep). We also include another baseline termed NPE-RF-Sep where a random forest (RF) model is first used for imputation, followed by NPE. Table 2 (right) reports the RMSE values on GLM and GLU tasks for different missingness proportion $\varepsilon$. We observe that training the imputation and inference networks jointly yields improvement in performance over training them separately.

We report the results from additional ablation studies for runtime comparisons, flow architecture, and simulation budget in Appendix C.

## 6 CONCLUSION AND LIMITATIONS

We analyzed the problem of performing SBI under missing data and showed that inaccurately imputing the missing values may lead to bias in the resulting posterior distributions. We then proposed RISE as a method that aims to reduce this bias under different notions of the underlying missingness mechanism. RISE combines the inference network of NPE with an imputation model based on neural processes (NPs) to achieve robustness to missing data whilst being amortized. Additionally, RISE can be trained in a meta-learning manner over the proportion of missing values in the data, thus allowing for amortization across datasets with varying levels of missingness. While RISE offers substantial advantages, there are limitations to address. RISE inherits the issues of NPE and may yield posteriors that are not well-calibrated (see, e.g., Hermans et al. (2022)). Moreover, the normality assumption in NPs may exhibit limited expressivity in practice when learning complex imputation distributions.

## ACKNOWLEDGEMENTS

YV and VG acknowledge support from the Research Council of Finland for the "Human-steered next-generation machine learning for reviving drug design" project (grant decision 342077). VG also acknowledges the support from Jane and Aatos Erkko Foundation (grant 7001703) for "Biodesign: Use of artificial intelligence in enzyme design for synthetic biology". AB is supported by the Research Council of Finland grant no. 362534. The experiments were performed using resources provided by the Aalto University Science-IT project and CSC – IT Center for Science, Finland. YV thanks Priscilla Ong for highlighting relevant related works on deep imputation models and for insightful discussions on various missingness mechanisms.

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

# A    APPENDIX

Appendix A.1 discusses the aspect of handling multiple observations and model misspecfication in the context of RISE . In Appendix A.2, we present the proofs for Proposition 1 and Proposition 2. Appendix A.3 provides a background on neural processes, and Appendix A.4 presents the implementation details for the experiments of Section 5. Appendix B contains additional metrics, coverage plots and visulaizations. Appendix C reports the results from additional ablation studies.

## A.1    DISCUSSION

**Handling multiple observations.**    Although so far we have focused on the single observation case where we have one data vector $\mathbf{x}$ for each $\theta$, RISE can straightforwardly be extended to the multiple observations case where we obtain $\mathbf{x}^{(1:m)} = (\mathbf{x}_1, \ldots, \mathbf{x}_m)$ for each $\theta$. Then, $\mathbf{x}^{(1:m)} = (\mathbf{x}_{\text{obs}}^{(1:m)}, \mathbf{x}_{\text{mis}}^{(1:m)})$, and the objective for RISE becomes

$$\arg\min_{\phi,\varphi,\kappa} -\mathbb{E}_{(\mathbf{x}_{\text{obs}}^{(1:m)},\theta)\sim p_{\text{true}}}\mathbb{E}_{\mathbf{x}_{\text{mis}}^{(1:m)}\sim\prod_{i=1}^m p(\mathbf{x}_{\text{mis}}^{(i)}|\mathbf{x}_{\text{obs}}^{(i)})}\left[\frac{1}{m}\sum_{i=1}^m \log\underbrace{\hat{p}_\varphi(\mathbf{x}_{\text{mis}}^{(i)} \mid \mathbf{x}_{\text{obs}}^{(i)})}_{\text{(imputation)}} + \log\underbrace{q_\phi(\theta \mid \eta_\kappa(\mathbf{x}_{\text{obs}}^{1:m}, \mathbf{x}_{\text{mis}}^{1:m}))}_{\text{(inference)}}\right].$$

Note that we can summarize the data using the network $\eta_\kappa$ (for instance, a deep set (Zaheer et al., 2017)) before passing the data into the inference network, which is standard practice when using NPE with multiple observations (Chan et al., 2018). Alternatively, one could use recent extensions based on score estimation (Geffner et al., 2023; Linhart et al., 2024) as well.

**Handling model misspecification.**    We conjecture that replacing the inference network in RISE from the usual NPE to a robust variant such as the method of Ward et al. (2022) or Huang et al. (2023) would help in addressing model misspecification issues. It would be an interesting avenue for future research to see how to train these robust NPE methods jointly with the imputation network of RISE, and how effective such an approach is. One way is to assume a certain error model over the observed data $\mathbf{x}$, corrupt the data to $\tilde{\mathbf{x}}$ by adding a Gaussian noise, and infer the correct $\theta$ via the inference network. This can be formulated as

$$\text{argmin}_{\phi,\varphi} -\mathbb{E}_{(\mathbf{x}_{\text{obs}},\theta)\sim p(\mathbf{x}_{\text{obs}},\theta),\tilde{\mathbf{x}}_{\text{obs}}\sim\mathcal{N}(\mathbf{x}_{\text{obs}},\sigma^2),\tilde{\mathbf{x}}_{\text{mis}}\sim p_{\text{true}}(\tilde{\mathbf{x}}_{\text{mis}}|\tilde{\mathbf{x}}_{\text{obs}},\theta)}\left[\log\hat{p}_\varphi(\tilde{\mathbf{x}}_{\text{mis}} \mid \tilde{\mathbf{x}}_{\text{obs}}) + \log q_\phi(\theta \mid \tilde{\mathbf{x}}_{\text{obs}}, \tilde{\mathbf{x}}_{\text{mis}})\right]. \tag{9}$$

Moreover, our method can also be readily extended to incorporate prior mis-specification:

$$\text{argmin}_{\phi,\varphi} -\mathbb{E}_{(\mathbf{x}_{\text{obs}},\theta)\sim p(\mathbf{x}_{\text{obs}},\theta),\tilde{\theta}\sim\mathcal{N}(\theta,\sigma^2),\tilde{\mathbf{x}}_{\text{mis}}\sim p_{\text{true}}(\tilde{\mathbf{x}}_{\text{mis}}|\tilde{\mathbf{x}}_{\text{obs}},\tilde{\theta})}\left[\log\hat{p}_\varphi(\tilde{\mathbf{x}}_{\text{mis}} \mid \tilde{\mathbf{x}}_{\text{obs}}) + \log q_\phi(\theta \mid \tilde{\mathbf{x}}_{\text{obs}}, \tilde{\mathbf{x}}_{\text{mis}})\right]. \tag{10}$$

## A.2    PROOFS

### A.2.1    PROOF FOR PROPOSITION 1

*Proof.* Using Equation (2) and Equation (3), we note that

$$\mathbb{E}_{\theta\sim p_{\text{SBI}}(\theta \mid \mathbf{x}_{\text{obs}})}[\theta] - \mathbb{E}_{\theta\sim\hat{p}_{\text{SBI}}(\theta \mid \mathbf{x}_{\text{obs}})}[\theta]$$

$$= \int \theta p_{\text{SBI}}(\theta \mid \mathbf{x}_{\text{obs}})d\theta - \int \theta\hat{p}_{\text{SBI}}(\theta \mid \mathbf{x}_{\text{obs}})d\theta$$

$$= \int \theta \left[p_{\text{SBI}}(\theta \mid \mathbf{x}_{\text{obs}}) - \hat{p}_{\text{SBI}}(\theta \mid \mathbf{x}_{\text{obs}})\right]d\theta$$

$$= \int \theta \left[\int p_{\text{SBI}}(\theta \mid \mathbf{x}_{\text{obs}}, \mathbf{x}_{\text{mis}})p_{\text{true}}(\mathbf{x}_{\text{mis}} \mid \mathbf{x}_{\text{obs}})d\mathbf{x}_{\text{mis}} - \int p_{\text{SBI}}(\theta \mid \mathbf{x}_{\text{obs}}, \mathbf{x}_{\text{mis}})\hat{p}(\mathbf{x}_{\text{mis}} \mid \mathbf{x}_{\text{obs}})d\mathbf{x}_{\text{mis}}\right]d\theta$$

$$= \int \theta \int p_{\text{SBI}}(\theta \mid \mathbf{x}_{\text{obs}}, \mathbf{x}_{\text{mis}})\left[p_{\text{true}}(\mathbf{x}_{\text{mis}} \mid \mathbf{x}_{\text{obs}}) - \hat{p}(\mathbf{x}_{\text{mis}} \mid \mathbf{x}_{\text{obs}})\right]d\mathbf{x}_{\text{mis}}d\theta .$$

Thus to ensure that the bias is zero, we require that $\hat{p}(\mathbf{x}_{\text{mis}} \mid \mathbf{x}_{\text{obs}})$ be aligned with $p_{\text{true}}(\mathbf{x}_{\text{mis}} \mid \mathbf{x}_{\text{obs}})$.    □

### A.2.2 PROOF FOR PROPOSITION 2

*Proof.* Recall our optimization problem from Equation (5):

$$\arg\min_{\psi} \mathbb{E}_{\mathbf{x}_{\text{obs}} \sim p_{\text{true}}} \text{KL}[p_{\text{true}}(\theta \mid \mathbf{x}_{\text{obs}}) \parallel r_{\psi}(\theta \mid \mathbf{x}_{\text{obs}})] .$$

Expanding the KL term, we note that the above is equivalent to

$$\arg\min_{\psi} \mathbb{E}_{\mathbf{x}_{\text{obs}} \sim p_{\text{true}}} \mathbb{E}_{\theta \sim p_{\text{true}}(\theta|\mathbf{x}_{\text{obs}})} \log\left(\frac{p_{\text{true}}(\theta \mid \mathbf{x}_{\text{obs}})}{r_{\psi}(\theta \mid \mathbf{x}_{\text{obs}})}\right) .$$

Since $p_{\text{true}}(\theta \mid \mathbf{x}_{\text{obs}})$ does not depend on $\psi$, we immediately note that the problem is equivalent to

$$\text{argmin}_{\psi} \ \mathbb{E}_{\mathbf{x}_{\text{obs}} \sim p_{\text{true}}} \mathbb{E}_{p_{\text{true}}(\theta|\mathbf{x}_{\text{obs}})} [-\log r_{\psi}(\theta \mid \mathbf{x}_{\text{obs}})]$$
$$= \arg\max_{\psi} \ \mathbb{E}_{(\mathbf{x}_{\text{obs}}, \theta) \sim p_{\text{true}}} [\log r_{\psi}(\theta \mid \mathbf{x}_{\text{obs}})] .$$

We now obtain a lower bound for $\mathbb{E}_{(\mathbf{x}_{\text{obs}}, \theta) \sim p_{\text{true}}} [\log r_{\psi}(\theta \mid \mathbf{x}_{\text{obs}})]$. Formally, we have

$$\mathbb{E}_{(\mathbf{x}_{\text{obs}}, \theta) \sim p_{\text{true}}} [\log r_{\psi}(\theta \mid \mathbf{x}_{\text{obs}})] = \mathbb{E}_{(\mathbf{x}_{\text{obs}}, \theta) \sim p_{\text{true}}} \log \int r_{\psi}(\theta, \mathbf{x}_{\text{mis}} \mid \mathbf{x}_{\text{obs}}) d\mathbf{x}_{\text{mis}}$$

$$= \mathbb{E}_{(\mathbf{x}_{\text{obs}}, \theta) \sim p_{\text{true}}} \log \int \frac{p(\mathbf{x}_{\text{mis}} \mid \mathbf{x}_{\text{obs}}) r_{\psi}(\theta, \mathbf{x}_{\text{mis}} \mid \mathbf{x}_{\text{obs}})}{p(\mathbf{x}_{\text{mis}} \mid \mathbf{x}_{\text{obs}})} d\mathbf{x}_{\text{mis}}$$

$$\geq \mathbb{E}_{(\mathbf{x}_{\text{obs}}, \theta) \sim p_{\text{true}}} \mathbb{E}_{\mathbf{x}_{\text{mis}} \sim p(\mathbf{x}_{\text{mis}}|\mathbf{x}_{\text{obs}})} \left[ \log \frac{r_{\psi}(\theta, \mathbf{x}_{\text{mis}} \mid \mathbf{x}_{\text{obs}})}{p(\mathbf{x}_{\text{mis}} \mid \mathbf{x}_{\text{obs}})} \right]$$

$$= \mathbb{E}_{(\mathbf{x}_{\text{obs}}, \theta) \sim p_{\text{true}}} \mathbb{E}_{\mathbf{x}_{\text{mis}} \sim p(\mathbf{x}_{\text{mis}}|\mathbf{x}_{\text{obs}})} \left[ \log \frac{r_{\psi}(\mathbf{x}_{\text{mis}} \mid \mathbf{x}_{\text{obs}}) r_{\psi}(\theta \mid \mathbf{x}_{\text{obs}}, \mathbf{x}_{\text{mis}})}{p(\mathbf{x}_{\text{mis}} \mid \mathbf{x}_{\text{obs}})} \right] ,$$

where we invoked the Jensen's inequality to swap the log and the conditional expectation. Splitting parameters $\psi$ into imputation parameters $\varphi$ and inference parameters $\phi$, and denoting the corresponding imputation and inference networks by $\hat{p}_{\varphi}$ and $q_{\phi}$ respectively, we immediately get

$$\mathbb{E}_{(\mathbf{x}_{\text{obs}}, \theta) \sim p_{\text{true}}} [\log r_{\phi, \varphi}(\theta \mid \mathbf{x}_{\text{obs}})] \geq \mathbb{E}_{(\mathbf{x}_{\text{obs}}, \theta) \sim p_{\text{true}}} \mathbb{E}_{\mathbf{x}_{\text{mis}} \sim p(\mathbf{x}_{\text{mis}}|\mathbf{x}_{\text{obs}})} \left[ \log \frac{\hat{p}_{\varphi}(\mathbf{x}_{\text{mis}} \mid \mathbf{x}_{\text{obs}}) q_{\phi}(\theta \mid \mathbf{x}_{\text{obs}}, \mathbf{x}_{\text{mis}})}{p(\mathbf{x}_{\text{mis}} \mid \mathbf{x}_{\text{obs}})} \right] .$$

Thus, we obtain the following variational objective:

$$\text{argmax}_{\phi, \varphi} \mathbb{E}_{(\mathbf{x}_{\text{obs}}, \theta) \sim p_{\text{true}}} \mathbb{E}_{\mathbf{x}_{\text{mis}} \sim p(\mathbf{x}_{\text{mis}}|\mathbf{x}_{\text{obs}})} \left[ \log \frac{\hat{p}_{\varphi}(\mathbf{x}_{\text{mis}} \mid \mathbf{x}_{\text{obs}}) q_{\phi}(\theta \mid \mathbf{x}_{\text{obs}}, \mathbf{x}_{\text{mis}})}{p(\mathbf{x}_{\text{mis}} \mid \mathbf{x}_{\text{obs}})} \right]$$

$$= \text{argmax}_{\phi, \varphi} \mathbb{E}_{(\mathbf{x}_{\text{obs}}, \theta) \sim p_{\text{true}}} \left( \mathbb{E}_{\mathbf{x}_{\text{mis}} \sim p(\mathbf{x}_{\text{mis}}|\mathbf{x}_{\text{obs}})} [\log \hat{p}_{\varphi}(\mathbf{x}_{\text{mis}} \mid \mathbf{x}_{\text{obs}}) + \log q_{\phi}(\theta \mid \mathbf{x}_{\text{obs}}, \mathbf{x}_{\text{mis}})] + \mathbb{H}(p(\mathbf{x}_{\text{mis}} \mid \mathbf{x}_{\text{obs}})) \right)$$

$$= \text{argmax}_{\phi, \varphi} \mathbb{E}_{(\mathbf{x}_{\text{obs}}, \theta) \sim p_{\text{true}}} \mathbb{E}_{\mathbf{x}_{\text{mis}} \sim p(\mathbf{x}_{\text{mis}}|\mathbf{x}_{\text{obs}})} \left[ \log \underbrace{\hat{p}_{\varphi}(\mathbf{x}_{\text{mis}} \mid \mathbf{x}_{\text{obs}})}_{\text{imputation}} + \log \underbrace{q_{\phi}(\theta \mid \mathbf{x}_{\text{obs}}, \mathbf{x}_{\text{mis}})}_{\text{inference}} \right] ,$$

since the entropy term $\mathbb{H}(p(\mathbf{x}_{\text{mis}} \mid \mathbf{x}_{\text{obs}})$ does not depend on the optimization variables $\phi$ and $\varphi$.

$\square$

### A.3 NEURAL PROCESS

Neural Process (Garnelo et al., 2018; Foong et al., 2020) models the predictive distribution over target locations $\mathbf{x}_t$ by, (i) constructing a learnable mapping $f_{\gamma}$ from the context set $(\mathbf{x}_c, \mathbf{y}_c)$ to a latent representation $\mathbf{r}$ as,

$$\mathbf{r} = f_{\gamma}(\mathbf{x}_c, \mathbf{y}_c) \tag{11}$$

and then (ii) utilizing the representation $\mathbf{r}$ to approximate the predictive distribution, given the target locations $\mathbf{x}_t$, via a learnable decoder $g_\omega$ as,

$$p(\mathbf{y}_t \mid \mathbf{x}_t, \mathbf{r}) = g_\omega(\mathbf{r}, \mathbf{x}_t) \tag{12}$$

where $\mathbf{x}_c, \mathbf{x}_t \in \mathcal{X} \subseteq \mathbb{R}^{d_x}$ are the input vectors (often locations or positions) and $\mathbf{y}_c, \mathbf{y}_t \in \mathcal{Y} \subseteq \mathbb{R}^{d_y}$ the output vectors. In practice, the predictive distribution is often assumed to factorize as a product of Gaussians:

$$p(\mathbf{y}_t \mid \mathbf{x}_t, \mathbf{r}) = \prod_{m=1}^{M} p(\mathbf{y}_{t,m} \mid \mathbf{x}_{t,m}, \mathbf{r}) = \prod_{m=1}^{M} \mathcal{N}(\mathbf{y}_{t,m} \mid \mu_{\omega,m}, \sigma_{\omega,m}^2) \tag{13}$$

where $\mu_{\omega,m}, \sigma_{\omega,m} = g_\omega(\mathbf{r}, \mathbf{x}_{t,m})$. For a fixed context $(\mathbf{x}_c, \mathbf{y}_c)$, using Kolmogorov's extension theorem (Oksendal, 2013), the collection of these finite dimensional distributions defines a stochastic process if these are consistent under (i) permutations of any entries of $(\mathbf{x}_t, \mathbf{y}_t)$ and (ii) marginalisations of any entries of $\mathbf{y}_t$.

## A.4 IMPLEMENTATION DETAILS

This section is arranged as follows:

- Appendix A.4.1: Description of SBI benchmarking simulators
- Appendix A.4.2: Prior distributions used for the SBI experiments
- Appendix A.4.3: Procedure for creating the missingness mask under MCAR and MNAR
- Appendix A.4.4: Details of the neural network settings.

### A.4.1 MODEL DESCRIPTIONS

**Ricker model** simulates the temporal evolution of population size in ecological systems. In this model, the population size $N_t$ at time $t$ evolves as $N_{t+1} = N_t \exp(\theta_1) \exp(N_t + e_t), t \in \{1, \ldots, T\}$. The parameter $\exp(\theta_1)$ represents the growth rate, while $e_t$ denote independent and identically distributed Gaussian noise terms with zero mean and variance $\sigma_e^2$. The initial population size is set to $N_0 = 1$. Observations $x_t$ are modeled as Poisson random variables with rate parameter $\theta_2 N_t$, such that $x_t \sim \text{Poiss}(\theta_2 N_t)$. For our simulations, we fixed $\sigma_e^2 = 0.09$ and focused on estimating the parameter vector $\theta = [\theta_1, \theta_2]^\top$. The prior distribution is set as a uniform distribution $\mathcal{U}([2, 8] \times [0, 20])$. We simulated the process for $T = 100$ time steps to generate sufficient data for inference, and considered a simulation budget of 1000 to create the dataset.

**Ornstein-Uhlenbeck process (OUP)** is a stochastic differential equation model widely used in financial mathematics and evolutionary biology. The OU process $x_t$ is defined as,

$$x_{t+1} = x_t + \Delta x_t, \quad t \in \{1, \ldots, T\} \tag{14}$$
$$\Delta x_t = \theta_1 [\exp(\theta_2) - x_t] \Delta t + 0.5 w \tag{15}$$

where $T = 25$, $\Delta t = 0.2$, $x_0 = 10$, and $w \sim \mathcal{N}(0, \Delta t)$.

**Generalized Linear Model (GLM).** A 10 parameter Generalized Linear Model (GLM) with Bernoulli observations.

**Gaussian Linear Uniform (GLU).** A 10 dimensional Gaussian model, where data points are simulated as $x \sim \mathcal{N}(x \mid \theta, \Sigma)$. The parameter $\theta$ is the mean, and the covariance $\Sigma = 0.1\mathbb{I}$ is fixed, with a uniform prior ($\theta \in \mathcal{U}([-1, 1]^{10})$). We refer to Lueckmann et al. (2021); Tejero-Cantero et al. (2020) for further details on these SBI tasks.

**Hodgkin Huxley Model.** Hodgkin Huxley Model is a real-world computational neuroscience simulator. It describes the intricate dynamics of the generation and propagation of action potentials along neuronal membranes with the capture of the time course of membrane voltage by modeling the behavior of ion channels, particularly sodium and potassium, as well as leak currents. It consists of two parameters: $\theta_1 \equiv \bar{g}_{\text{Na}}$, and $\theta_2 \equiv \bar{g}_{\text{K}}$, which describe the density of Na and K specifically. The

dynamics are parameterized as a set of differential equations,

$$C_m \frac{dV}{dt} = g_1(E - V) + \theta_1 m^3 h(E_{\text{Na}} - V) + \theta_2 n^4 h(E_{\text{K}} - V) + \bar{g}_{\text{M}} p(E_{\text{M}} - V) + I_{\text{inj}} + \sigma\eta(t)$$

$$\frac{dq}{dt} = \frac{q_\infty(V) - q}{\tau_q(V)}, \quad q \in \{m, h, n, p\}$$

Here, $V$ represents the membrane potential, $C_m$ the membrane capacitance, $g_1$ is the leak conductance, $E_1$ is the membrane reverse potential, $\theta_1, \theta_2$ are the densities of Na and K channel, $\bar{g}_{\text{M}}$ is the density for M channel, $E_{\text{Na,K,M}}$ denotes the reversal potential, and $\sigma\eta(t)$ is the intrinsic neural noise. The right hand side of the voltage dynamics is composed of a leak current, a voltage-dependent $\text{Na}^+$, a delayed rectifier $\text{K}^+$, a slow voltage-dependent $\text{K}^+$ current responsible for spike-frequency adaptation, and an injected current $I_{\text{inj}}$. Channel gating variables $q$ have dynamics fully characterized by the neuron membrane potential $V$, given the respective steady-state $q_\infty(V)$ and time constant $\tau_q(V)$. For more details, see Pospischil et al. (2008).

### A.4.2 Prior distributions

We utilize the following prior distributions for our experiment tasks:

- Ricker: Uniform distribution $\mathcal{U}([2, 8] \times [0, 20])$

- OUP: Uniform prior $\mathcal{U}([0, 2] \times [-2, 2])$

- Hodgkin-Huxley: Uniform distribution $\mathcal{U}([10^{-4}, -0.5] \times [15.0, 100.0])$

- GLU: Uniform distribution $\mathcal{U}([-1, 1]^{10})$

- GLM: A multivariate normal $\mathcal{N}(0, (\mathbf{F}^\top \mathbf{F})^{-1})$ computed as follows,

$$\mathbf{F}_{i,i-2} = 1, \mathbf{F}_{i,i-1} = -2, \mathbf{F}_{i,i} = 1 + \sqrt{\frac{i-1}{9}}, \mathbf{F}_{i,j} = 0 \text{ otherwise}, 1 \leq i, j \leq 9 \quad (16)$$

### A.4.3 Creating the missingness mask

**MCAR.** We adopted random masking to simulate the MCAR scenario. For a given missingness degree $\varepsilon$, we randomly mask out $\varepsilon\%$ of the data sample.

**MNAR.** We employed the *self-masking* or *self-censoring* approach as outlined by Ipsen et al. (2020). For a given data sample $\mathbf{x} \in \mathbb{R}^d$, and following Sinelnikov et al. (2024); Ong et al. (2024), the probability of a particular data-point to be missing depends on its value. Specifically, we sample the mask $s_i$ for $i^{th}$ value for data sample as,

$$s_i \sim \text{Bern}(p_i), \quad p_i = \varepsilon \cdot \frac{x_i}{\max_d(\mathbf{x})} \quad (17)$$

where $0 \leq i \leq d$, $\max_d(\mathbf{x})$ represents the maximum value in the data sample and $p_i$ is the masking probability for data-point $x_i$ which is computed using the proportion of missing values $\varepsilon$.

### A.4.4 Network Parametrization

**Summary Networks.** For the Ricker and Huxley model, the summary network is composed of 1D convolutional layers, whereas for the OUP, it is a combination of bidirectional long short-term memory (LSTM) recurrent modules and 1D convolutional layers. The dimension of the statistic space is set to four for both the models. We do not use summary networks for GLM and GLU.

**Imputation Model.** The parameters for the neural process-based imputation model used in RISE are given in Table 3 and Table 4.

Table 3: Default hyperparameters for imputation model $\hat{p}_\varphi$ for Ricker, OUP and Huxley model.

| Module | Hyperparameter | Meaning | Value |
|---|---|---|---|
| Encoder | CNN blocks | Number of CNN layers | 1 |
| | Hidden dimension | Number of output channels of each CNN layer | 64 |
| | Kernel size | Kernel size of each convolution layer | 9 |
| | Stride | Stride of each convolution layer | 1 |
| | Padding | Padding size of each convolution layer | 4 |
| Latent | CNN blocks | Number of CNN layers | 2 |
| | Hidden dimension | Number of output channels of each CNN layer | 32 |
| | Kernel size | Kernel size of each convolution layer | 3 |
| | Stride | Stride of each convolution layer | 1 |
| | Padding | Padding size of each convolution layer | 1 |
| Decoder | CNN blocks | Number of CNN layers | [6,1] |
| | Hidden dimension | Number of output channels of each CNN layer | [32,2] |
| | Kernel size | Kernel size of each convolution layer | 5 |
| | Stride | Stride of each convolution layer | 1 |
| | Padding | Padding size of each convolution layer | 2 |

Table 4: Default hyperparameters for imputation model $\hat{p}_\varphi$ for GLM and GLU.

| Module | Hyperparameter | Meaning | Value |
|---|---|---|---|
| Encoder | MLP blocks | Number of MLP layers | [1,1] |
| | Hidden dimension | Number of output channels of each MLP layer | [32,64] |
| Latent | MLP blocks | Number of MLP layers | 2 |
| | Hidden dimension | Number of output channels of each MLP layer | 32 |
| Decoder | MLP blocks | Number of MLP layers | [6,1] |
| | Hidden dimension | Number of output channels of each MLP layer | [32,10] |

**Inference model.** Our inference model implementations are based on publicly available code from the sbi library `https://github.com/mackelab/sbi`. We use the NPE-C model (Greenberg et al., 2019) with Masked Autoregressive Flow (MAF) (Papamakarios et al., 2017) as the backbone inference network, and adopt the default configuration with 20 hidden units and 5 transforms for MAF. Throughout our experiments, we maintained a consistent batch size of 50 and a fixed learning rate of $5 \times 10^{-4}$.

## B  ADDITIONAL RESULTS

Figure 5 shows how accurate our proposed method is in imputing the values of missing data simulated from the SBI benchmark models compared to the baselines. The performance is measured in terms of RMSE of the imputed values. Our method (denoted in red) performs the best in imputing the missing values (which eventually helps in improving the estimation of the posterior distribution).

### B.1  MMD

In Table 5, we report the MMD values for the experiment on SBI benchmark simulators presented in Section 5.1. Similar to the NLPP and C2ST results of Table 1, we observe that RISE yields lowest MMD for almost all the cases, especially for Ricker and OUP where RISE beats the baselines comprehensively.

### B.2  COVERAGE PLOTS

We compute the expected coverage (Hermans et al., 2022) of our method on various confidence levels. Figure 6 shows the expected coverage for the HH task and GLU at various levels of missingness. We

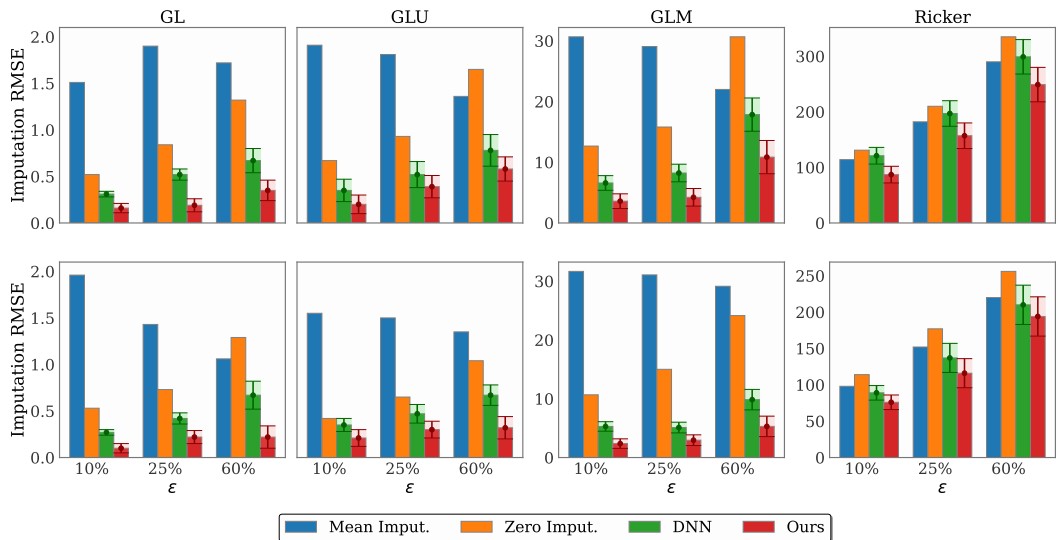

Figure 5: Imputation RMSE for MCAR (**top**) and MNAR (**bottom**) over various synthetic datasets. Here GL refers to a 10 dimension Gaussian linear model, see Lueckmann et al. (2021) for details.

Table 5: MMD under MCAR and MNAR scenarios, with missing value proportion $\varepsilon$. RISE demonstrates superior posterior estimation performance. For MNAR scenarios, the proportion of missing values averages below $\varepsilon$ due to self-censoring (details in Appendix A.4.3). Note that Simformer results are unavailable for Ricker and OUP due to the lack of official implementation.

| | Dataset | $\epsilon$ | MCAR | | | | MNAR | | | |
|---|---|---|---|---|---|---|---|---|---|---|
| | | | NPE-NN | Wang et al. | Simformer | RISE | NPE-NN | Wang et al. | Simformer | RISE |
| MMD | GLU | 10% | $0.21 \pm 0.02$ | $0.20 \pm 0.03$ | $0.20 \pm 0.01$ | $\mathbf{0.18} \pm 0.01$ | $0.25 \pm 0.03$ | $0.23 \pm 0.02$ | $0.18 \pm 0.01$ | $\mathbf{0.16} \pm 0.01$ |
| | | 25% | $0.27 \pm 0.02$ | $0.27 \pm 0.01$ | $0.27 \pm 0.01$ | $\mathbf{0.26} \pm 0.01$ | $0.29 \pm 0.02$ | $0.26 \pm 0.02$ | $0.25 \pm 0.01$ | $\mathbf{0.22} \pm 0.02$ |
| | | 60% | $0.40 \pm 0.04$ | $0.36 \pm 0.02$ | $0.39 \pm 0.03$ | $\mathbf{0.33} \pm 0.03$ | $0.33 \pm 0.02$ | $0.31 \pm 0.02$ | $0.32 \pm 0.02$ | $\mathbf{0.27} \pm 0.02$ |
| | GLM | 10% | $0.15 \pm 0.01$ | $0.15 \pm 0.02$ | $0.17 \pm 0.01$ | $\mathbf{0.12} \pm 0.01$ | $0.16 \pm 0.01$ | $0.14 \pm 0.02$ | $0.16 \pm 0.01$ | $\mathbf{0.13} \pm 0.01$ |
| | | 25% | $0.37 \pm 0.02$ | $0.30 \pm 0.02$ | $0.31 \pm 0.03$ | $\mathbf{0.27} \pm 0.03$ | $0.18 \pm 0.02$ | $0.22 \pm 0.02$ | $0.25 \pm 0.03$ | $\mathbf{0.17} \pm 0.01$ |
| | | 60% | $0.50 \pm 0.04$ | $0.44 \pm 0.02$ | $0.52 \pm 0.03$ | $\mathbf{0.38} \pm 0.05$ | $0.62 \pm 0.04$ | $0.53 \pm 0.02$ | $0.50 \pm 0.03$ | $\mathbf{0.47} \pm 0.02$ |
| | Ricker | 10% | $0.45 \pm 0.01$ | $0.38 \pm 0.02$ | - | $\mathbf{0.31} \pm 0.01$ | $0.49 \pm 0.01$ | $0.32 \pm 0.02$ | - | $\mathbf{0.27} \pm 0.02$ |
| | | 25% | $0.47 \pm 0.02$ | $0.39 \pm 0.02$ | - | $\mathbf{0.35} \pm 0.02$ | $0.49 \pm 0.02$ | $0.41 \pm 0.02$ | - | $\mathbf{0.36} \pm 0.03$ |
| | | 60% | $0.51 \pm 0.02$ | $0.43 \pm 0.02$ | - | $\mathbf{0.37} \pm 0.01$ | $0.57 \pm 0.01$ | $0.46 \pm 0.02$ | - | $\mathbf{0.41} \pm 0.05$ |
| | OUP | 10% | $0.35 \pm 0.02$ | $0.31 \pm 0.02$ | - | $\mathbf{0.29} \pm 0.02$ | $0.42 \pm 0.03$ | $0.41 \pm 0.02$ | - | $\mathbf{0.38} \pm 0.03$ |
| | | 25% | $0.36 \pm 0.02$ | $0.33 \pm 0.02$ | - | $\mathbf{0.30} \pm 0.02$ | $0.43 \pm 0.02$ | $0.41 \pm 0.02$ | - | $\mathbf{0.38} \pm 0.02$ |
| | | 60% | $0.39 \pm 0.03$ | $0.37 \pm 0.02$ | - | $\mathbf{0.35} \pm 0.02$ | $0.44 \pm 0.03$ | $0.41 \pm 0.02$ | - | $\mathbf{0.39} \pm 0.02$ |

observe that RISE is able to produce conservative posterior approximations, and is better calibrated than NPE-NN.

### B.3    ADDITIONAL VISUALIZATIONS

The fig. 7 offers further insight into the posterior bias illustrated in fig. 1, specifically from the perspective of learned statistics. Our observations indicate that statistics for augmented datasets deviate from the fully observed statistic value, consequently causing a shift in the corresponding NPE posterior away from the true parameter value.

## C    ADDITIONAL ABLATION STUDIES

**Performance as function of simulation budget.**    We conduct a study to quantify RISE's performance as a function of the simulation budget on GLU and GLM dataset. Table 7 shows C2ST and MMD for different simulation budgets for RISE, for 10% missingness level. As the budget increases, the performance improves. We also visualize the posterior obtained for different simulation budgets for Ricker and OUP in Figure 8.

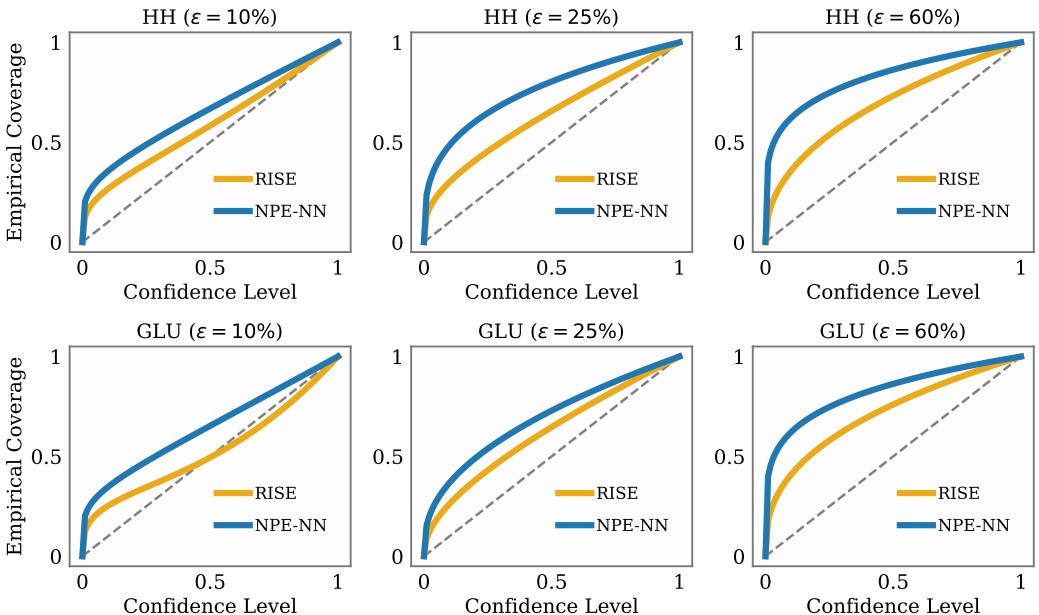

Figure 6: Expected coverage of RISE and NPE-NN for HH (top) and GLU (bottom) task over various level of missingness. The estimator becomes more conservative with increase in missingness due to the lack of information to estimate posterior and imputation distribution.

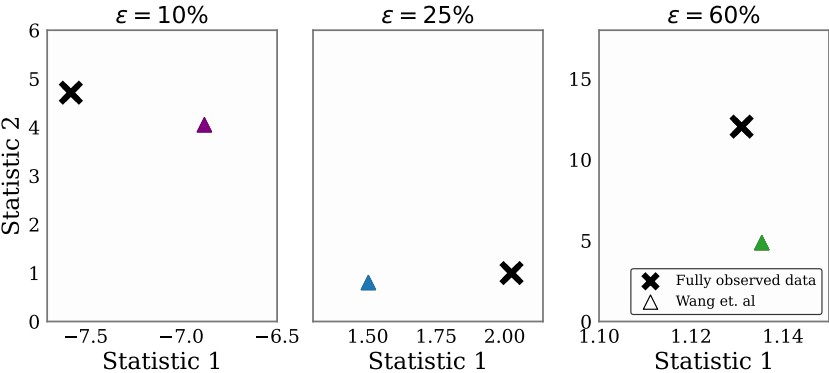

Figure 7: **Shifting Statistics.** The corresponding learned statistics for fully observed and augmented datasets as described in fig. 1. Observe that the statistics for augmented datasets shift away from the fully observed statistic value, thereby leading to a shift in the corresponding NPE posterior away from the true parameter value. Note that as we re-train the method for each missingness level, the three statistics plots should not be compared with each other.

**Runtime comparison.** We perform an ablation study to compare the computational complexity of RISE to that of standard NPE. Table 7 describes the time (in seconds) per epoch to train different models on a single V100 GPU. We observe that there is a minimal increase in runtime due to the inclusion of the imputation model. The training time remains the same with respect to missingness levels over a certain data dimensionality.

**Flow architecture.** Our final experiment involves comparing RISE's performance for different flow architectures. We utilize neural spline flow (Durkan et al., 2019) and masked autoregressive flow as competing architectures and evaluate on the GLM model under $10\%$ missingness. Table 6 shows that both NSF and MAF yield similar results.

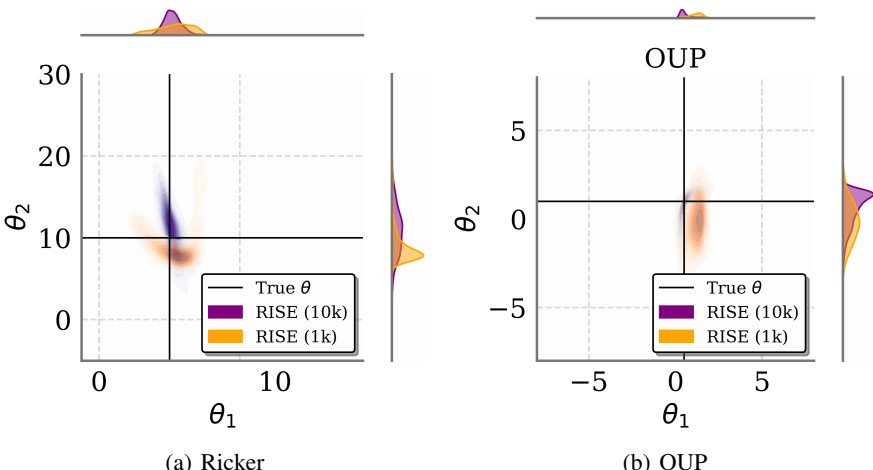

(a) Ricker  (b) OUP

Figure 8: Visualization of posterior estimated by RISE for Ricker and OUP under 1000 and 10000 simulation budget, respectively, and 25% missingness level. We observe that the posterior estimate improves with increase in the simulation budget.

Table 6: Ablation on flow architectures (*left*) and meta learning the missingness (*right*).

| Method | C2ST | RMSE | MMD |
|---|---|---|---|
| RISE-MAF | 0.80 | 0.65 | 0.12 |
| RISE-NSF | 0.80 | 0.67 | 0.11 |

| Method | Ricker | | OUP | |
|---|---|---|---|---|
| | RMSE | MMD | RMSE | MMD |
| NPE-NN | 1.97 | 0.51 | 1.32 | 0.50 |
| RISE-Meta | **1.52** | **0.42** | **0.89** | **0.45** |

Table 7: Runtime comparisons (*left*) and Simulation budget comparisons (*right*).

| Method | GLM | GLU |
|---|---|---|
| NPE | 0.12 | 0.10 |
| RISE | 0.18 | 0.16 |

| Budget | GLU | | GLM | |
|---|---|---|---|---|
| | C2ST | MMD | C2ST | MMD |
| 1000 | 0.83 | 0.18 | 0.80 | 0.12 |
| 10000 | 0.78 | 0.15 | 0.75 | 0.10 |

