# OpenReview forum: "Robust Simulation-Based Inference under Missing Data via Neural Processes"
_ICLR.cc/2025/Conference — ICLR 2025 Poster_

### Official Review · Reviewer_NU6H · 2024-10-16

**Soundness:** 2
**Presentation:** 3
**Contribution:** 2
**Rating:** 3
**Confidence:** 2

**Summary:**

The authors provide a method that aims to simultaneously learn an imputation model, alongside a posterior approximation, under an assumed missing data distribution. They fit both models by minimizing the forward KL divergence using samples generated from the assumed models joint distribution (including both missingness and the simulator component). A neural process model is used for imputation, and a masked autoregressive flow for the inference component.

**Strengths:**

This paper addresses handling missing data simulation-based inference , which is common problem in fields like astrophysics. The experiments include reasonable examples and the results seem to be good. The method is well motivated by including a probabilistic model over the missingness. Overall the paper is clear and well structured.

**Weaknesses:**

There were a few substantial issues which I feel would need to be addressed.

- From the onset, the paper should make it clear that we are interested in either the case where we have multiple observations, or expect multiple observations, so wish to maintain an amortized estimator. Otherwise, we can simply remove the missing indices from the simulated output, training the posterior estimate only on the observed sites. The examples unfortunately also focus on the single observation setting.

- The review of previous work is not sufficiently thorough. For example, the authors state "the latter problem of missing data in SBI has received little to no attention. The only exception is the work of Wang et al. (2024)". A quick search pointed me some other examples https://arxiv.org/abs/2211.03747, https://iopscience.iop.org/article/10.3847/2041-8213/ace361/meta. The summary of Wang et al. (2024) is further very poor, and perhaps should at least mention the use of training the NPE model using augmented simulations with artificially missing values, alongside a missing indicator variable. It is not clearly justified why this approach is a bad idea, especially for flexible neural network conditioning models used in normalizing flows.

- In my opinion, the notation with $x_{obs}$ is likely to confuse some readers, e.g. as it is used to refer to the subset of simulated data which is not missing in the observed data, not the observed data itself.

- The title itself I would argue is too broad, closer to describing an area of research, rather than being informative to the presented method.

- At least from the objective described in equation 5, I can see no reason to think joint training provides any benefits. The expectation is taken over a fixed distribution, and each model (inference/imputation model) has disjoint parameter sets. From my understanding you could separate the objective into two independent objectives, and expect very similar results, if not identical results, depending on the optimizer.

- No code has been provided to the reviewers.

**Questions:**

Does NPE-Mean and NPE-Zero refer to imputing only after training the NPE model? If so, I think it would be better to compare to methods  listed in Wang et al. (2024), which ensure the inference model learns to approximate the posterior using missing data during training. Otherwise, we are simply relying on neural networks to generalize to possibly out of distribution points, which is in my opinion not a fair comparison.

Why might the "joint" training procedure be useful? Have I missinterpreted the fact the objective could be partitioned into two independent objectives?

---

> ### Author Response · Authors · 2024-11-21
> **Response**
>
> Thank you very much for many excellent suggestions! We have acted on all of these, and additionally, address all your questions and comments below.
>
> > The title itself I would argue is too broad....
>
> Thank you for your suggestion, we have added 'via neural processes' in the title to be more informative about the method.
>
> > The review of previous work is not sufficiently thorough.....
>
> Many thanks for pointing to these works and apologies for missing them.  \url{https://arxiv.org/abs/2211.03747} and \url{https://iopscience.iop.org/article/10.3847/2041-8213/ace361/meta} propose to impute the missing values by sampling from a kernel density estimate (KDE) of the training data or using a nearest-neighbor search, respectively. As these approaches neglect the underlying missingness mechanisms, they can distort the relationships between variable. Moreover, they are not scalable to higher data dimensions unlike RISE. We have included this discussion in the related works section in our revised version of the paper. We have also revised the introduction correspondingly and removed our claim of the missing data problem receiving ''little to no attention".
>
> > ........ compare to methods listed in Wang et al. (2024).......
>
> **Comparison with Wang et al. (2024)** We have performed a study to compare to the method listed in Wang et al. (2024), by using data augmentation with constant values (zero) and a binary mask indicator while training the NPE on GLU and GLM data sets at various levels of missingness. The table below presents the results. We observe that RISE is able to outperform the baseline in numerous cases, even at higher level of missingness.
>
> | Missingness             | Method             | GLU           | GLU           | GLM           | GLM           |
> |-------------------------|--------------------|---------------|---------------|---------------|---------------|
> |                         |                    | RMSE          | C2ST          | MMD           | RMSE          | C2ST          | MMD           |
> | 10% | Wang et al. (2024) | 0.47          | 0.87          | 0.28          | 0.61          | 0.86          | 0.24          |
> |                         | RISE               | **0.41** | **0.83** | **0.18** | 0.65          | **0.80** | **0.12** |
> | 25% | Wang et al. (2024) | 0.45          | 0.92          | 0.31          | 0.86          | 0.94          | 0.35          |
> |                         | RISE               | **0.43** | **0.89** | **0.26** | 0.93          | **0.91** | **0.27** |
> | 60% | Wang et al. (2024) | 0.65          | 0.97          | 0.35          | 1.53          | 0.99          | 0.45          |
> |                         | RISE               | **0.56** | **0.93** | **0.33** | **1.27** | **0.97** | 0.50          |
>
> > .... we have multiple observations, or expect multiple observations, so wish to maintain an amortized estimator.....
>
> We focus on the case where there is a single vector $\mathbf{x}$ (such as a time-series) of dimension $d$ for each parameter vector, where some values in this data vector can be missing. We will explicitly state this in the beginning of Section 2. Note that our formulation can be extended straightforwardly to the multiple observations case where we obtain $\mathbf x^{(1:m)} = (\mathbf x_1, \dots, \mathbf x_m)$ for each $\theta$. Then, $\mathbf{x}^{(1:m)} = (\mathbf{x}^{(1:m)}\_{\text{obs}}, \mathbf{x}^{(1:m)}\_{\text{mis}})$, and the objective for RISE becomes
>
> $$
> \arg \min\_{\phi, \varphi, \kappa} -   \mathbb{E}\_{(\mathbf{x}^{(1:m)}\_{\text{obs}},\theta) \sim p\_{\text{true}}} \mathbb{E}_{\mathbf{x}^{(1:m)}]\_{\text{mis}} \sim \prod\_{i=1}^m p(\mathbf{x}^{(i)}\_{\text{mis}} \mid \mathbf{x}^{(i)}\_{\text{obs}})} \left[\dfrac{1}{m} \sum\_{i=1}^m\log  \underbrace{\hat{p}\_{\varphi}(\mathbf{x}^{(i)}\_{\text{mis}} \mid \mathbf{x}^{(i)}\_{\text{obs}})}\_{\textbf{(imputation)}} + \log \underbrace{q\_{\phi}(\theta \mid \eta\_\kappa (\mathbf{x}^{1:m}\_{\text{obs}}, \mathbf{x}^{1:m}\_{\text{mis}}))}\_{\textbf{(inference)}}\right].
> $$
>
> Note that here we have to summarize the data using the network $\eta_\kappa$ before passing the data into the inference network, as NPE is unable to handle multiple observations, unless we use recent extensions based on score estimation [1,2]. We have included this discussion in Appendix A.1.
>
> [1] Geffner et al. (2023). Compositional score modeling for simulation-based inference. ICML.
>
> [2] Linhart et al. (2024). Diffusion posterior sampling for simulation-based inference in tall data settings, arxiv 2404.07593.

---

> > ### Author Response · Authors · 2024-11-21
> > **Response Contd.**
> >
> > > At least from the objective described in equation 5....
> >
> > Please recall that our optimization objective can be compactly written as
> >
> > $\mathrm{argmin}_{\phi, \varphi} \,\mathbb{E}\_{\mathbf{x}\_{\text{obs}} \sim p\_{\text{true}}}  \text{KL}[p\_{\text{true}} (\theta,\mathbf{x}\_{\text{mis}} \mid \mathbf{x}\_{\text{obs}} ) \mid \mid \underbrace{r\_{\phi,\varphi}(\theta,\mathbf{x}\_{\text{mis}} \mid \mathbf{x}\_{\text{obs}})}\_{\textbf{(joint imputation and inference)}}]$
> >
> > where our goal is to learn $r_{\phi,\varphi}$ parameterized by $\psi = (\phi, \varphi$).
> >
> > In case we know $p\_{\text{true}}(\mathbf{x}\_{\text{mis}} \mid \mathbf{x}\_{\text{obs}}$) (e.g., in controlled simulation settings), the above objective can indeed be written in terms of a joint expectation over the augmented model, as you noted. However, here we also address typical real-world scenarios, where we do not have access to $p\_{\text{true}}(\mathbf{x}\_{\text{mis}} \mid \mathbf{x}\_{\text{obs}})$. Since samples for $\mathbf{x}_{\text{mis}}$ are required during training, one needs to resort to methods such as variational approximation, or expectation maximization (EM).
> >
> > Here, we aim to jointly model the distribution via a probabilistic imputation model, treating $\mathbf{x}\_{\text{mis}}$ as latents, in a setting akin to how variational autoencoders (VAEs) leverage an encoder-decoder style architecture for variational approximation. Specifically, the imputation model needs to estimate these latents for the inference network to map them to the output space.  Both networks are highly coupled, as the distribution induced by the imputation model (or our encoder) shapes the input of the inference network (i.e., our decoder), and training them independently can lead to a distributional misalignment.
> >
> > We formalize this explicitly to make the exposition clear based on your comments.
> >  We begin by noting that the above objective is equivalent to
> >
> > $$
> > \mathrm{argmax}\_{\psi} \, \mathbb{E}\_{\mathbf{x}\_{\text{obs}} \sim p\_{\text{true}}} \, \mathbb{E}\_{p\_{\text{true}}(\theta,\mathbf{x}\_{\text{mis}} \mid \mathbf{x}\_{\text{obs}} )} [\log\, r\_{\psi}(\theta,\mathbf{x}\_{\text{mis}} \mid \mathbf{x}\_{\text{obs}})] \\
> > = \mathrm{argmax}\_{\psi} \, \mathbb{E}\_{(\mathbf{x}\_{\text{obs}}, \theta, \mathbf{x}\_{\text{mis}}) \sim p\_{\text{true}}} \, [\log\, r\_{\psi}(\theta,\mathbf{x}\_{\text{mis}} \mid \mathbf{x}\_{\text{obs}})]$$
> >
> > Typically, in realistic scenarios,  we do not know $p\_{\text{true}}(\mathbf{x}\_{\text{mis}})$, but can only assume access to samples $(\mathbf{x}\_{\text{obs}}, \theta) \sim p_{\text{true}}$. Therefore, we treat $\mathbf{x}\_{\text{mis}}$ as latent variables sampled from some tractable (e.g., Gaussian) distribution $p(\cdot|\mathbf{x}\_{\text{obs}})$  conditioned on $\mathbf{x}\_{\text{obs}}$, and resort to a variational approximation.
> >
> > Specifically, we maximize a lower bound on
> > $\mathbb{E}\_{(\mathbf{x}\_{\text{obs}}, \theta) \sim p\_{\text{true}}} \, [\log\, r\_{\psi}(\theta \mid \mathbf{x}\_{\text{obs}})]$. Formally, we have
> >
> > \begin{align}
> > \mathbb{E}\_{(\mathbf{x}\_{\text{obs}},\theta) \sim p\_{\text{true}}} \log r\_{\psi}(\theta \mid \mathbf{x}\_{\text{obs}}) = \mathbb{E}\_{(\mathbf{x}\_{\text{obs}},\theta) \sim p\_{\text{true}}} \log \int r\_{\psi}(\theta,\mathbf{x}\_{\text{mis}} \mid \mathbf{x}\_{\text{obs}})d\mathbf{x}\_{\text{mis}} \\\\
> > = \mathbb{E}\_{(\mathbf{x}\_{\text{obs}},\theta) \sim p\_{\text{true}}} \log \int \frac{p(\mathbf{x}\_{\text{mis}} \mid \mathbf{x}\_{\text{obs}})r\_{\psi}(\theta,\mathbf{x}\_{\text{mis}} \mid \mathbf{x}\_{\text{obs}})}{p(\mathbf{x}\_{\text{mis}} \mid \mathbf{x}\_{\text{obs}})} d\mathbf{x}\_{\text{mis}} \\\\
> > \geq \mathbb{E}\_{(\mathbf{x}\_{\text{obs}},\theta) \sim p\_{\text{true}}} \mathbb{E}\_{\mathbf{x}\_{\text{mis}} \sim p(\mathbf{x}\_{\text{mis}} \mid \mathbf{x}\_{\text{obs}})} \left[\log  \frac{r\_{\psi}(\theta,\mathbf{x}\_{\text{mis}} \mid \mathbf{x}\_{\text{obs}})}{p(\mathbf{x}\_{\text{mis}} \mid \mathbf{x}\_{\text{obs}})} \right] \\\\
> > = \mathbb{E}\_{(\mathbf{x}\_{\text{obs}},\theta) \sim p\_{\text{true}}} \mathbb{E}\_{\mathbf{x}\_{\text{mis}} \sim p(\mathbf{x}\_{\text{mis}} \mid \mathbf{x}\_{\text{obs}})} \left[\log  \frac{r\_{\psi}(\mathbf{x}\_{\text{mis}} \mid \mathbf{x}\_{\text{obs}}) r\_{\psi}(\theta \mid \mathbf{x}\_{\text{obs}}, \mathbf{x}\_{\text{mis}})}{p(\mathbf{x}\_{\text{mis}} \mid \mathbf{x}\_{\text{obs}})} \right]
> > \end{align}
> >
> > where we invoked the Jensen's inequality to swap the log and the conditional expectation. Splitting parameters $\psi$ into imputation parameters $\varphi$ and inference parameters $\phi$, and denoting the corresponding imputation and inference networks by $\hat{p}\_{\varphi}$ and $q\_{\phi}$ respectively, we immediately get.... (continued in next response)

---

> ### Author Response · Authors · 2024-11-21
> **Response Contd.**
>
> \begin{align}
> \mathbb{E}\_{(\mathbf{x}\_{\text{obs}},\theta) \sim p\_{\text{true}}} \log r\_{\phi, \varphi}(\theta \mid \mathbf{x}\_{\text{obs}}) \geq  \mathbb{E}\_{(\mathbf{x}\_{\text{obs}},\theta) \sim p\_{\text{true}}} \mathbb{E}\_{\mathbf{x}\_{\text{mis}} \sim p(\mathbf{x}\_{\text{mis}} \mid \mathbf{x}\_{\text{obs}})} \left[\log  \frac{\hat{p}\_{\varphi}(\mathbf{x}\_{\text{mis}} \mid \mathbf{x}\_{\text{obs}}) q\_{\phi}(\theta \mid \mathbf{x}\_{\text{obs}}, \mathbf{x}\_{\text{mis}})}{p(\mathbf{x}\_{\text{mis}} \mid \mathbf{x}\_{\text{obs}})} \right]
> \end{align}
>
> Thus, we obtain the following variational objective:
>
> \begin{align}
>  \mathrm{argmax}\_{\phi, \varphi}   \mathbb{E}\_{(\mathbf{x}\_{\text{obs}},\theta) \sim p\_{\text{true}}} \mathbb{E}\_{\mathbf{x}\_{\text{mis}} \sim p(\mathbf{x}\_{\text{mis}} \mid \mathbf{x}\_{\text{obs}})} \left[\log  \frac{\hat{p}\_{\varphi}(\mathbf{x}\_{\text{mis}} \mid \mathbf{x}\_{\text{obs}}) q_{\phi}(\theta \mid \mathbf{x}\_{\text{obs}}, \mathbf{x}\_{\text{mis}})}{p(\mathbf{x}\_{\text{mis}} \mid \mathbf{x}\_{\text{obs}})} \right]
> \end{align}
>
> \begin{align}
> = \mathrm{argmax}\_{\phi, \varphi}  \mathbb{E}\_{(\mathbf{x}\_{\text{obs}},\theta) \sim p\_{\text{true}}} \left(\mathbb{E}\_{\mathbf{x}\_{\text{mis}} \sim p(\mathbf{x}\_{\text{mis}} \mid \mathbf{x}\_{\text{obs}})} \left[\log  \hat{p}\_{\varphi}(\mathbf{x}\_{\text{mis}} \mid \mathbf{x}\_{\text{obs}}) + \log q\_{\phi}(\theta \mid \mathbf{x}\_{\text{obs}}, \mathbf{x}\_{\text{mis}})\right] + \mathbb{H}(p(\mathbf{x}\_{\text{mis}} \mid \mathbf{x}\_{\text{obs}})\right)
> \end{align}
>
> \begin{align}
>  =  \mathrm{argmax}\_{\phi, \varphi} \mathbb{E}\_{(\mathbf{x}\_{\text{obs}},\theta) \sim p\_{\text{true}}} \mathbb{E}\_{\mathbf{x}\_{\text{mis}} \sim p(\mathbf{x}\_{\text{mis}} \mid \mathbf{x}\_{\text{obs}})} \left[\log  \underbrace{\hat{p}\_{\varphi}(\mathbf{x}\_{\text{mis}} \mid \mathbf{x}\_{\text{obs}})}\_{\text{imputation}} + \log \underbrace{q\_{\phi}(\theta \mid \mathbf{x}\_{\text{obs}}, \mathbf{x}\_{\text{mis}})}\_{\text{inference}}\right]~,
> \end{align}
>
> since the entropy term $\mathbb{H}(p(\mathbf{x}\_{\text{mis}} \mid \mathbf{x}\_{\text{obs}})$ does not depend on the optimization variables $\phi$ and $\varphi$.
>
> This is exactly the objective that we optimize for. Based on your comment, we have now revised section 3.2, Proposition 2, and it's proof in Appendix A.2.2 in the paper to make this clear.
>
>
> > No code has been provided to the reviewers
>
> We are in the process of cleaning and streamlining the code for much easier readability, usability, and integration into current packages. We will release the code latest by Monday next week. Apologies for the delay.
>
> We are grateful for your thoughtful feedback and suggestions. Acting on your feedback has helped appropriately position and reinforce the strengths of RISE.

---

> > ### Comment · Area_Chair_egPw · 2024-11-26
> >
> > Dear reviewer,
> >
> > Please make to sure to read, at least acknowledge, and possibly further discuss the authors' responses to your comments. Update or maintain your score as you see fit.
> >
> > The AC.

---

> > ### Comment · Reviewer_NU6H · 2024-11-26
> >
> > The authors claim "NPE is unable to handle multiple observations", which is not true. Using exchangable/deep set networks https://arxiv.org/abs/1802.06153, https://arxiv.org/abs/1703.06114 provide a simple way to handle multiple observations which share a parameter (or amortized inference can be used if the parameters are "local").
> >
> > I have two main criticisms:
> > 1) In the single observation setting, the best approach will likely be to be just to remove the missing indices. The authors do not consider this, despite being in the single observation setting.
> > 2) The descriptions of the methods is far from being precise enough. Presumably the zero/mean imputation methods involve training using the full simulations, and then imputing for the observed data when performing inference. I strongly believe that this should not be used as a benchmark if this is the case. This is comparing to a method that no reasonable person would use as it risks relying on neural networks to be able to generalize to out of distribution points.
> >
> > In the updated paper there are comments like
> >
> > > which attempts to handle missing data by augmenting and imputing
> > constant values (e.g., zero or sample mean) and training NPE with a binary mask indicator, but
> > this approach can lead to biased estimates, reduced variability, and distorted relationships between
> > variables (Graham et al., 2007). This is exemplified in Figure 1
> >
> > > ... training NPE with a binary mask indicator, but
> > this approach can lead to biased posterior estimates, as we saw in Figure 1 and Section 3.1.
> >
> > Does figure 1 use the binary indicator variable? Do the reported results use a binary indicator variable? I cannot find any evidence of this being the case in the code. In fact I do not see where in the code the baselines methods are ran. The code itself is a mess, I understand this is not a software paper, but I would have concerns about reproducibility.
> >
> > The figure itself seems to imply it possibly does not. When we get to the results, again it seems the missingness indicator variable may not be included. It is well known we should not rely a neural network to generalize to out of distiribution points, and as such I feel without training with an indicator variable and sampling missingness, it is a useless "baseline".

---

> > > ### Author Response · Authors · 2024-11-27
> > > **Response**
> > >
> > > We are grateful to the reviewer for your engagement, and thoughtful comments. We address your concerns below.
> > >
> > > > The authors claim "NPE is unable to handle multiple observations", which is not true. Using exchangable/deep set networks https://arxiv.org/abs/1802.06153, https://arxiv.org/abs/1703.06114 provide a simple way to handle multiple observations which share a parameter (or amortized inference can be used if the parameters are "local"
> > >
> > > **Regarding NPE handling multiple observations:** Thank you for the opportunity to clarify this. Here, while referring to NPE, we meant the classical NPE setting that employs only the inference network.  Certainly, as you pointed out, the NPE framework is rather flexible and can be augmented with a summary network (such as a Deep Set) to handle multiple observations. In fact, we adopt exactly this perspective for Ricker and OUP tasks, similar to Huang et al. 2023, to accommodate multiple observations.
> > >
> > > Based on your feedback, we have now expanded Section 2 (where we introduce NPE) to elucidate the multiple observations scenario in the context of NPE. We have also replaced lines 826-829 with the following:
> > >
> > > _Note that we can summarize the data using the network_ $\eta_\kappa$ _(for instance, a deep set (Zaheer et al. 2017) before passing the data into the inference network, which is standard practice when using NPE with multiple observations (Chan et al., 2018). Alternatively, one could use recent extensions based on score estimation (Geffner et al. 2023, Linhart et al. 2024) as well._
> > >
> > > > The descriptions of the methods is far from being precise enough. Presumably the zero/mean imputation methods involve training using the full simulations, and then imputing for the observed data when performing inference. I strongly believe that this should not be used as a benchmark if this is the case. This is comparing to a method that no reasonable person would use as it risks relying on neural networks to be able to generalize to out of distribution points. Does figure 1 use the binary indicator variable? Do the reported results use a binary indicator variable? I cannot find any evidence of this being the case in the code.
> > >
> > > **Update on Figure 1:** Thank you for your incisive remark, and apologies for an oversight on our part. We previously forgot to replace, in Figure 1, NPE-zero and NPE-mean with the approach by Wang et al. (2024). Specifically, as you suggested, we had already implemented this method that tracks the missing indices during training via a binary masking indicator.
> > >
> > > We have updated Fig. 1, and it now uses the binary mask indicator variable as described by Wang et. al 2024 and augmentation of the missing data with constant (zero) values. Figure 1(a) demonstrates the shift in the posterior distribution as a function of missingness level when NPE is trained via Wang et. al approach. Figure 1(b) shows the corresponding learned statistics for fully observed and augmented datasets consisting of the binary mask indicator. We observe that the statistics for augmented datasets shift away from the fully observed statistic value, thereby leading to a shift in the corresponding NPE posterior away from the true parameter value. These observations are consistent with what we had before, though the shift in posterior was more pronounced for the earlier case. Note that as we re-train the method for each missingness level, the three statistics plots should not be compared with each other.

---

> ### Author Response · Authors · 2024-11-27
> **Response Contd.**
>
> > In the single observation setting, the best approach will likely be to be just to remove the missing indices. The authors do not consider this, despite being in the single observation setting.
>
> **Approach in single observation setting:** Good point. In fact, the method SIMFORMER included here as a baseline - based on suggestions from reviewers La5K, 3tVu, and XWnt (please see Table 6 and Table 7 on page 22 of the manuscript) -  effectively accomplishes what you suggest via masking. We include the results below for convenience.
>
> ####  MCAR:
> | Missingness             | Method    | GLU                       | GLU                      | GLU                      | GLM                       | GLM                      | GLM                       |
> |-------------------------|-----------|---------------------------|--------------------------|--------------------------|---------------------------|--------------------------|---------------------------|
> |                         |           | NLPP                      | MMD                      | C2ST                     | NLPP                      | MMD                      | C2ST                      |
> | 10%                     | Simformer | -2.45 $\pm$ 0.12          | 0.20 $\pm$ 0.01          | 0.86 $\pm$ 0.01          | -6.47 $\pm$ 0.16          | 0.17 $\pm$ 0.02          | 0.84 $\pm$ 0.01           |
> |  10%                        | RISE      | **-2.3** $\pm$ 0.10 | **0.18** $\pm$ 0.01 | **0.83** $\pm$ 0.02 | **-6.32** $\pm$ 0.15 | **0.12** $\pm$ 0.02 | **0.80** $\pm$ 0.02 |
> | 25% | Simformer | **-3.65** $\pm$ 0.17 | 0.27  $\pm$ 0.02         | 0.90 $\pm$ 0.01          | -7.37 $\pm$ 0.13          | 0.31 $\pm$ 0.03          | 0.92 $\pm$ 0.01           |
> |    25%                     | RISE      | -3.71 $\pm$ 0.11          | **0.26** $\pm$ 0.01 | **0.89**$\pm$ 0.02  | **-7.22** $\pm$ 0.17 | **0.27** $\pm$ 0.01 | **0.91** $\pm$ 0.01  |
> | 60% | Simformer | -6.62 $\pm$ 0.27          | 0.39 $\pm$ 0.03          | 0.95 $\pm$ 0.01          | -8.93 $\pm$ 0.18          | 0.52 $\pm$ 0.03          | 0.98 $\pm$ 0.01           |
> |    60%                     | RISE      | **-6.21** $\pm$ 0.11 | **0.33** $\pm$ 0.02 | **0.93** $\pm$ 0.01 | **-8.71** $\pm$ 0.14 | **0.50** $\pm$ 0.01 | **0.96** $\pm$ 0.01  |
>
> ####  MNAR:
>
> | Missingness             | Method    | GLU                       | GLU                      | GLU                      | GLM                       | GLM                      | GLM                      |
> |-------------------------|-----------|---------------------------|--------------------------|--------------------------|---------------------------|--------------------------|--------------------------|
> |                         |           | NLPP                      | MMD                      | C2ST                     | NLPP                      | MMD                      | C2ST                     |
> | 10% | Simformer | -2.15 $\pm$ 0.10          | 0.18 $\pm$ 0.01          | 0.84 $\pm$ 0.02          | -6.17 $\pm$ 0.18          | 0.16 $\pm$ 0.02          | 0.83 $\pm$ 0.01          |
> |    10%                     | RISE      | **-1.90** $\pm$ 0.09 | **0.16** $\pm$ 0.01 | **0.80** $\pm$ 0.01 | **-5.82** $\pm$ 0.11 | **0.13** $\pm$ 0.02 | **0.79** $\pm$ 0.02 |
> | 25% | Simformer | **-3.12** $\pm$ 0.12 | 0.25  $\pm$ 0.02         | 0.89 $\pm$ 0.01          | -6.57 $\pm$ 0.14          | 0.25 $\pm$ 0.03          | 0.90 $\pm$ 0.01          |
> |    25%                     | RISE      | -3.26 $\pm$ 0.10          | **0.22** $\pm$ 0.01 | **0.86** $\pm$ 0.02 | **-6.12** $\pm$ 0.15 | **0.17** $\pm$ 0.01 | **0.88** $\pm$ 0.01 |
> | 60% | Simformer | -6.02 $\pm$ 0.12          | 0.32 $\pm$ 0.03          | 0.92 $\pm$ 0.01          | -7.56 $\pm$ 0.15          | 0.50 $\pm$ 0.03          | 0.95 $\pm$ 0.01          |
> |     60%                     | RISE      | **-5.80** $\pm$ 0.13 | **0.27** $\pm$ 0.04 | **0.90** $\pm$ 0.01 | **-7.11** $\pm$ 0.17 | **0.47** $\pm$ 0.03 | **0.92** $\pm$ 0.01 |

---

> > ### Author Response · Authors · 2024-11-27
> > **Response Contd.**
> >
> > > The figure itself seems to imply it possibly does not. When we get to the results, again it seems the missingness indicator variable may not be included. It is well known we should not rely a neural network to generalize to out of distribution points, and as such I feel without training with an indicator variable and sampling missingness, it is a useless "baseline".
> >
> > **Comparison with Wang et al. 2024** Agreed. In addition to comparisons with Simformer provided above, please find below empirical evidence on GLU and GLM datasets at various levels of missingness that RISE compares favorably with the method from Wang. et al 2024, which trains NPE with a binary mask indicator variable.
> >
> >
> > | Missingness             | Method             | GLU           | GLU    | GLU       | GLM           | GLM           |  GLM |
> > |-----|--------------------|---------------|---------------|---------------|---------------| --------| --------|
> > |                       |                    | RMSE          | C2ST          | MMD           | RMSE          | C2ST          | MMD           |
> > | 10% | Wang et al. (2024) | 0.47          | 0.87          | 0.28          | **0.61**          | 0.86          | 0.24          |
> > |                         | RISE               | **0.41** | **0.83** | **0.18** | 0.65          | **0.80** | **0.12** |
> > | 25% | Wang et al. (2024) | 0.45          | 0.92          | 0.31          | **0.86**          | 0.94          | 0.35          |
> > |                         | RISE               | **0.43** | **0.89** | **0.26** | 0.93          | **0.91** | **0.27** |
> > | 60% | Wang et al. (2024) | 0.65          | 0.97          | 0.35          | 1.53          | 0.99          | **0.45**          |
> > |                         | RISE               | **0.56** | **0.93** | **0.33** | **1.27** | **0.97** | 0.50          |
> >
> > > In fact I do not see where in the code the baselines methods are ran. The code itself is a mess, I understand this is not a software paper, but I would have concerns about reproducibility.
> >
> > **Details about the code:**  We apologize for not providing sufficient details earlier about the  code. We strongly believe in transparency and reproducibility, and will work towards enhancing the readability of the code upon acceptance.  Here we provide all the key details to  dispel any concerns in this regard.
> >
> > We have built our code using the sbi library (https://github.com/mackelab/sbi) and on top of the robust-sbi (https://github.com/huangdaolang/robust-sbi) code of Huang et al. 2023.
> >
> > We are working towards making our code more user-readable and including comments. Consistent with the best established practice, in order to ensure we do not induce any errors of our own, we used the official implementations for all the baseline, strictly following the respective instructions from the official repositories (including, the recommended hyperparameter settings). Specifically, these official implementations are publicly available for
> >
> > - Simformer at (https://github.com/mackelab/simformer)
> >
> > - NPE-NN (Lueckmann et al 2017) at (https://github.com/mackelab/delfi), and
> >
> > - Wang et al. at (https://journals.plos.org/ploscompbiol/article?id=10.1371/journal.pcbi.1012184); in particular, please see the lines 104-107 in ```train_glm_wang.py/glu_wang.py``` for how they use a concatenation operation to include the binary mask.
> >
> > We have now described the baselines in the 'readme' file of the attached code repository (snapshot below), along with the commands to run RISE. Please let us know if anything is still unclear.
> >
> > ````
> > # Robust Simulation-based Inference under missing data (RISE): Code implementation
> >
> > The code is built on the top of the following repo. Please follow the guidelines in the reference repository to install requisite packages.
> >
> > Ref: [Learning Robust Statistics for Simulation-Based Inference Under Model Misspecification](https://github.com/huangdaolang/robust-sbi)
> >
> > Note: Please change the device (GPU or CPU) in the files accordingly depending on which you are running.
> >
> > ## Baselines
> >
> > We utilised the official code implementation of [Simformer](https://github.com/mackelab/simformer) to run the baseline.
> > For NPE-NN (Lueckmann et. al), we followed code implementation  [here](https://github.com/mackelab/delfi).
> > For Wang et. al 2024, we train the NPE with augmentation and the binary mask indicator (as described in [official paper](https://journals.plos.org/ploscompbiol/article?id=10.1371/journal.pcbi.1012184)), and can be run by
> >
> > ```
> > python -u train_glm_wang/_glu_wang.py --degree degree --type mcar
> >
> > ```
> > ## Running RISE
> >
> > To run the task on GLU and GLM dataset for mcar/mnar under a certain degree run,
> >
> > ```
> > python -u train_glm/glu.py --degree degree --type mcar/mnar
> >
> > ```
> > ## License
> > This code is released under the MIT License.
> >
> > ````

---

> > > ### Author Response · Authors · 2024-11-27
> > > **Response Contd**
> > >
> > > We greatly appreciate your time and effort in engaging with us. Your closer scrutiny of this work has helped us significantly in improving the quality of this work, with respect to both theoretical exposition as well as empirical substantiation. We hope your concerns are addressed. We would love to engage further if you have any further questions, concerns, or suggestions.

---

> > > > ### Author Response · Authors · 2024-12-02
> > > > **Gentle Reminder**
> > > >
> > > > Dear Reviewer NU6H
> > > >
> > > > We are grateful for your thoughtful, insightful feedback and closer scrutiny regarding our work. As the discussion phase for authors and reviewers concludes, we hope we have addressed all of your concerns. Based on your comments, we have incorporated additional comparisons with Simformer, Wang et al 2024 which keep track of missing variables via binary indicator mask, and clarifications regarding requirement for joint training of our method.
> > > >
> > > > We believe these updates sufficiently address your feedback and would greatly appreciate your stronger support for this paper.
> > > > We would love to engage further if you have any further questions, concerns, or suggestions.

---

### Official Review · Reviewer_XWnt · 2024-10-25

**Soundness:** 3
**Presentation:** 3
**Contribution:** 3
**Rating:** 6
**Confidence:** 4

**Summary:**

The paper proposes an approach to deal with missing data in the simulation-based inference (SBI) setting. In SBI, parameters are to be inferred for a model with an intractable likelihood function. The authors propose an imputation approach utilising latent neural processes, in which parameters of the neural process and the posterior estimator are jointly optimised. The resulting method (RISE) is compared against alternative approaches on a number of problems showing good performance in terms of MMD and RSME.

**Strengths:**

- The paper addresses are timely, relevant topic.
- Motivation and approach are clearly laid out.
- The evaluation spans statistical problems and real world data sets.
- RISE outperforms baselines on problems considered.

**Weaknesses:**

- Some comparisons are missing, see questions for details.
- HH example would benefit from quantitative evaluation.
- Previous literature could be cited more accurately, some examples:
  - L46: See also Lueckmann et al. (2017) where NN-based imputation for SBI was used.
  - L50: Citing Radev et al. (2022) for NPE seems out of place, consider crediting NPE to Papamakarios and Murray (2016) and subsequent work based on it.
  - L206: The VAE paper was published in 2013, not 2022.

**Questions:**

- Have you considered running simulation-based calibration to check posteriors after imputation? This would allow going beyond the qualitative analysis on the HH example.
- How does NPE-NN perform on the HH example?
- What are meta-learning results for the remaining statistical problems (Ricker, OUP)?
- How does RISE compare against the method by Gloeckner et al. for MCAR cases?
- Are posterior distributions for all statistical examples unimodal?

---

> ### Author Response · Authors · 2024-11-21
> **Response**
>
> Thank you very much for excellent suggestions! We have acted on all of these, and additionally, address all your questions and comments below.
>
> > Some comparisons are missing, calibration plots, HH evaluations.
>
> **Calibration plots for RISE**: Good point. Based on your suggestion, we have added expected coverage plots to evaluate the calibration of the posterior estimator for HH and GLU for RISE and NPE-NN over various missingness levels in Appendix B.7. We observe that RISE is better calibrated than NPE-NN for all missingness levels.
>
> > Previous literature could be cited more accurately,
>
> Thank you for pointing this out. We have included these changes in the revised version.
>
> > What are meta-learning results for the remaining statistical problems (Ricker, OUP)?
>
> **Meta learning results for Ricker and OUP**. Here are the meta-learning results for Ricker and OUP, which we have included in Appendix B.4 of the paper.
>
> | Method    | Ricker        | Ricker        | OUP           | OUP           |
> |-----------|---------------|---------------|---------------|---------------|
> |           | RMSE          | MMD           | RMSE          | MMD           |
> | NPE-Zero  | 3.98          | 0.95          | 3.10          | 0.57          |
> | NPE-Mean  | 2.31          | 0.67          | 1.73          | 0.53          |
> | NPE-NN    | 1.97          | 0.51          | 1.32          | 0.50          |
> | RISE-Meta | **1.52** | **0.42** | **0.89** | **0.45** |
>
> > How does RISE compare against the method by Gloeckner et al. for MCAR cases?
>
> **Comparison with Simformer**. Based on your suggestion, we have compared RISE's performance against Simformer for the GLU and GLM task. The results (in terms of MMD and nominal log posterior probability (NLPP)) for both the MCAR and the MNAR (also in Appendix B.2 of the revised paper) are shown below.
>
> | Missingness             | Method    | GLU                       | GLU                      | GLM                       | GLM                      |
> |-------------------------|-----------|---------------------------|--------------------------|---------------------------|--------------------------|
> |                         |           | NLPP                      | MMD                      | NLPP                      | MMD                      |
> | 10 %                  | Simformer | -2.45 $\pm$ 0.12          | 0.20 $\pm$ 0.01          | -6.47 $\pm$ 0.16          | 0.17 $\pm$ 0.02          |
> |                         | RISE      | **-2.31** $\pm$ 0.10 | **0.18** $\pm$ 0.01 | **-6.32** $\pm$ 0.15 | **0.12** $\pm$ 0.02 |
> | 25% | Simformer | **-3.65** $\pm$ 0.17 | 0.27  $\pm$ 0.02         | -7.37 $\pm$ 0.13          | 0.31 $\pm$ 0.03          |
> |                         | RISE      | -3.71 $\pm$ 0.11          | **0.26** $\pm$ 0.01 | **-7.22** $\pm$ 0.17 | **0.27** $\pm$ 0.01 |
> | 60 % | Simformer | -6.62 $\pm$ 0.27          | 0.39 $\pm$ 0.03          | -8.93 $\pm$ 0.18          | 0.52 $\pm$ 0.03          |
> |                         | RISE      | **-6.21** $\pm$ 0.11 | **0.33** $\pm$ 0.02 | **-8.71** $\pm$ 0.14 | **0.50** $\pm$ 0.01 |
>
> | Missingness | Method    | GLU                       | GLU                      | GLM                       | GLM                      |
> |-------------|-----------|---------------------------|--------------------------|---------------------------|--------------------------|
> |             |           | NLPP                      | MMD                      | NLPP                      | MMD                      |
> | 10%      | Simformer | -2.15 $\pm$ 0.10          | 0.18 $\pm$ 0.01          | -6.17 $\pm$ 0.18          | 0.16 $\pm$ 0.02          |
> |             | RISE      | **-1.90** $\pm$ 0.09 | **0.16** $\pm$ 0.01 | **-5.82** $\pm$ 0.11 | **0.13** $\pm$ 0.02 |
> | 25%      | Simformer | **-3.12** $\pm$ 0.12 | 0.25  $\pm$ 0.02         | -6.57 $\pm$ 0.14          | 0.25 $\pm$ 0.03          |
> |             | RISE      | -3.26 $\pm$ 0.10          | **0.22** $\pm$ 0.01 | **-6.12** $\pm$ 0.15 | **0.17** $\pm$ 0.01 |
> | 60%      | Simformer | -6.02 $\pm$ 0.12          | 0.32 $\pm$ 0.03          | -7.56 $\pm$ 0.15          | 0.50 $\pm$ 0.03          |
> |             | RISE      | **-5.80** $\pm$ 0.13 | **0.27** $\pm$ 0.04 | **-7.11** $\pm$ 0.17 | **0.47** $\pm$ 0.03 |
>
> For the MCAR (top) case, both RISE and Simformer perform similarly, with RISE yielding slightly better results. The difference in performance is more stark in the MNAR (bottom) case, as expected, since Simformer does not explicitly model the missingness mechanism.
>
> > Are posterior distributions for all statistical examples unimodal?
>
> We utilize normalizing flows to model the posterior distribution, which can model any complex distribution. In our setting, we observe only unimodal posterior distributions.
>
> We are grateful for your thoughtful comments and suggestions. We hope our response addresses your concerns and reinforces your support for this work.

---

> > ### Comment · Reviewer_XWnt · 2024-11-26
> >
> > Thank you for your answers. I think the revisions improved the manuscript and I increased my score to reflect this.
> >
> > Just a quick clarification: I asked whether the posterior distributions were unimodal because MMD might be misleading for multimodal distributions, depending on how it is used exactly. Since methods are only benchmarked on unimodal posteriors this should not be an issue.

---

> > > ### Author Response · Authors · 2024-11-26
> > >
> > > Yes, that's a good point about MMD. We will make a note of that in the paper when we introduce the performance metrics.
> > >
> > > Thank you for increasing the score, and for helping us improve the paper. Much appreciated.

---

### Official Review · Reviewer_3tVu · 2024-10-27

**Soundness:** 3
**Presentation:** 3
**Contribution:** 2
**Rating:** 6
**Confidence:** 4

**Summary:**

The paper explores the challenge of SBI under conditions of missing data and introduces a novel method named RISE. By integrating Neural Posterior Estimation (NPE) with neural processes, RISE effectively tackles the problem of missing data. Experimental results show that RISE offers a more robust posterior estimation than other imputation methods across varying levels of missingness.

**Strengths:**

This work presents the RISE method, which innovatively combines data imputation and neural posterior estimation to address the issue of missing data in SBI. This combination is novel within the SBI domain. The authors have validated the effectiveness of the method using multiple benchmark models and real-world datasets under various experimental settings. The paper is well-organized and the figures are easy for readers to understand.

**Weaknesses:**

1. Reporting of Benchmark Metrics:

The authors report only MMD and RMSE as performance metrics. However, additional metrics commonly used in SBI literature or benchmarks [1] would provide a more comprehensive evaluation. For example, metrics like the negative log probability of true parameters (to assess the density of true parameters $\theta$ within the approximate posterior), median distances (to evaluate the distance between samples generated from $\theta$ under the approximate posterior and observations), and Classifier 2-Sample Tests (C2ST, to measure the closeness between approximate and true posteriors) are not included in this paper. C2ST is sensitive to subtle differences between distributions in high-dimensional spaces, potentially revealing nuances that MMD may miss due to kernel dependence [2].

To enhance the robustness and comprehensiveness of the evaluation, I recommend the authors include C2ST as an additional metric.

2. Comparisons to NPE-Zero and NPE-Mean:

The RISE method is compared with NPE-Zero and NPE-Mean methods, which fill missing inputs using zeros or sample means (Line 48, 323). These two imputation strategies are overly simplistic and introduce significant bias (as seen in Figure 1.(c)), resulting in biased posterior distributions. For a more rigorous comparison, it may be helpful to add a baseline such as Gloeckler et al. (2024) under the MCAR cases (Line 289).

3. Choice of Conditional Density Estimator:

For the conditional density estimator, the authors do not use the Neural Spline Flows (NSFs) structure [3] employed in the SBI library. The NSFs is more performant and flexible than MAFs and has been widely used for SBI tasks in recent works [1, 4, 5, 6]. It would be helpful if the authors provided a rationale for selecting MAFs over NSFs or included an ablation study comparing RISE performance using MAFs and NSFs.

---
[1]. Lueckmann, et al. "Benchmarking simulation-based inference." 2021.

[2]. Bischoff, et al. "A practical guide to statistical distances for evaluating generative models in science." 2024.

[3]. Durkan, et al. "Neural spline flows." 2019.

[4]. Deistler, et al. "Truncated proposals for scalable and hassle-free simulation-based inference." 2022.

[5]. Kelly, et al. "Misspecification-robust sequential neural likelihood." 2023.

[6]. Ward, et al. "Robust neural posterior estimation and statistical model criticism." 2022.

**Questions:**

1. The RISE loss function (Eq. (5)) includes two parts: $\hat{p_\varphi} $ to generate $x_{mis}$ given $x_{obs}$, and $q_\phi$ to generate $\theta$ given $x_{obs}$ and $x_{mis}$. Can these two parts be trained separately? Alternatively, is it possible to train a flow directly from $x_{obs}$ or $(x_{obs}, c_{obs})$ to $\theta$?

2. (For Weaknesses 2): In Line 289, the authors mention that the method proposed by Gloeckler et al. (2024) is unequipped to handle MAR and MNAR cases. Why not compare this method with RISE under MCAR cases?

3. (For Weaknesses 2): Could the authors include additional illustrations to show how the summary statistics produced by their data imputation method deviate from the true statistics, similar to what is shown in Figure 1.(c)? I believe this would help clarify why the RISE method reduces posterior bias.

---

> ### Author Response · Authors · 2024-11-21
> **Response**
>
> Many thanks for your constructive review. We address your concerns and incorporate all your suggestions below.
>
> > Reporting of Benchmark Metrics:
>
> **C2ST scores and additional metrics.** Thank you for the suggestion. We have now added the C2ST scores and the nominal log posterior probability (NLPP) metric for the benchmark tasks. The results are shown in the table below. The conclusion remains the same, with RISE outperforming the baselines in all but one case (in terms of C2ST).
>
> | Missingness             | Method   | GLU           | GLU                       | GLM           | GLM                       | Ricker        | Ricker                    | OUP           | OUP                       |
> |-------------------------|----------|---------------|---------------------------|---------------|---------------------------|---------------|---------------------------|---------------|---------------------------|
> |                         |          | C2ST          | NLPP                      | C2ST          | NLPP                      | C2ST          | NLPP                      | C2ST          | NLPP                      |
> | 10% | NPE-Zero | 0.89          | -2.77 $\pm$ 0.13          | 0.87          | -6.92 $\pm$ 0.14          | 0.95          | -5.15 $\pm$ 0.12          | 0.90          | -2.61 $\pm$ 0.16          |
> |                         | NPE-Mean | 0.88          | -2.67 $\pm$ 0.16          | 0.85          | -6.83  $\pm$ 0.14         | 0.95          | -5.10  $\pm$ 0.21         | 0.90          | -2.51 $\pm$ 0.11          |
> |                         | NPE-NN   | 0.87          | -2.51 $\pm$ 0.11          | 0.84          | -6.57$\pm$ 0.13           | 0.94          | -4.90  $\pm$ 0.16         | 0.89          | -2.25  $\pm$ 0.18         |
> |                         | RISE     | **0.83** | **-2.31** $\pm$ 0.10 | **0.80** | **-6.32** $\pm$ 0.15 | 0.90          | **-4.20** $\pm$ 0.09 | **0.87** | **-2.09** $\pm$ 0.11 |
> | 25% | NPE-Zero | **0.88** | -4.11 $\pm$ 0.17          | 0.97          | -8.05 $\pm$ 0.20          | 0.96          | -5.10 $\pm$ 0.16          | 0.92          | -2.97 $\pm$ 0.13          |
> |                         | NPE-Mean | 0.90          | -3.99 $\pm$ 0.21          | 0.94          | -7.92  $\pm$ 0.14         | 0.96          | -5.05 $\pm$ 0.11          | 0.92          | -2.84 $\pm$ 0.15          |
> |                         | NPE-NN   | 0.91          | -3.92 $\pm$ 0.11          | 0.93          | -7.72  $\pm$ 0.16         | 0.95          | -4.94$\pm$ 0.17           | 0.90          | -2.74 $\pm$ 0.18          |
> |                         | RISE     | 0.89          | **-3.71** $\pm$ 0.11 | **0.91** | **-7.22** $\pm$ 0.17 | **0.92** | **-4.64** $\pm$ 0.15 | **0.89** | **-2.43** $\pm$ 0.15 |
> | 60% | NPE-Zero | 0.97          | -6.98 $\pm$ 0.18          | 1.00          | -9.63 $\pm$ 0.14          | 0.97          | -5.17  $\pm$ 0.15         | 0.96          | -3.07 $\pm$0.12           |
> |                         | NPE-Mean | 0.98          | -6.76 $\pm$ 0.09          | 0.99          | -9.27  $\pm$ 0.14         | 0.97          | -5.10 $\pm$ 0.18          | 0.95          | -2.97 $\pm$ 0.12          |
> |                         | NPE-NN   | 0.96          | -6.37  $\pm$ 0.12         | 0.99          | -9.02 $\pm$ 0.17          | 0.96          | -4.97 $\pm$ 0.17          | 0.95          | -2.87 $\pm$ 0.19          |
> |                         | RISE     | **0.93** | **-6.21** $\pm$ 0.11 | **0.97** | **-8.71** $\pm$ 0.14 | **0.94** | **-4.72** $\pm$ 0.17 | **0.93** | **-2.52** $\pm$ 0.11 |
>
> > For the conditional density estimator, the authors do not use the Neural Spline Flows (NSFs)......
>
> **Ablation with Flow architecture**. To ensure a fair comparison, we have followed the same hyper-parameters and training setup as described in [1]. Based on your suggestion, we have performed an ablation study on GLM dataset with $10\%$ missigness level and comparisons are below for RISE. We observe that both NSF and MAF give similar results.
>
> | Method | C2ST | RMSE | MMD|
> |----|----|----|----|
> |RISE-MAF|0.80|0.65|0.12|
> |RISE-NSF|0.80|0.67|0.11|
>
>
> > Could the authors include additional illustrations.....
>
> **Additional Visualization**. Based on your suggestion, we have added a visualisation in Appendix B.8 to show the deviation in summary statistics produced by our method. Indeed, RISE yields statistics that are closer to the true statistics value (when there is no missing values) than the constant imputation baselines.

---

> ### Author Response · Authors · 2024-11-21
> **Response Contd.**
>
> > For a more rigorous comparison, it may be helpful to add a baseline such as Gloeckler et al. (2024) under the MCAR cases....
>
> **Benchmarking with Simformer**. We have compared RISE’s performance against Simformer for the GLU and GLM task. The results (in terms of MMD and nominal log posterior probability (NLPP)) for both the MCAR and the MNAR (also in Appendix B.2 of the revised paper). For the MCAR case, both RISE and Simformer perform similarly, with RISE yielding slightly better results.
> The difference in performance is more stark in the MNAR case, as expected, since Simformer does not
> explicitly model the missingness mechanism.
>
> ### MCAR:
> | Missingness             | Method    | GLU                       | GLU                      | GLM                       | GLM                      |
> |-------------------------|-----------|---------------------------|--------------------------|---------------------------|--------------------------|
> |                         |           | NLPP                      | MMD                      | NLPP                      | MMD                      |
> | 10%                  | Simformer | -2.45 $\pm$ 0.12          | 0.20 $\pm$ 0.01          | -6.47 $\pm$ 0.16          | 0.17 $\pm$ 0.02          |
> |                         | RISE      | **-2.31** $\pm$ 0.10 | **0.18** $\pm$ 0.01 | **-6.32** $\pm$ 0.15 | **0.12** $\pm$ 0.02 |
> | 25% | Simformer | **-3.65** $\pm$ 0.17 | 0.27  $\pm$ 0.02         | -7.37 $\pm$ 0.13          | 0.31 $\pm$ 0.03          |
> |                         | RISE      | -3.71 $\pm$ 0.11          | **0.26** $\pm$ 0.01 | **-7.22** $\pm$ 0.17 | **0.27** $\pm$ 0.01 |
> | 60% | Simformer | -6.62 $\pm$ 0.27          | 0.39 $\pm$ 0.03          | -8.93 $\pm$ 0.18          | 0.52 $\pm$ 0.03          |
> |                         | RISE      | **-6.21** $\pm$ 0.11 | **0.33** $\pm$ 0.02 | **-8.71** $\pm$ 0.14 | **0.50** $\pm$ 0.01 |
>
> ### MNAR:
> | Missingness | Method    | GLU                       | GLU                      | GLM                       | GLM                      |
> |-------------|-----------|---------------------------|--------------------------|---------------------------|--------------------------|
> |             |           | NLPP                      | MMD                      | NLPP                      | MMD                      |
> | 10%      | Simformer | -2.15 $\pm$ 0.10          | 0.18 $\pm$ 0.01          | -6.17 $\pm$ 0.18          | 0.16 $\pm$ 0.02          |
> |             | RISE      | **-1.90** $\pm$ 0.09 | **0.16** $\pm$ 0.01 | **-5.82** $\pm$ 0.11 | **0.13** $\pm$ 0.02 |
> | 25%      | Simformer | **-3.12** $\pm$ 0.12 | 0.25  $\pm$ 0.02         | -6.57 $\pm$ 0.14          | 0.25 $\pm$ 0.03          |
> |             | RISE      | -3.26 $\pm$ 0.10          | **0.22** $\pm$ 0.01 | **-6.12** $\pm$ 0.15 | **0.17** $\pm$ 0.01 |
> | 60%      | Simformer | -6.02 $\pm$ 0.12          | 0.32 $\pm$ 0.03          | -7.56 $\pm$ 0.15          | 0.50 $\pm$ 0.03          |
> |             | RISE      | **-5.80** $\pm$ 0.13 | **0.27** $\pm$ 0.04 | **-7.11** $\pm$ 0.17 | **0.47** $\pm$ 0.03 |
>
>
> > The RISE loss function (Eq. (5)) includes two parts....
>
> Please recall that our optimization objective can be compactly written as
>
> $\mathrm{argmin}_{\phi, \varphi} \,\mathbb{E}\_{\mathbf{x}\_{\text{obs}} \sim p\_{\text{true}}}  \text{KL}[p\_{\text{true}} (\theta,\mathbf{x}\_{\text{mis}} \mid \mathbf{x}\_{\text{obs}} ) \mid \mid \underbrace{r\_{\phi,\varphi}(\theta,\mathbf{x}\_{\text{mis}} \mid \mathbf{x}\_{\text{obs}})}\_{\textbf{(joint imputation and inference)}}]$
>
> where our goal is to learn $r_{\phi,\varphi}$ parameterized by $\psi = (\phi, \varphi$).
>
> In case we know $p\_{\text{true}}(\mathbf{x}\_{\text{mis}} \mid \mathbf{x}\_{\text{obs}}$) (e.g., in controlled simulation settings), the above objective can indeed be written in terms of a joint expectation over the augmented model, as you noted. However, here we also address typical real-world scenarios, where we do not have access to $p\_{\text{true}}(\mathbf{x}\_{\text{mis}} \mid \mathbf{x}\_{\text{obs}})$. Since samples for $\mathbf{x}_{\text{mis}}$ are required during training, one needs to resort to methods such as variational approximation, or expectation maximization (EM).
>
> Here, we aim to jointly model the distribution via a probabilistic imputation model, treating $\mathbf{x}\_{\text{mis}}$ as latents, in a setting akin to how variational autoencoders (VAEs) leverage an encoder-decoder style architecture for variational approximation. Specifically, the imputation model needs to estimate these latents for the inference network to map them to the output space.  Both networks are highly coupled, as the distribution induced by the imputation model (or our encoder) shapes the input of the inference network (i.e., our decoder), and training them independently can lead to a distributional misalignment.

---

> > ### Author Response · Authors · 2024-11-21
> > **Response Contd.**
> >
> > We formalize this explicitly to make the exposition clear based on your comments.
> >  We begin by noting that the above objective is equivalent to
> >
> > $$
> > \mathrm{argmax}\_{\psi} \, \mathbb{E}\_{\mathbf{x}\_{\text{obs}} \sim p\_{\text{true}}} \, \mathbb{E}\_{p\_{\text{true}}(\theta,\mathbf{x}\_{\text{mis}} \mid \mathbf{x}\_{\text{obs}} )} [\log\, r\_{\psi}(\theta,\mathbf{x}\_{\text{mis}} \mid \mathbf{x}\_{\text{obs}})] \\
> > = \mathrm{argmax}\_{\psi} \, \mathbb{E}\_{(\mathbf{x}\_{\text{obs}}, \theta, \mathbf{x}\_{\text{mis}}) \sim p\_{\text{true}}} \, [\log\, r\_{\psi}(\theta,\mathbf{x}\_{\text{mis}} \mid \mathbf{x}\_{\text{obs}})]$$
> >
> > Typically, in realistic scenarios,  we do not know $p\_{\text{true}}(\mathbf{x}\_{\text{mis}})$, but can only assume access to samples $(\mathbf{x}\_{\text{obs}}, \theta) \sim p_{\text{true}}$. Therefore, we treat $\mathbf{x}\_{\text{mis}}$ as latent variables sampled from some tractable (e.g., Gaussian) distribution $p(\cdot|\mathbf{x}\_{\text{obs}})$  conditioned on $\mathbf{x}\_{\text{obs}}$, and resort to a variational approximation.
> >
> > Specifically, we maximize a lower bound on
> > $\mathbb{E}\_{(\mathbf{x}\_{\text{obs}}, \theta) \sim p\_{\text{true}}} \, [\log\, r\_{\psi}(\theta \mid \mathbf{x}\_{\text{obs}})]$. Formally, we have
> >
> > \begin{align}
> > \mathbb{E}\_{(\mathbf{x}\_{\text{obs}},\theta) \sim p\_{\text{true}}} \log r\_{\psi}(\theta \mid \mathbf{x}\_{\text{obs}}) = \mathbb{E}\_{(\mathbf{x}\_{\text{obs}},\theta) \sim p\_{\text{true}}} \log \int r\_{\psi}(\theta,\mathbf{x}\_{\text{mis}} \mid \mathbf{x}\_{\text{obs}})d\mathbf{x}\_{\text{mis}} \\\\
> > = \mathbb{E}\_{(\mathbf{x}\_{\text{obs}},\theta) \sim p\_{\text{true}}} \log \int \frac{p(\mathbf{x}\_{\text{mis}} \mid \mathbf{x}\_{\text{obs}})r\_{\psi}(\theta,\mathbf{x}\_{\text{mis}} \mid \mathbf{x}\_{\text{obs}})}{p(\mathbf{x}\_{\text{mis}} \mid \mathbf{x}\_{\text{obs}})} d\mathbf{x}\_{\text{mis}} \\\\
> > \geq \mathbb{E}\_{(\mathbf{x}\_{\text{obs}},\theta) \sim p\_{\text{true}}} \mathbb{E}\_{\mathbf{x}\_{\text{mis}} \sim p(\mathbf{x}\_{\text{mis}} \mid \mathbf{x}\_{\text{obs}})} \left[\log  \frac{r\_{\psi}(\theta,\mathbf{x}\_{\text{mis}} \mid \mathbf{x}\_{\text{obs}})}{p(\mathbf{x}\_{\text{mis}} \mid \mathbf{x}\_{\text{obs}})} \right] \\\\
> > = \mathbb{E}\_{(\mathbf{x}\_{\text{obs}},\theta) \sim p\_{\text{true}}} \mathbb{E}\_{\mathbf{x}\_{\text{mis}} \sim p(\mathbf{x}\_{\text{mis}} \mid \mathbf{x}\_{\text{obs}})} \left[\log  \frac{r\_{\psi}(\mathbf{x}\_{\text{mis}} \mid \mathbf{x}\_{\text{obs}}) r\_{\psi}(\theta \mid \mathbf{x}\_{\text{obs}}, \mathbf{x}\_{\text{mis}})}{p(\mathbf{x}\_{\text{mis}} \mid \mathbf{x}\_{\text{obs}})} \right]
> > \end{align}
> >
> > where we invoked the Jensen's inequality to swap the log and the conditional expectation. Splitting parameters $\psi$ into imputation parameters $\varphi$ and inference parameters $\phi$, and denoting the corresponding imputation and inference networks by $\hat{p}\_{\varphi}$ and $q\_{\phi}$ respectively, we immediately get
> >
> > \begin{align}
> > \mathbb{E}\_{(\mathbf{x}\_{\text{obs}},\theta) \sim p\_{\text{true}}} \log r\_{\phi, \varphi}(\theta \mid \mathbf{x}\_{\text{obs}}) \geq  \mathbb{E}\_{(\mathbf{x}\_{\text{obs}},\theta) \sim p\_{\text{true}}} \mathbb{E}\_{\mathbf{x}\_{\text{mis}} \sim p(\mathbf{x}\_{\text{mis}} \mid \mathbf{x}\_{\text{obs}})} \left[\log  \frac{\hat{p}\_{\varphi}(\mathbf{x}\_{\text{mis}} \mid \mathbf{x}\_{\text{obs}}) q\_{\phi}(\theta \mid \mathbf{x}\_{\text{obs}}, \mathbf{x}\_{\text{mis}})}{p(\mathbf{x}\_{\text{mis}} \mid \mathbf{x}\_{\text{obs}})} \right]
> > \end{align}
> >
> > Thus, we obtain the following variational objective:
> >
> > \begin{align}
> >  \mathrm{argmax}\_{\phi, \varphi}   \mathbb{E}\_{(\mathbf{x}\_{\text{obs}},\theta) \sim p\_{\text{true}}} \mathbb{E}\_{\mathbf{x}\_{\text{mis}} \sim p(\mathbf{x}\_{\text{mis}} \mid \mathbf{x}\_{\text{obs}})} \left[\log  \frac{\hat{p}\_{\varphi}(\mathbf{x}\_{\text{mis}} \mid \mathbf{x}\_{\text{obs}}) q_{\phi}(\theta \mid \mathbf{x}\_{\text{obs}}, \mathbf{x}\_{\text{mis}})}{p(\mathbf{x}\_{\text{mis}} \mid \mathbf{x}\_{\text{obs}})} \right]
> > \end{align}
> >
> > \begin{align}
> > = \mathrm{argmax}\_{\phi, \varphi}  \mathbb{E}\_{(\mathbf{x}\_{\text{obs}},\theta) \sim p\_{\text{true}}} \left(\mathbb{E}\_{\mathbf{x}\_{\text{mis}} \sim p(\mathbf{x}\_{\text{mis}} \mid \mathbf{x}\_{\text{obs}})} \left[\log  \hat{p}\_{\varphi}(\mathbf{x}\_{\text{mis}} \mid \mathbf{x}\_{\text{obs}}) + \log q\_{\phi}(\theta \mid \mathbf{x}\_{\text{obs}}, \mathbf{x}\_{\text{mis}})\right] + \mathbb{H}(p(\mathbf{x}\_{\text{mis}} \mid \mathbf{x}\_{\text{obs}})\right)
> > \end{align}

---

> > > ### Author Response · Authors · 2024-11-21
> > > **Response Contd.**
> > >
> > > \begin{align}
> > >  =  \mathrm{argmax}\_{\phi, \varphi} \mathbb{E}\_{(\mathbf{x}\_{\text{obs}},\theta) \sim p\_{\text{true}}} \mathbb{E}\_{\mathbf{x}\_{\text{mis}} \sim p(\mathbf{x}\_{\text{mis}} \mid \mathbf{x}\_{\text{obs}})} \left[\log  \underbrace{\hat{p}\_{\varphi}(\mathbf{x}\_{\text{mis}} \mid \mathbf{x}\_{\text{obs}})}\_{\text{imputation}} + \log \underbrace{q\_{\phi}(\theta \mid \mathbf{x}\_{\text{obs}}, \mathbf{x}\_{\text{mis}})}\_{\text{inference}}\right]~,
> > > \end{align}
> > >
> > > since the entropy term $\mathbb{H}(p(\mathbf{x}\_{\text{mis}} \mid \mathbf{x}\_{\text{obs}})$ does not depend on the optimization variables $\phi$ and $\varphi$.
> > >
> > > This is exactly the objective that we optimize for. Based on your comment, we have now revised section 3.2, Proposition 2, and it's proof in Appendix A.2.2 in the paper to make this clear.
> > >
> > > We are grateful for your thoughtful review. We hope our response has addressed your questions and concerns.

---

> > > > ### Comment · Area_Chair_egPw · 2024-11-26
> > > >
> > > > Dear reviewer,
> > > >
> > > > Please make to sure to read, at least acknowledge, and possibly further discuss the authors' responses to your comments. Update or maintain your score as you see fit.
> > > >
> > > > The AC.

---

> > > > > ### Author Response · Authors · 2024-12-02
> > > > > **Gentle Reminder**
> > > > >
> > > > > Dear Reviewer 3tVu,
> > > > >
> > > > > We are grateful for your thoughtful and insightful feedback regarding our work. As the discussion phase for authors and reviewers concludes, we hope we have addressed all of your concerns. Based on your suggestions, we have incorporated additional comparisons with Simformer, included C2ST scores as benchmarking metrics, and conducted an ablation study on the flow architecture.
> > > > >
> > > > > We believe these modifications sufficiently address your concerns and would greatly appreciate your stronger support for this paper. We would love to engage further if you have any further questions, concerns, or suggestions.

---

> > > > ### Comment · Reviewer_3tVu · 2024-12-02
> > > > **Thanks for the response**
> > > >
> > > > After reviewing the authors' responses and considering the clarifications they provided, I have updated my score to reflect the improved understanding of their work.

---

### Official Review · Reviewer_riJc · 2024-10-31

**Soundness:** 3
**Presentation:** 2
**Contribution:** 3
**Rating:** 6
**Confidence:** 4

**Summary:**

This paper addresses the problem of missing data in simulation-based inference (SBI). The authors formalize how missing data can introduce bias into SBI posterior estimates when naive imputation methods are used. They propose RISE (Robust Inference under imputed SimulatEd data) to tackle this issue. This novel method jointly learns an imputation model and an inference network within a neural posterior estimation (NPE) framework. The imputation model is based on neural processes, allowing RISE to handle different missingness mechanisms (MCAR, MAR, MNAR). The method is amortized and can be generalized across varying levels of missingness. Extensive experiments on SBI benchmarks demonstrate that RISE outperforms baseline methods in inference and imputation tasks.

**Strengths:**

- **Contribution**: The paper tackles a significant and underexplored problem in SBI—handling missing data—and provides a novel solution by jointly learning imputation and inference models.

- **Theoretical foundation**: The authors provide a formal analysis showing how naive imputation leads to biased posterior estimates, strengthening the motivation for their method. The cases for MNAR, MAR, and MCAR have been nicely formalized. The simplification of $\mathcal{L}\_{RISE}$ by $\mathcal{L}\_{NPE}$ in Proposition 2 is particularly convenient.

- **Amortization**: The method can generalize across varying levels of missing data (RISE-Meta), making it practical for real-world applications.

**Weaknesses:**

- **Clarity of presentation**: Some parts, particularly the mathematical formulations and explanations of the method, could be clearer. For instance, in Section 1 (Introduction), the authors refer to Figure 1, which has the axes $\theta_{1}$ and $\theta_{2}$ but only in Section 2 (Preliminaries) define what $\theta$ is.

- **Literature for handling missing values is limited**: While the literature review mentions prominent methods for handling missing values from the deep learning literature, such as GAIN [1], more traditional and frequently used techniques have been excluded. For example, the authors could have compared their methods or, at the very least, acknowledged imputation techniques such as expectation-maximization (EM) found even in the deep learning literature [2] and other traditional approaches such as MICE [3].

- **Computational efficiency**: The paper does not provide a detailed analysis of the computational cost of RISE. This is especially true because the paper uses normalizing flows, which can be computationally expensive. Given the added complexity of jointly learning the imputation model and the inference network, it would be useful to understand the trade-offs.

- **Some important limitations, such as the Gaussian assumption, have not been highlighted**: Even though the authors mention in line 255 that the Gaussian assumption does not limit the expressivity, this is still a limitation that has to be clearly highlighted. The argument for infinite mixtures may be valid theoretically but computationally infeasible. This is especially true as the authors mention in line 476 that the credibility of the posterior estimates needs to be further examined, which is directly impacted by the normality assumption.

- **Source code for reproduction**: It would have been good to have the source code available for a more careful examination and reproduction of the results.

### References:

- [1] https://proceedings.mlr.press/v80/yoon18a
- [2] https://proceedings.neurips.cc/paper/2018/hash/411ae1bf081d1674ca6091f8c59a266f-Abstract.html
- [3] https://www.jstatsoft.org/article/view/v045i03

**Questions:**

- **Confusion in the contributions**: As mentioned above, in line 84, the most important contribution of the paper is that the proposed method is 'robust' to shift in the posterior distributions, but at the same time, the authors mention that the credibility of the posterior estimate after imputation remains an area for exploration. This seems contradictory. Can the authors clarify this point?

- **Discussion of possible alternatives**: Why not use the EM algorithm?

- **Extension to multiple observations**: I didn't quite understand if RISE can be extended to handle multiple observed data points per parameter setting. If so, what modifications would be necessary?

- **Small remark on the structure of Section 5**: I initially missed the referred datasets. Is it because you once showed the SBI benchmarks and the other datasets (Adrenergic and Kinase)? Section 5 was structured confusingly; I recommend restructuring it or adding a small paragraph before *performance metrics*  to more clearly explain the section structure for the datasets.

- **Sensitivity to neural process architecture**: How sensitive is the performance of RISE to the choice of neural process architecture and hyperparameters? Have the authors conducted ablation studies on this aspect?

- **Computational efficiency**: Can the authors provide insights into the computational complexity of RISE compared to baseline methods? How does training time scale with data dimensionality and missingness levels?

- **Handling model misspecification**: While the paper assumes a well-specified simulator, how would RISE perform under model misspecification? Can the method be adapted to account for this?

---

> ### Author Response · Authors · 2024-11-21
> **Response**
>
> Thanks so much for your thoughtful comments and excellent suggestions! We’ve acted on all of them,
> and also address all your concerns, as we describe below.
>
> >  Clarity of presentation:
>
> Thank you for pointing this out. We will replace the notation with textual labels in Figure 1.
>
> > Literature for handling missing values is limited...
>
> Thank you for providing these references. We have added them in our related works section. The link provided for reference [2] seems to be broken. Could you please provide the paper's title so we can add it to the related works, along with MICE [3]?
>
> > Computational efficiency....
>
> **Computational Complexity**. Thank you for your suggestion. We have performed an ablation study to evaluate the computational complexity of our method in comparison to standard NPE. The table below shows the time (in seconds) per epoch to train different models on a single V100 GPU. We observe that there is roughly 50\% increase in runtime per epoch due to the addition of the imputation model.
>
>
> | Batch Size| Method | GLU | GLM|
> |------|------|------|-------|
> | 50| NPE| 0.12| 0.10|
> | 50| RISE| 0.18| 0.16|
> | 100| NPE|0.20 |0.21 |
> | 100| RISE| 0.31| 0.29|
>
>
> > Some important limitations, such as the Gaussian assumption, have not been highlighted:
>
> Agreed. We have highlighted this in the conclusion section.
>
> > Confusion in the contributions
>
> We understand the confusion. What we meant by ``credibility of the posterior'' is that neural network-based SBI methods (like NPE) have been found to yield overconfident posteriors [1] (thus questioning their credibility). So even though RISE reduces the bias in posterior under missing data, it inherits the issues of NPE and does not guarantee well-calibrated posteriors (which is an active area of research [2]). We have revised the conclusion to make this point clear.
>
> [1] Hermans et al. (2022). A Crisis In Simulation-Based Inference? Beware, Your Posterior Approximations Can Be Unfaithful. TMLR.
>
> [2] Falkiewicz et al. (2023). Calibrating Neural Simulation-Based Inference with Differentiable Coverage Probability. NeurIPS.
>
> > Discussion of possible alternatives
>
> Indeed, one could use an EM algorithm to learn the imputation model instead of a VAE like approach. However, EM struggles with high-dimensional data, whereas our method is scalable w.r.t high dimensionality of data. Moreover, RISE can perform end-to-end learning rather than iterating between the expectation and maximization steps of EM, and can model complex distributions as compared to EM.
>
> > Extension to multiple observations:
>
> RISE can be straightforwardly extended to handle multiple observations where we obtain $\mathbf x^{(1:m)} = (\mathbf x_1, \dots, \mathbf{x}\_m)$ for each $\theta$. Then, $\mathbf{x}^{(1:m)} = (\mathbf{x}^{(1:m)}\_{\text{obs}}, \mathbf{x}^{(1:m)}\_{\text{mis}})$, and the objective for RISE becomes
> $$
> \arg \min\_{\phi, \varphi, \kappa} -   \mathbb{E}\_{(\mathbf{x}^{(1:m)}\_{\text{obs}},\theta) \sim p\_{\text{true}}} \mathbb{E}_{\mathbf{x}^{(1:m)}]\_{\text{mis}} \sim \prod\_{i=1}^m p(\mathbf{x}^{(i)}\_{\text{mis}} \mid \mathbf{x}^{(i)}\_{\text{obs}})} \left[\dfrac{1}{m} \sum\_{i=1}^m\log  \underbrace{\hat{p}\_{\varphi}(\mathbf{x}^{(i)}\_{\text{mis}} \mid \mathbf{x}^{(i)}\_{\text{obs}})}\_{\textbf{(imputation)}} + \log \underbrace{q\_{\phi}(\theta \mid \eta\_\kappa (\mathbf{x}^{1:m}\_{\text{obs}}, \mathbf{x}^{1:m}\_{\text{mis}}))}\_{\textbf{(inference)}}\right].
> $$
>
> Note that here we have to summarize the data using the network $\eta\_\kappa$ before passing the data into the inference network, as NPE is unable to handle multiple observations, unless we use recent extensions based on score estimation [1,2].
>
> [1] Geffner et al. (2023). Compositional score modeling for simulation-based inference. ICML.
>
> [2] Linhart et al. (2024). Diffusion posterior sampling for simulation-based inference in tall data settings, arxiv 2404.07593.
>
> > Small remark on the structure of Section 5
>
> The Adrenergic and Kinase datasets are not SBI benchmarks, and are only used for the ablation study in Section 5.4 to test the imputation performance of RISE. That is why we did not introduce it earlier, as we focused on posterior estimation results. We have added the following sentences before the performance metrics, as suggested:
>
> ``This section is arranged as follows: We first provide results on SBI benchmarks in Sections 5.1, 5.2 and 5.3. In Section 5.4, we perform ablation studies to evaluate the imputation performance of RISE on real-world bio-activity datasets.''

---

> > ### Author Response · Authors · 2024-11-21
> > **Response Contd.**
> >
> > > Sensitivity to neural process architecture
> >
> > **Ablation regarding NP architecture** Based on your suggestion, we have performed an ablation study regarding the architecture of neural processes and hyperparameters. We varied the hidden dimension size of MLPs parameterizing the neural process and evaluated on GLM dataset with missingness level 10%. The results are shown below, and we observe that both of the parameter settings give similar results.
> >
> > | Hidden dim size| C2ST | RMSE | MMD|
> > |-------|-------|------|----|
> > |64|0.80|0.65|0.12|
> > |128|0.78|0.68|0.10|
> >
> > > Handling model misspecification
> >
> > **Handling model misspecification** : Good question. We conjecture that replacing the inference network in RISE from the usual NPE to its robust variant such as the method of Ward et al. (2022) or Huang et al. (2023) would help in addressing model misspecification issues. It would be an interesting avenue for future research to see how to train these robust NPE methods jointly with the imputation network of RISE, and how effective such an approach is. One way is to assume a certain error model over the observed data $\mathbf{x}$ and corrupt the data $\tilde{\mathbf{x}}$ via adding Gaussian noise and infer the correct $\theta$ via the inference network. This can also be described via the objective as
> >
> > \begin{align}
> >     \mathrm{argmin}\_{\phi,\varphi} - \mathbb{E}\_{
> >  (\mathbf{x}\_{\text{obs}}, \theta) \sim p(\mathbf{x}\_{\text{obs}}, \theta), \tilde{\mathbf{x}}\_{\text{obs}} \sim \mathcal{N}(\mathbf{x}\_{\text{obs}},\sigma^2),  {\mathbf{\tilde{x}}}\_{\text{mis}} \sim p\_{\text{true}}( \mathbf{\tilde{x}}\_{\text{mis}} \mid \mathbf{\tilde{x}}\_{\text{obs}}, \theta )} \left[ \log \hat{p}\_\varphi(\mathbf{\tilde{x}}\_{\text{mis}} \mid \mathbf{\tilde{x}}\_{\text{obs}}) + \log q\_\phi(\theta \mid \mathbf{\tilde{x}}\_{\text{obs}},{\mathbf{\tilde{x}}}\_{\text{mis}})  \right]
> > \end{align}
> >
> > Moreover, this can also be readily extended to  incorporate prior miss-specification via similar way as,
> > \begin{align}
> >     \mathrm{argmin}\_{\phi,\varphi} - \mathbb{E}\_{
> >  (\mathbf{x}\_{\text{obs}}, \theta) \sim p(\mathbf{x}\_{\text{obs}}, \theta), \tilde{\theta} \sim \mathcal{N}(\theta,\sigma^2),  {\mathbf{\tilde{x}}}\_{\text{mis}} \sim p\_{\text{true}}( \mathbf{\tilde{x}}\_{\text{mis}} \mid \mathbf{\tilde{x}}\_{\text{obs}}, \tilde{\theta} )} \left[ \log \hat{p}\_\varphi(\mathbf{\tilde{x}}\_{\text{mis}} \mid \mathbf{\tilde{x}}\_{\text{obs}}) + \log q\_\phi(\theta \mid \mathbf{\tilde{x}}\_{\text{obs}},{\mathbf{\tilde{x}}}\_{\text{mis}}) \right]
> > \end{align}
> >
> > We have included this in the Discussion in Appendix A.1.
> >
> > > Source Code
> >
> > We are in the process of cleaning and streamlining the code for much easier readability, usability, and integration into current packages. We will release the code latest by Monday next week. Apologies for the delay.
> >
> > We are grateful for your thoughtful feedback and suggestions. Acting on your feedback has helped appropriately position and reinforce the strengths of RISE.

---

> > > ### Comment · Reviewer_riJc · 2024-11-26
> > >
> > > I thank the authors for their responses and clarifications. The link to reference [2] is not broken (please try clicking on it again), but here is the citation in case _"Śmieja, M., Struski, u., Tabor, J., Zieliński, B., & Spurek, P. (2018). Processing of missing data by neural networks. In Advances in Neural Information Processing Systems."_. With the authors' clarifications, I have increased my confidence score.

---

> > > > ### Author Response · Authors · 2024-11-26
> > > >
> > > > Thank you. Much appreciated.
> > > >
> > > > We have included this reference in the related work. As promised, we have also uploaded the code as supplementary material. Thanks again for engaging with us.

---

### Official Review · Reviewer_jx4r · 2024-11-02

**Soundness:** 3
**Presentation:** 3
**Contribution:** 2
**Rating:** 8
**Confidence:** 4

**Summary:**

The paper emphasizes the problem of missing data in simulation-based inference (SBI) and proposes to learn a latent variable model of the predictive imputation distribution . This problem has a long-standing tradition in statistics and probabilistic modeling and extends way beyond SBI. However, it has received relatively little attention in the SBI literature, one reason being that simple data augmentation approaches are sufficient in most cases. The main value of the paper lies in the generality of the proposed method, which is clearly justified and well motivated.

**Strengths:**

- The paper is clearly written and easy to follow; it can be appreciated by both readers familiar with SBI and newcomers to the field. It addresses an important practical problem and is of interest to the community.

- The joint training of a posterior surrogate and latent variable missing data model is original, quite general, and useful for SBI applications where the simulator is cheap to run.

- The evaluation features a variety of different models (even though the selected four SBI benchmarks are rather uninteresting and not specifically designed to assess the impact of missingness).

**Weaknesses:**

*Major points*

- The first contribution, namely, the “formalization” of the problem in SBI, is a bit of a stretch, as it is simply the standard missing data setting (i.e., equations 2–3 are not specific to approximate posteriors obtained via SBI). Propositions 1 and 2 are straightforward, and there is some notational confusion which may lead one to think that $x_{obs}$ is the real (non-simulated) data, while it is just a partition of the simulated data. Why not simply write the expectation as running over the augmented joint model $p(x_{miss}, x_{obs},\theta)$ in Proposition 2?

- The method is somewhat of an overkill for simple data, such as the 1D Ricker model used to pitch it, where the bias can easily be eliminated via data augmentation and a missingness mask provided as an additional input to the network. The demonstration in Figure 1 is simply bad practice and an example of model misspecification, as the networks have never seen imputed values during training. The work by Wang et al. (2024) is also somewhat misrepresented in the current paper, as they do not simply impute the data with constant values but use data augmentation and a mask indicator, which can be a much more efficient, albeit less sophisticated, approach for simpler cases (e.g., MCAR / MAR) [see also point on potential difficulties in learning the imputation model).

- The paper lacks key ablation studies demonstrating the impact of a bad $p(x_{miss}∣x_{obs})$ model. I assume such a model will lead to unreasonable variance inflation of the posterior and hence miscalibration (see next point). The authors themselves acknowledge that “learning the imputation model correctly is central to RISE’s performance,” so it seems paramount to quantify the impact of approximation error in $p(x_{miss}∣x_{obs})$.

- It is important to add calibration error as an additional, practically relevant coverage metric to all experiments, besides simply looking at MMD and Bayes RMSE. I suspect—and am open to being proven wrong—that difficulties in learning $p(x_{miss}∣x_{obs})$ will result in rather poor calibration, which can easily go undetected by RMSE or MMD (values are heavily kernel-dependent).

*Minor points*

- The notation needs some polishing, e.g., bold font is used for data vectors, but parameter vectors are not bold. On a related note, I believe it would be informative to introduce precise notation for sequences of vectors vs. vectors and focus the discussion on sequences, since, e.g., missing points in set-based data can be trivially handled by summarizing the reduced set, whereas missing points in temporal or spatiotemporal data present a real challenge.

- Some citations have inaccuracies, e.g., Gloeckler et al. is not an arxiv paper but an ICML paper, Radev et al. (2022) should be Radev et al. (2020), and so on.

- It would be nice for **Algorithm 1** to illustrate that the method results in an ensemble of posteriors (i.e., one for each sample from the imputation model) and that this ensemble uncertainty is integrated out for inference.

- I could not find any details on simulation budgets for the experiments. It would be nice to quantify performance as a function of simulation budget in at least one of the experiments.

**Questions:**

- What are the benefits of learning a latent variable model for the predictive missing data distribution instead of directly parameterizing it using another flexible generative network (e.g., a diffusion model)?

- What is M in equation 7?

- What are the error bars computed over in Figures 3 and 5?

---

> ### Author Response · Authors · 2024-11-21
> **Response**
>
> Many thanks for your constructive review. We address your concerns and incorporate all your suggestions below.
>
> > Why not simply write the expectation as running over the augmented joint model $p(x\_{obs},x\_{mis},\theta)$ in Proposition 2?
>
> Please recall that our optimization objective can be compactly written as
>
> $\mathrm{argmin}_{\phi, \varphi} \,\mathbb{E}\_{\mathbf{x}\_{\text{obs}} \sim p\_{\text{true}}}  \text{KL}[p\_{\text{true}} (\theta,\mathbf{x}\_{\text{mis}} \mid \mathbf{x}\_{\text{obs}} ) \mid \mid \underbrace{r\_{\phi,\varphi}(\theta,\mathbf{x}\_{\text{mis}} \mid \mathbf{x}\_{\text{obs}})}\_{\textbf{(joint imputation and inference)}}]$
>
> where our goal is to learn $r_{\phi,\varphi}$ parameterized by $\psi = (\phi, \varphi$).
>
> In case we know $p\_{\text{true}}(\mathbf{x}\_{\text{mis}} \mid \mathbf{x}\_{\text{obs}}$) (e.g., in controlled simulation settings), the above objective can indeed be written in terms of a joint expectation over the augmented model, as you noted. However, here we also address typical real-world scenarios, where we do not have access to $p\_{\text{true}}(\mathbf{x}\_{\text{mis}} \mid \mathbf{x}\_{\text{obs}})$. Since samples for $\mathbf{x}_{\text{mis}}$ are required during training, one needs to resort to methods such as variational approximation, or expectation maximization (EM).
>
> Here, we aim to jointly model the distribution via a probabilistic imputation model, treating $\mathbf{x}\_{\text{mis}}$ as latents, in a setting akin to how variational autoencoders (VAEs) leverage an encoder-decoder style architecture for variational approximation. Specifically, the imputation model needs to estimate these latents for the inference network to map them to the output space.  Both networks are highly coupled, as the distribution induced by the imputation model (or our encoder) shapes the input of the inference network (i.e., our decoder), and training them independently can lead to a distributional misalignment.
>
> We formalize this explicitly to make the exposition clear based on your comments.
>  We begin by noting that the above objective is equivalent to
>
> $$
> \mathrm{argmax}\_{\psi} \, \mathbb{E}\_{\mathbf{x}\_{\text{obs}} \sim p\_{\text{true}}} \, \mathbb{E}\_{p\_{\text{true}}(\theta,\mathbf{x}\_{\text{mis}} \mid \mathbf{x}\_{\text{obs}} )} [\log\, r\_{\psi}(\theta,\mathbf{x}\_{\text{mis}} \mid \mathbf{x}\_{\text{obs}})] \\
> = \mathrm{argmax}\_{\psi} \, \mathbb{E}\_{(\mathbf{x}\_{\text{obs}}, \theta, \mathbf{x}\_{\text{mis}}) \sim p\_{\text{true}}} \, [\log\, r\_{\psi}(\theta,\mathbf{x}\_{\text{mis}} \mid \mathbf{x}\_{\text{obs}})]$$
>
> Typically, in realistic scenarios,  we do not know $p\_{\text{true}}(\mathbf{x}\_{\text{mis}})$, but can only assume access to samples $(\mathbf{x}\_{\text{obs}}, \theta) \sim p_{\text{true}}$. Therefore, we treat $\mathbf{x}\_{\text{mis}}$ as latent variables sampled from some tractable (e.g., Gaussian) distribution $p(\cdot|\mathbf{x}\_{\text{obs}})$  conditioned on $\mathbf{x}\_{\text{obs}}$, and resort to a variational approximation.
>
> Specifically, we maximize a lower bound on
> $\mathbb{E}\_{(\mathbf{x}\_{\text{obs}}, \theta) \sim p\_{\text{true}}} \, [\log\, r\_{\psi}(\theta \mid \mathbf{x}\_{\text{obs}})]$. Formally, we have
>
> \begin{align}
> \mathbb{E}\_{(\mathbf{x}\_{\text{obs}},\theta) \sim p\_{\text{true}}} \log r\_{\psi}(\theta \mid \mathbf{x}\_{\text{obs}}) = \mathbb{E}\_{(\mathbf{x}\_{\text{obs}},\theta) \sim p\_{\text{true}}} \log \int r\_{\psi}(\theta,\mathbf{x}\_{\text{mis}} \mid \mathbf{x}\_{\text{obs}})d\mathbf{x}\_{\text{mis}} \\\\
> = \mathbb{E}\_{(\mathbf{x}\_{\text{obs}},\theta) \sim p\_{\text{true}}} \log \int \frac{p(\mathbf{x}\_{\text{mis}} \mid \mathbf{x}\_{\text{obs}})r\_{\psi}(\theta,\mathbf{x}\_{\text{mis}} \mid \mathbf{x}\_{\text{obs}})}{p(\mathbf{x}\_{\text{mis}} \mid \mathbf{x}\_{\text{obs}})} d\mathbf{x}\_{\text{mis}} \\\\
> \geq \mathbb{E}\_{(\mathbf{x}\_{\text{obs}},\theta) \sim p\_{\text{true}}} \mathbb{E}\_{\mathbf{x}\_{\text{mis}} \sim p(\mathbf{x}\_{\text{mis}} \mid \mathbf{x}\_{\text{obs}})} \left[\log  \frac{r\_{\psi}(\theta,\mathbf{x}\_{\text{mis}} \mid \mathbf{x}\_{\text{obs}})}{p(\mathbf{x}\_{\text{mis}} \mid \mathbf{x}\_{\text{obs}})} \right] \\\\
> = \mathbb{E}\_{(\mathbf{x}\_{\text{obs}},\theta) \sim p\_{\text{true}}} \mathbb{E}\_{\mathbf{x}\_{\text{mis}} \sim p(\mathbf{x}\_{\text{mis}} \mid \mathbf{x}\_{\text{obs}})} \left[\log  \frac{r\_{\psi}(\mathbf{x}\_{\text{mis}} \mid \mathbf{x}\_{\text{obs}}) r\_{\psi}(\theta \mid \mathbf{x}\_{\text{obs}}, \mathbf{x}\_{\text{mis}})}{p(\mathbf{x}\_{\text{mis}} \mid \mathbf{x}\_{\text{obs}})} \right]
> \end{align}
>
> where we invoked the Jensen's inequality to swap the log and the conditional expectation. Splitting parameters $\psi$ into imputation parameters $\varphi$ and inference parameters $\phi$, and denoting the corresponding imputation and inference networks by $\hat{p}\_{\varphi}$ and $q\_{\phi}$ respectively, we immediately get.... (continued in next response)

---

> > ### Author Response · Authors · 2024-11-21
> > **Response Contd.**
> >
> > \begin{align}
> > \mathbb{E}\_{(\mathbf{x}\_{\text{obs}},\theta) \sim p\_{\text{true}}} \log r\_{\phi, \varphi}(\theta \mid \mathbf{x}\_{\text{obs}}) \geq  \mathbb{E}\_{(\mathbf{x}\_{\text{obs}},\theta) \sim p\_{\text{true}}} \mathbb{E}\_{\mathbf{x}\_{\text{mis}} \sim p(\mathbf{x}\_{\text{mis}} \mid \mathbf{x}\_{\text{obs}})} \left[\log  \frac{\hat{p}\_{\varphi}(\mathbf{x}\_{\text{mis}} \mid \mathbf{x}\_{\text{obs}}) q\_{\phi}(\theta \mid \mathbf{x}\_{\text{obs}}, \mathbf{x}\_{\text{mis}})}{p(\mathbf{x}\_{\text{mis}} \mid \mathbf{x}\_{\text{obs}})} \right]
> > \end{align}
> >
> > Thus, we obtain the following variational objective:
> >
> > \begin{align}
> >  \mathrm{argmax}\_{\phi, \varphi}   \mathbb{E}\_{(\mathbf{x}\_{\text{obs}},\theta) \sim p\_{\text{true}}} \mathbb{E}\_{\mathbf{x}\_{\text{mis}} \sim p(\mathbf{x}\_{\text{mis}} \mid \mathbf{x}\_{\text{obs}})} \left[\log  \frac{\hat{p}\_{\varphi}(\mathbf{x}\_{\text{mis}} \mid \mathbf{x}\_{\text{obs}}) q_{\phi}(\theta \mid \mathbf{x}\_{\text{obs}}, \mathbf{x}\_{\text{mis}})}{p(\mathbf{x}\_{\text{mis}} \mid \mathbf{x}\_{\text{obs}})} \right]
> > \end{align}
> >
> > \begin{align}
> > = \mathrm{argmax}\_{\phi, \varphi}  \mathbb{E}\_{(\mathbf{x}\_{\text{obs}},\theta) \sim p\_{\text{true}}} \left(\mathbb{E}\_{\mathbf{x}\_{\text{mis}} \sim p(\mathbf{x}\_{\text{mis}} \mid \mathbf{x}\_{\text{obs}})} \left[\log  \hat{p}\_{\varphi}(\mathbf{x}\_{\text{mis}} \mid \mathbf{x}\_{\text{obs}}) + \log q\_{\phi}(\theta \mid \mathbf{x}\_{\text{obs}}, \mathbf{x}\_{\text{mis}})\right] + \mathbb{H}(p(\mathbf{x}\_{\text{mis}} \mid \mathbf{x}\_{\text{obs}})\right)
> > \end{align}
> >
> > \begin{align}
> >  =  \mathrm{argmax}\_{\phi, \varphi} \mathbb{E}\_{(\mathbf{x}\_{\text{obs}},\theta) \sim p\_{\text{true}}} \mathbb{E}\_{\mathbf{x}\_{\text{mis}} \sim p(\mathbf{x}\_{\text{mis}} \mid \mathbf{x}\_{\text{obs}})} \left[\log  \underbrace{\hat{p}\_{\varphi}(\mathbf{x}\_{\text{mis}} \mid \mathbf{x}\_{\text{obs}})}\_{\text{imputation}} + \log \underbrace{q\_{\phi}(\theta \mid \mathbf{x}\_{\text{obs}}, \mathbf{x}\_{\text{mis}})}\_{\text{inference}}\right]~,
> > \end{align}
> >
> > since the entropy term $\mathbb{H}(p(\mathbf{x}\_{\text{mis}} \mid \mathbf{x}\_{\text{obs}})$ does not depend on the optimization variables $\phi$ and $\varphi$.
> >
> > This is exactly the objective that we optimize for. Based on your comment, we have now revised section 3.2, Proposition 2, and it's proof in Appendix A.2.2 in the paper to make this clear.
> >
> > > The first contribution, namely, the “formalization” of the problem in SBI,.... simulated data.
> >
> > Our objective was to bring to the attention that irrespective of how good the inference network is, inaccurate imputation can be detrimental to the performance of SBI methods. Based on your comments,  we have now revised the pdf with "motivation" replacing "formalization". We have also added a footnote in page 3 of the revised manuscript clarifying the notational confusion about observed data.
> >
> > > Some citations have inaccuracies,
> >
> > Thank you for pointing this out. We haved fixed it in the revised version of the paper.
> >
> > > simulation budgets for the experiments....
> >
> > **Ablation w.r.t simulation budget**. We used a simulation budget of 1000 for each task and experiment. As per your suggestion, we have conducted a study to quantify the performance as a function of the simulation budget on GLU and GLM dataset. The table below shows C2ST and MMD among various simulation budgets for RISE, for $10 \%$ missingness level. As the budget increases, the performance improves.
> >
> > | Budget | GLU  | GLU  | GLM  | GLM  |
> > |--------|------|------|------|------|
> > |        | C2ST | MMD  | C2ST | MMD  |
> > | 1000   | 0.83 | 0.18 | 0.80 | 0.12 |
> > | 10000  | 0.78 | 0.15 | 0.75 | 0.10 |
> >
> > > What is M in equation 7?
> >
> > It was a typo. It should be $m$ (instead of $M$), which is the number of Monte-Carlo samples from the latent distribution. Thanks for noticing.
> >
> > > What are the error bars computed over in Figures 3 and 5?
> >
> > The error bars are computed over 10 seed runs of the model.
> >
> > > It would be nice for Algorithm 1 to illustrate that the method results in an ensemble of posteriors
> >
> > Good point. We have added the following sentence to Section 3.3 (lines 309-311) of the revised manuscript to highlight this point:
> >
> > ``For each sample from the imputation model, we obtain a posterior distribution via the inference network, thus resulting in an ensemble of posterior distributions across all samples.''

---

> > > ### Author Response · Authors · 2024-11-21
> > > **Response Contd.**
> > >
> > > > It is important to add calibration error as an additional.....
> > >
> > > **Calibration Plots for RISE**. Based on suggestion, we have added the expected coverage plot for the HH and GLU task in Appendix B.7 (Figure 7), which shows that RISE yields better calibrated posteriors than NPE-NN baseline.
> > > In addition, we have also evaluated our methods on nominal log posterior probability metric and C2ST for all missingness levels, shown below. We have added the results in the revised manuscript in Appendix B.3.
> > >
> > > | Missingness             | Method   | GLU           | GLU                       | GLM           | GLM                       | Ricker        | Ricker                    | OUP           | OUP                       |
> > > |-------------------------|----------|---------------|---------------------------|---------------|---------------------------|---------------|---------------------------|---------------|---------------------------|
> > > |                         |          | C2ST          | NLPP                      | C2ST          | NLPP                      | C2ST          | NLPP                      | C2ST          | NLPP                      |
> > > | 10% | NPE-Zero | 0.89          | -2.77 $\pm$ 0.13          | 0.87          | -6.92 $\pm$ 0.14          | 0.95          | -5.15 $\pm$ 0.12          | 0.90          | -2.61 $\pm$ 0.16          |
> > > |                         | NPE-Mean | 0.88          | -2.67 $\pm$ 0.16          | 0.85          | -6.83  $\pm$ 0.14         | 0.95          | -5.10  $\pm$ 0.21         | 0.90          | -2.51 $\pm$ 0.11          |
> > > |                         | NPE-NN   | 0.87          | -2.51 $\pm$ 0.11          | 0.84          | -6.57$\pm$ 0.13           | 0.94          | -4.90  $\pm$ 0.16         | 0.89          | -2.25  $\pm$ 0.18         |
> > > |                         | RISE     | **0.83** | **-2.31** $\pm$ 0.10 | **0.80** | **-6.32** $\pm$ 0.15 | 0.90          | **-4.20** $\pm$ 0.09 | **0.87** | **-2.09** $\pm$ 0.11 |
> > > | 25% | NPE-Zero | **0.88** | -4.11 $\pm$ 0.17          | 0.97          | -8.05 $\pm$ 0.20          | 0.96          | -5.10 $\pm$ 0.16          | 0.92          | -2.97 $\pm$ 0.13          |
> > > |                         | NPE-Mean | 0.90          | -3.99 $\pm$ 0.21          | 0.94          | -7.92  $\pm$ 0.14         | 0.96          | -5.05 $\pm$ 0.11          | 0.92          | -2.84 $\pm$ 0.15          |
> > > |                         | NPE-NN   | 0.91          | -3.92 $\pm$ 0.11          | 0.93          | -7.72  $\pm$ 0.16         | 0.95          | -4.94$\pm$ 0.17           | 0.90          | -2.74 $\pm$ 0.18          |
> > > |                         | RISE     | 0.89          | **-3.71** $\pm$ 0.11 | **0.91** | **-7.22** $\pm$ 0.17 | **0.92** | **-4.64** $\pm$ 0.15 | **0.89** | **-2.43** $\pm$ 0.15 |
> > > | 60% | NPE-Zero | 0.97          | -6.98 $\pm$ 0.18          | 1.00          | -9.63 $\pm$ 0.14          | 0.97          | -5.17  $\pm$ 0.15         | 0.96          | -3.07 $\pm$0.12           |
> > > |                         | NPE-Mean | 0.98          | -6.76 $\pm$ 0.09          | 0.99          | -9.27  $\pm$ 0.14         | 0.97          | -5.10 $\pm$ 0.18          | 0.95          | -2.97 $\pm$ 0.12          |
> > > |                         | NPE-NN   | 0.96          | -6.37  $\pm$ 0.12         | 0.99          | -9.02 $\pm$ 0.17          | 0.96          | -4.97 $\pm$ 0.17          | 0.95          | -2.87 $\pm$ 0.19          |
> > > |                         | RISE     | **0.93** | **-6.21** $\pm$ 0.11 | **0.97** | **-8.71** $\pm$ 0.14 | **0.94** | **-4.72** $\pm$ 0.17 | **0.93** | **-2.52** $\pm$ 0.11 |
> > >
> > >
> > > > The notation needs some polishing....
> > >
> > > Thank you for pointing this out. We will make all the vectors bold. We focus on the case where there is a single vector $\mathbf{x}$ (such as a time-series) of dimension $d$ for each parameter vector, where some values in this data vector can be missing. However, our formulation can be extended straightforwardly to the multiple observations case where we obtain $\mathbf x^{(1:m)} = (\mathbf x_1, \dots, \mathbf x_m)$ for each $\theta$. Then, $\mathbf{x}^{(1:m)} = (\mathbf{x}^{(1:m)}\_{\text{obs}}, \mathbf{x}^{(1:m)}\_{\text{mis}})$, and the objective for RISE becomes
> > >
> > > $$
> > > \arg \min\_{\phi, \varphi, \kappa} -   \mathbb{E}\_{(\mathbf{x}^{(1:m)}\_{\text{obs}},\theta) \sim p\_{\text{true}}} \mathbb{E}_{\mathbf{x}^{(1:m)}]\_{\text{mis}} \sim \prod\_{i=1}^m p(\mathbf{x}^{(i)}\_{\text{mis}} \mid \mathbf{x}^{(i)}\_{\text{obs}})} \left[\dfrac{1}{m} \sum\_{i=1}^m\log  \underbrace{\hat{p}\_{\varphi}(\mathbf{x}^{(i)}\_{\text{mis}} \mid \mathbf{x}^{(i)}\_{\text{obs}})}\_{\textbf{(imputation)}} + \log \underbrace{q\_{\phi}(\theta \mid \eta\_\kappa (\mathbf{x}^{1:m}\_{\text{obs}}, \mathbf{x}^{1:m}\_{\text{mis}}))}\_{\textbf{(inference)}}\right].
> > > $$
> > >
> > > Note that here we have to summarize the data using the network $\eta\_\kappa$ before passing the data into the inference network, as NPE is unable to handle multiple observations, unless we use recent extensions based on score estimation [3,4].

---

> ### Author Response · Authors · 2024-11-21
> **Response Contd.**
>
> [3] Geffner et al. (2023). Compositional score modeling for simulation-based inference. ICML.
>
> [4] Linhart et al. (2024). Diffusion posterior sampling for simulation-based inference in tall data settings, arxiv 2404.07593.
>
> > work by Wang et al. (2024) is also somewhat misrepresented......
>
> Thank you for bringing this to our attention. We have revised and added a detailed description of Wang et al. (2024) in the revised version of the manuscript (section 4).
>
> **Comparison with Wang et al. (2024)** Agreed. Thank you for pointing this out. We have performed a study to compare to the method listed in Wang et al. (2024), by using data augmentation with constant values (zero) and a binary mask indicator while training the NPE on GLU and GLM data sets at various levels of missingness. The table below presents the results. We observe that RISE is able to outperform the baseline in numerous cases, even at higher level of missingness.
>
> | Missingness             | Method             | GLU           | GLU           | GLM           | GLM           |
> |-------------------------|--------------------|---------------|---------------|---------------|---------------|
> |                         |                    | RMSE          | C2ST          | MMD           | RMSE          | C2ST          | MMD           |
> | 10% | Wang et al. (2024) | 0.47          | 0.87          | 0.28          | 0.61          | 0.86          | 0.24          |
> |                         | RISE               | **0.41** | **0.83** | **0.18** | 0.65          | **0.80** | **0.12** |
> | 25% | Wang et al. (2024) | 0.45          | 0.92          | 0.31          | 0.86          | 0.94          | 0.35          |
> |                         | RISE               | **0.43** | **0.89** | **0.26** | 0.93          | **0.91** | **0.27** |
> | 60% | Wang et al. (2024) | 0.65          | 0.97          | 0.35          | 1.53          | 0.99          | 0.45          |
> |                         | RISE               | **0.56** | **0.93** | **0.33** | **1.27** | **0.97** | 0.50          |
>
> > ......Impact of a bad $p(x\_{mis} \mid x\_{obs})$.....
>
> Thank you for the suggestion. We actually did perform a study to test the imputation performance of RISE on real-world bio-activity datasets, see the first paragraph of Section 5.4. There we used the $R^2$ metric which is defined as  $R^2 = 1 - \left( \sum\_i ( \tilde{x}\_{\text{mis},i} - x\_{\text{mis},i})^2/(x\_{\text{mis},i} - \bar{x}\_{\text{mis}})^2\right)$, where $\tilde{x}\_{\text{mis},i}$ and $x\_{\text{mis},i}$ are the $i^{th}$ predicted and true missing value, respectively, and $\bar{x}\_{\text{mis}}$ is the sample mean, to quantify the imputation accuracy, and compared RISE with the popular imputation methods from that field.
>
> We copy the results from Table 1 below for convenience. We observe that the neural processes-based imputation network of RISE achieves competitive imputation performance.
>
> | Method   | Adrenergic               | Kinase                   |
> |----------|--------------------------|--------------------------|
> | QSAR     | (N/A)                    | -0.19 $\pm$ 0.01         |
> | CMF      | 0.59 $\pm$ 0.02          | -0.11 $\pm$ 0.01         |
> | DNN      | 0.60 $\pm$ 0.05          | 0.11 $\pm$ 0.01          |
> | NP       | 0.61 $\pm$ 0.03          | 0.17 $\pm$ 0.04          |
> | Conduilt | 0.62 $\pm$ 0.04          | 0.22 $\pm$ 0.03          |
> | CNP      | 0.65 $\pm$ 0.04          | 0.24 $\pm$ 0.02          |
> | RISE     | **0.67** $\pm$ 0.03 | **0.26** $\pm$ 0.03 |
>
> > benefits of learning a latent variable model......
>
> Recall that our optimization objective can be compactly written as
> \begin{align}
>           \mathrm{argmin}\_{\phi, \varphi} \mathbb{E}\_{\mathbf{x}\_{\text{obs}} \sim p\_{\text{true}}} \, \text{KL}\_{\text{true}}(\theta,\mathbf{x}\_{\text{mis}} \mid \mathbf{x}\_{\text{obs}} ) \mid \mid \underbrace{r\_{\phi,\varphi}(\theta,\mathbf{x}\_{\text{mis}} \mid \mathbf{x}\_{\text{obs}})}\_{\textbf{(joint imputation and inference)}}]~,
> \end{align}
> where our goal is to learn $r_{\phi,\varphi}$ parameterized by $\psi = (\phi, \varphi$).
>
> In principle, one could model the joint distribution $r_{\phi,\varphi}$ via a flexible generative framework such as diffusion models. However, sampling and training diffusion models are computationally expensive, which is why we resort to jointly modeling the distribution via a probabilistic imputation model, treating $\mathbf{x}_{\text{mis}}$ as latents, in a setting akin to how variational autoencoders (VAEs) leverage an encoder-decoder style architecture for variational approximation, which is more efficient.
>
> We are grateful for your thoughtful comments and suggestions, which have allowed us to emphasize some salient aspects, and shed light on subtle facets of the proposed method. We hope that our response addresses your concerns.

---

> > ### Comment · Area_Chair_egPw · 2024-11-26
> >
> > Dear reviewer,
> >
> > Please make to sure to read, at least acknowledge, and possibly further discuss the authors' responses to your comments. Update or maintain your score as you see fit.
> >
> > The AC.

---

> > ### Comment · Reviewer_jx4r · 2024-11-26
> >
> > I thank the authors for their comprehensive response, clarifications, and additional results. I believe this is a valuable work for the field and recommend acceptance.

---

### Official Review · Reviewer_La5K · 2024-11-05

**Soundness:** 3
**Presentation:** 3
**Contribution:** 3
**Rating:** 8
**Confidence:** 5

**Summary:**

The paper introduces a method for addressing the issue of missing data in the context of
simulation-based inference (SBI). The approach uses neural processes to learn an
imputation model from simulated data and then combines the learned imputation model
with standard neural posterior estimation (NPE) to perform SBI. The authors first show
how previous approaches for addressing missing data with zero- or mean-value imputation
can lead to biased SBI posteriors. They then evaluate their approach on a set of
tractable SBI benchmarking tasks and two intractable tasks. Additionally, they perform
ablation studies and show how their method can be extended to generalize over different
levels of missingness in the data.

**Strengths:**

The paper is well-written and clearly structured. The figures are visually appealing,
and the results are reported over a reasonable number of repetitions and with error
bars. The introduction and background section are concise and easy to follow, except
maybe for section 3.3, for which additional background is given in the appendix.

### Originality

The paper addresses a known problem that has been tackled in the field of SBI before.
However, it shows that previous approaches had limited success. It then proposes a new
method that combines techniques from existing research on learning imputation. Thus,
overall, using neural processes for learning imputation models for neural SBI is not a
strong technical contribution in itself, but can still be a valuable contribution to the
field of SBI. However, while most of the related work on missing data in SBI is cited
accurately, it seems that the work by Gloeckler et al. is actually very similar to the
approach proposed here. See below for details.

### Quality

The derivations for combining imputation models with NPE appear technically sound. The
experimental results support the initial claims that the proposed method is more robust
to missing data than the baselines. However, the choice of performance metrics and the
comparison to previous methods should be improved (see below).

### Clarity

As mentioned above, I believe the paper is presented quite clearly. What I am missing is
a discussion of the potential limitations of the presented method, see below.

### Significance

The paper addresses an important problem in the field of SBI, as actual SBI applications
usually deal with real-world observations that often have missing data. This problem has
gained only little attention in the literature so far. If the concerns on the evaluation
listed below are addressed and the method turns out to perform better than previous
approaches, it will be a valuable addition to the field.

**Weaknesses:**

### Missing discussion of previous work

- There is one early SBI paper that addressed imputation in SBI that is missing and
  should be discussed: Lueckmann et al. 2017 automatically learn imputation values for
  NPE using an MDN embedding network, actually on the same Hodgkin-Huxley benchmark as
  used here. A discussion of this paper and a comparison to their proposed method would be
  appropriate. In particular, in section 3.3 and figure 4, they use an imputation model in the last layer of the MDN, to which you could
  compare. A comparison to this approach could be straightforward as your NPE-NN baseline approach seems quite similar to their
  approach. Ideally, you can show how your approach leads to more accurate imputation values on the benchmarking or the HH task, as
  they actually observed that their learned imputation values tend to be close to the sample mean of the feature.

- The discussion of the work by Gloeckler et al. is not accurate. The Simformer actually
  learns the imputation of the missing data as well. In that sense, it is actually very
  similar to the approach proposed here. Relating to the example given in the paper:

  > However, this method estimates the partial posterior distribution p(θ | x1, x3)
  > given x = [x1, x2, x3, x4, x5], where x1 & x3 are the only observed variables,
  > without considering the mechanisms that lead to missing data.

  This is not correct. The SIMFORMER can perform *arbitrary* conditioning and
  evaluation. Thus, when given only x1 & x3, it can actually predict p(θ, x2, x4, x5 |
  x1, x3), which then serves as an imputation model for the missing data points (e.g.,
  just sampling from p(θ, x2, x4, x5 | x1, x3) and ignoring x2, x4, x5 would amount to
  obtain an approximation to p(θ | x2, x4, x5, x1, x3)).  A more detailed and more
  accurate discussion of this work would be essential here. The code for applying the
  Simformer is available at https://github.com/mackelab/simformer and appears to be
  well-documented. Ideally, it would possible to show on one of the benchmarking tasks how RISE leads to better imputation by being able to explicitly model different types of missingness, which seems to be the distinctive feature compared to the Simformer.

### Choice of performance metrics

The choice of MMD and RMSE is not ideal. MMD can be misleading depending on the choice
of kernel bandwidth (Lueckmann et al. 2021) and RMSE seems to measure accurate parameter
discovery, although posteriors do not have to be centered on the true parameters at all.
I suggest the following
- In addition to MMD, calculate **C2ST** as well as it gives an absolut and interpretable and not just a relative comparison to reference posteriors.
- instead of RMSE, calculate the the **nominal log posterior probability**, i.e., obtaining the NPE posterior for each x in the test set,
and then averaging in the log probs of the corresponding thetas (as done in Papamakarios
et al. 2019, Greenberg et al. 2019, and discussed in detail in Lueckmann et al. 2021). The log nominal density, when averaged over many test data points, is a relative comparison metric of posterior accuracy. This would be more appropriate than using RMSE because it measures posterior accuracy and not just parameter discovery accuracy.

Additionally, it would be good to also evaluate the calibration properties of the
inferred posteriors, e.g., by calculating the SBC or expected coverage, at least on the
four benchmarking tasks. This will show how well-calibrated the different methods are.

### Discussion of data and computational requirements

It would be essential to give more details on how the training data set for RISE is
constructed. Given the large number of additional NN parameters required for training
the imputation model, I would expect that RISE needs more simulations for training
compared to naive imputation or NPE-NN. Concretely, how many simulations where used for training RISE, or more generally, what are the simulation budgets used for the different benchmark tasks and methods?

Same for the computational resources. How much more effort in terms of data and
resources does the user have to put in in order to use RISE? It would be good for the reader to see comparisons of training time and memory usage of the different methods.

**Questions:**

What are the simulation budgets used for the different benchmarks and methods?

---

> ### Author Response · Authors · 2024-11-21
> **Response**
>
> Many thanks for your feedback and suggestions. We address all your comments below.
> > There is one early SBI paper .... the sample mean of the feature
>
> **RISE compares favorably with Lueckmann et al. 2017.** Thank you for providing the reference and apologies for missing it. We have included that reference in the related works. Indeed, Lueckmann et al. 2017 learn a single imputation value for the missing feature, and utilize it to perform inference. However, they perform the imputation at the input layer of their posterior network instead of the last layer (Section 2.3 of their paper, in the ``Bad feature'' paragraph), which turns out to be the same as our NPE-NN baseline. We have made a note of this in the main text of our revised manuscript, and plotted the coverage of RISE and this baseline in Figure 7 (Appendix B.7). We observe that RISE is able to produce conservative posterior approximations better than NPE-NN and is better calibrated than NPE-NN for all missingness levels.
>
> > The discussion of the work by Gloeckler et al. ..... this work would be essential here.
>
> Thank you very much for bringing this to our attention. Based on your comment, we have revised the description of Simformer in our related works.
>
> > The code for applying the Simformer ..... compared to the Simformer.
>
> **Benchmarking with Simformer.** Based on your suggestion, we have compared RISE's performance against Simformer for the GLU and GLM task. The results (in terms of MMD and nominal log posterior probability (NLPP)) for both the MCAR and the MNAR (also in Appendix B.2 of the revised paper). For the MCAR case, both RISE and Simformer perform similarly, with RISE yielding slightly better results. The difference in performance is more stark in the MNAR case, as expected, since Simformer does not explicitly model the missingness mechanism.
>
> ### MCAR:
> | Missingness             | Method    | GLU                       | GLU                      | GLM                       | GLM                      |
> |-------------------------|-----------|---------------------------|--------------------------|---------------------------|--------------------------|
> |                         |           | NLPP                      | MMD                      | NLPP                      | MMD                      |
> | 10%                  | Simformer | -2.45 $\pm$ 0.12          | 0.20 $\pm$ 0.01          | -6.47 $\pm$ 0.16          | 0.17 $\pm$ 0.02          |
> |                         | RISE      | **-2.31** $\pm$ 0.10 | **0.18** $\pm$ 0.01 | **-6.32** $\pm$ 0.15 | **0.12** $\pm$ 0.02 |
> | 25% | Simformer | **-3.65** $\pm$ 0.17 | 0.27  $\pm$ 0.02         | -7.37 $\pm$ 0.13          | 0.31 $\pm$ 0.03          |
> |                         | RISE      | -3.71 $\pm$ 0.11          | **0.26** $\pm$ 0.01 | **-7.22** $\pm$ 0.17 | **0.27** $\pm$ 0.01 |
> | 60% | Simformer | -6.62 $\pm$ 0.27          | 0.39 $\pm$ 0.03          | -8.93 $\pm$ 0.18          | 0.52 $\pm$ 0.03          |
> |                         | RISE      | **-6.21** $\pm$ 0.11 | **0.33** $\pm$ 0.02 | **-8.71** $\pm$ 0.14 | **0.50** $\pm$ 0.01 |
>
> ### MNAR:
> | Missingness | Method    | GLU                       | GLU                      | GLM                       | GLM                      |
> |-------------|-----------|---------------------------|--------------------------|---------------------------|--------------------------|
> |             |           | NLPP                      | MMD                      | NLPP                      | MMD                      |
> | 10%      | Simformer | -2.15 $\pm$ 0.10          | 0.18 $\pm$ 0.01          | -6.17 $\pm$ 0.18          | 0.16 $\pm$ 0.02          |
> |             | RISE      | **-1.90** $\pm$ 0.09 | **0.16** $\pm$ 0.01 | **-5.82** $\pm$ 0.11 | **0.13** $\pm$ 0.02 |
> | 25%      | Simformer | **-3.12** $\pm$ 0.12 | 0.25  $\pm$ 0.02         | -6.57 $\pm$ 0.14          | 0.25 $\pm$ 0.03          |
> |             | RISE      | -3.26 $\pm$ 0.10          | **0.22** $\pm$ 0.01 | **-6.12** $\pm$ 0.15 | **0.17** $\pm$ 0.01 |
> | 60%      | Simformer | -6.02 $\pm$ 0.12          | 0.32 $\pm$ 0.03          | -7.56 $\pm$ 0.15          | 0.50 $\pm$ 0.03          |
> |             | RISE      | **-5.80** $\pm$ 0.13 | **0.27** $\pm$ 0.04 | **-7.11** $\pm$ 0.17 | **0.47** $\pm$ 0.03 |

---

> ### Author Response · Authors · 2024-11-21
> **Response Contd.**
>
> > The choice of MMD and RMSE is not ideal
>
>  **C2ST & Additional metrics** Thank you for another excellent suggestion. We have now added the C2ST scores and the nominal log posterior probability (NLPP) metric for the benchmark tasks. The results are shown in the table below. The conclusion remains the same, with RISE outperforming the baselines in all but one case (in terms of C2ST). We have also included expected coverage plots for the HH and GLU task over various missingness levels in Figure 7 (Appendix B.7). We observe that the posterior produced by RISE becomes more conservative as the proportion of missing values in the data increases, which makes sense.
>
>    | Missingness             | Method   | GLU           | GLU                       | GLM           | GLM                       | Ricker        | Ricker                    | OUP           | OUP                       |
> |-------------------------|----------|---------------|---------------------------|---------------|---------------------------|---------------|---------------------------|---------------|---------------------------|
> |                         |          | C2ST          | NLPP                      | C2ST          | NLPP                      | C2ST          | NLPP                      | C2ST          | NLPP                      |
> | 10% | NPE-Zero | 0.89          | -2.77 $\pm$ 0.13          | 0.87          | -6.92 $\pm$ 0.14          | 0.95          | -5.15 $\pm$ 0.12          | 0.90          | -2.61 $\pm$ 0.16          |
> |                         | NPE-Mean | 0.88          | -2.67 $\pm$ 0.16          | 0.85          | -6.83  $\pm$ 0.14         | 0.95          | -5.10  $\pm$ 0.21         | 0.90          | -2.51 $\pm$ 0.11          |
> |                         | NPE-NN   | 0.87          | -2.51 $\pm$ 0.11          | 0.84          | -6.57$\pm$ 0.13           | 0.94          | -4.90  $\pm$ 0.16         | 0.89          | -2.25  $\pm$ 0.18         |
> |                         | RISE     | **0.83** | **-2.31** $\pm$ 0.10 | **0.80** | **-6.32** $\pm$ 0.15 | 0.90          | **-4.20** $\pm$ 0.09 | **0.87** | **-2.09** $\pm$ 0.11 |
> | 25% | NPE-Zero | **0.88** | -4.11 $\pm$ 0.17          | 0.97          | -8.05 $\pm$ 0.20          | 0.96          | -5.10 $\pm$ 0.16          | 0.92          | -2.97 $\pm$ 0.13          |
> |                         | NPE-Mean | 0.90          | -3.99 $\pm$ 0.21          | 0.94          | -7.92  $\pm$ 0.14         | 0.96          | -5.05 $\pm$ 0.11          | 0.92          | -2.84 $\pm$ 0.15          |
> |                         | NPE-NN   | 0.91          | -3.92 $\pm$ 0.11          | 0.93          | -7.72  $\pm$ 0.16         | 0.95          | -4.94$\pm$ 0.17           | 0.90          | -2.74 $\pm$ 0.18          |
> |                         | RISE     | 0.89          | **-3.71** $\pm$ 0.11 | **0.91** | **-7.22** $\pm$ 0.17 | **0.92** | **-4.64** $\pm$ 0.15 | **0.89** | **-2.43** $\pm$ 0.15 |
> | 60% | NPE-Zero | 0.97          | -6.98 $\pm$ 0.18          | 1.00          | -9.63 $\pm$ 0.14          | 0.97          | -5.17  $\pm$ 0.15         | 0.96          | -3.07 $\pm$0.12           |
> |                         | NPE-Mean | 0.98          | -6.76 $\pm$ 0.09          | 0.99          | -9.27  $\pm$ 0.14         | 0.97          | -5.10 $\pm$ 0.18          | 0.95          | -2.97 $\pm$ 0.12          |
> |                         | NPE-NN   | 0.96          | -6.37  $\pm$ 0.12         | 0.99          | -9.02 $\pm$ 0.17          | 0.96          | -4.97 $\pm$ 0.17          | 0.95          | -2.87 $\pm$ 0.19          |
> |                         | RISE     | **0.93** | **-6.21** $\pm$ 0.11 | **0.97** | **-8.71** $\pm$ 0.14 | **0.94** | **-4.72** $\pm$ 0.17 | **0.93** | **-2.52** $\pm$ 0.11 |
>
> > Computational cost ...memory usage of the different methods
>
> **Computational Complexity**: We used a simulation budget of 1000 for each task and experiment. We have performed an ablation study to evaluate the computational complexity of our method in comparison to standard NPE. The table below shows the time (in seconds) per epoch to train different models on a single V100 GPU with a batch size of 50 and 100. We observe that there is a moderate increase in runtime (roughly 50\% per epoch), which is expected due to the inclusion of the imputation model.
>
> | Batch Size| Method | GLU | GLM|
> |------|------|------|-------|
> | 50| NPE| 0.12| 0.10|
> | 50| RISE| 0.18| 0.16|
> | 100| NPE|0.20 |0.21 |
> | 100| RISE| 0.31| 0.29|
>
> We are grateful for your thoughtful feedback and suggestions. Acting on your feedback has helped appropriately position and reinforce the strengths of RISE, and we would appreciate if the same can be reflected in your stronger support for this work. Many thanks!

---

> > ### Comment · Reviewer_La5K · 2024-11-22
> >
> > Thank you for the rebuttal and for running additional comparisons with the SIMFORMER approach and calculating new metrics.
> > The results show that RISE performs on par or better. However, a couple of questions and concerns remain:
> >
> > - as pointed out in my review, I think C2ST and NLPP would be more appropriate metrics and I would hope the figures and tables in the main text will be adapted as well
> > - just to be sure, both NPE-NN and RISE are trained with only 1000 simulations in total?
> > - The performance in terms of C2ST on the benchmarks is rather poor for all methods, but it is based on only 1000 simulations. I think it would be informative to compare it with more simulations, say 10k, to get a clearer picture how the methods perform. With C2ST around 0,9, the posteriors will probably be really broad (can you show and example posterior from the benchmarks?) and the comparison is noisy in itself. Especially if you do not provide error bars on the comparison.
> > - as pointed out by other reviewers as well, I agree that the citations of previous work is incorrect to some extent and really should be fixed. In particular, for NPE. NPE as such, in its basic single-round version, was introduced by Papamakrios & Murray in 2016. Later, follow-up work extended it to other sequential variants (Lueckmann et al., Greenberg et al. Deistler et al.) or variants with permutation invariant embedding nets (radev et al. 2020). However, the amortized, single-round version you are using here goes back to Papamakrios & Murray in 2016 only. I think this should reflected in the manuscript, i.e., you should just remove the citations on Greenberg et al. Radev et al. and Zammit-Mangion et al. when you talk about NPE.

---

> > > ### Author Response · Authors · 2024-11-22
> > > **Response**
> > >
> > > Many thanks for your reply. We address your concerns below:
> > >
> > > > as pointed out in my review, I think C2ST and NLPP would be more appropriate metrics and I would hope the figures and tables in the main text will be adapted as well
> > >
> > > Absolutely. We will replace RMSE with C2St and NLPP in the main text. We just put these results in the appendix for now so that it is easier for all the reviewers to find them in one place.
> > >
> > > > just to be sure, both NPE-NN and RISE are trained with only 1000 simulations in total?
> > >
> > > Yes, both of the methods are trained with only 1000 simulations.
> > >
> > > > The performance in terms of C2ST on the benchmarks is rather poor for all methods, but it is based on only 1000 simulations. I think it would be informative to compare it with more simulations, say 10k, to get a clearer picture how the methods perform. With C2ST around 0,9, the posteriors will probably be really broad (can you show and example posterior from the benchmarks?) and the comparison is noisy in itself. Especially if you do not provide error bars on the comparison.
> > >
> > > Indeed, the C2ST scores are high due to limited number of simulations. Based on your suggestion, we ran the experiment with 10k simulations for Ricker and OUP tasks at $25\%$ missingness level. The C2ST scores (shown in the table below) decrease for both NPE-NN and RISE as the simulations increase. In the final version, we aim to do this for the other tasks (GLU and GLM) as well.
> > >
> > > | Budget | Method | Ricker | OUP|
> > > |------|------|------|------|
> > > |1000|NPE-NN| 0.95| 0.90|
> > > |1000| RISE| 0.92| 0.89|
> > > |10000|NPE-NN| 0.85| 0.81|
> > > |10000| RISE| **0.80**| **0.77**|
> > >
> > > We have also added a plot showing the Ricker and OUP posterior estimates obtained using RISE for both 1k and 10k simulations at 25% missingness level, see Figure 9 in Appendix B.8. As you expected, the posterior concentrates around the true parameter value as simulation budget increases.
> > >
> > > > as pointed out by other reviewers as well, I agree that the citations of previous work is incorrect ........
> > >
> > > Agreed. We have now included the Papamakarios and Murray 2016 paper as the primary reference for NPE.

---

> > > > ### Comment · Reviewer_La5K · 2024-11-26
> > > >
> > > > Thanks for your response and for running additional experiments. Overall, I am no more confident that your work is a valuable contribution. I will adapt my score accordingly.

---

> > > > > ### Author Response · Authors · 2024-11-26
> > > > >
> > > > > Thank you so much. We greatly appreciate your thorough feedback which helped us improve our work considerably.

---

> > > > > > ### Comment · Reviewer_La5K · 2024-11-26
> > > > > > **NPE with multipe observations**
> > > > > >
> > > > > > I agree with reviewer NU6H with the concern that your statement about NPE is incorrect. In the updated paper, lines 826-29 you are stating that NPE cannot handle multiple observations (IDD data) unless extended with recent score estimation approaches. This is not correct. NPE can easily handle IID data when used with a vanilla permutation-invariant embedding network (Zhan et al.), e.g., as shown in Chan et al. 2018.

---

> > > > > > > ### Author Response · Authors · 2024-11-27
> > > > > > > **Response**
> > > > > > >
> > > > > > > Many thanks for your time and constructive feedback. We fully agree with your remark - indeed, as you pointed out,  the NPE framework is very flexible and can accommodate multiple observations, e.g., when the inference network is augmented with a summary network (such as a Deep Set) in addition to the inference network.  Apologies for the confusion - the context for this particular statement was the NPE setting that employed only an inference network.
> > > > > > >
> > > > > > > We have now expanded Section 2 (where we introduce NPE) to elucidate the multiple observations scenario in the context of NPE. We have also replaced lines 826-829 with the following:
> > > > > > >
> > > > > > > > Note that we can summarize the data using the network $\eta_\kappa$ (for instance, a deep set (Zaheer et al. 2017)) before passing the data into the inference network, which is standard practice when using NPE with multiple observations (Chan et al., 2018). Alternatively, one could use recent extensions based on score estimation (Geffner et al. 2023, Linhart et al. 2024) as well.
> > > > > > >
> > > > > > > We hope this clarifies the versatility of the NPE framework. We're grateful for your insightful remarks that have helped us improve the overall presentation significantly

---

### Author Response · Authors · 2024-11-21
**Global Response**

We would like to thank all the reviewers for their constructive feedback and the time and effort applied in reviewing our manuscript. We are happy that they found our paper well-written, clearly structured (La5K, jX4R, 3tVu, NU6A), and addressing an important problem (La5K, jX4R, riJc, XWnt, NU6H). We provide individual responses to each review, but we also summarize some of the common points here.

### **Comparison with SIMFORMER (La5K, 3tVu, XWnt)**
Based on their suggestions, we have compared RISE's performance against Simformer for the GLU and GLM task. The results (in terms of MMD and nominal log posterior probability (NLPP)) for both the MCAR and the MNAR case are provided in the individual responses below (also in Appendix B.2 of the revised paper). For the MCAR case, both RISE and Simformer perform similarly, with RISE yielding slightly better results. The difference in performance is more stark in the MNAR case, as expected, since Simformer does not explicitly model the missingness mechanism.

### **Additional performance metrics (La5K, jx4r, 3tVu, XWnt)**
We have replaced the RMSE metric with the C2ST scores and the nominal log posterior probability (NLPP) metric for the benchmark tasks, as suggested by several reviewers. The results are shown in the individual responses (also in Table 8, Appendix B.3). The conclusion remains the same, with RISE outperforming the baselines in all but one case (in terms of C2ST). We have also included expected coverage plots for the Hodgkin-Huxley and GLU tasks over various missingness levels in Figure 7 (Appendix B.7). We observe that the posterior produced by RISE becomes more conservative as the proportion of missing values in the data increases, which makes sense.

### **Analysis of computational cost (La5K, jx4r, riJc)**

We have performed an ablation study to evaluate the computational complexity of our method in comparison to standard NPE (see individual responses or Appendix B.5). We observe a roughly 50% increase in runtime per epoch due to the addition of the imputation model. The training time remains the same w.r.t missingness levels over a certain data dimensionality.

### **Clarifications regarding joint training (jx4r, 3tVu, NU6H)**

We have revised Section 3.2 and the proof of Proposition 2 to clarify the joint optimization objective of RISE comprising of the imputation model and the inference network. We copy this explanation in the individual responses to the reviewers for convenience.

We believe that acting on the reviewers’ feedback has significantly improved our experimental evaluation, and we thank them again for their constructive and actionable comments.

---

### Comment · Area_Chair_egPw · 2024-11-26

Dear all,

The deadline for the authors-reviewers phase is approaching (December 2).

@For reviewers, please read, acknowledge and possibly further discuss the authors' responses to your comments. While decisions do not need to be made at this stage, please make sure to reevaluate your score in light of the authors' responses and of the discussion.

- You can increase your score if you feel that the authors have addressed your concerns and the paper is now stronger.
- You can decrease your score if you have new concerns that have not been addressed by the authors.
- You can keep your score if you feel that the authors have not addressed your concerns or that remaining concerns are critical.

Importantly, you are not expected to update your score. Nevertheless, to reach fair and informed decisions, you should make sure that your score reflects the quality of the paper as you see it now. Your review (either positive or negative) should be based on factual arguments rather than opinions. In particular, if the authors have successfully answered most of your initial concerns, your score should reflect this, as it otherwise means that your initial score was not entirely grounded by the arguments you provided in your review. Ponder whether the paper makes valuable scientific contributions from which the ICLR community could benefit, over subjective preferences or unreasonable expectations.

@For authors, please respond to remaining concerns and questions raised by the reviewers. Make sure to provide short and clear answers. If needed, you can also update the PDF of the paper to reflect changes in the text. Please note however that reviewers are not expected to re-review the paper, so your response should ideally be self-contained.

The AC.

---

### Author Response · Authors · 2024-11-26

Dear all,

We have now attached the source code (for GLM and GLU tasks, rest will follow soon) as supplementary material, which was requested by reviewers riJc and NU6H.

Thank you.

---

### Author Response · Authors · 2024-12-03
**Global Response**

Since the discussion period ends soon, we would like to take this opportunity to express our gratitude to the reviewers for their constructive feedback,  and the AC for facilitating the process.  We sketch here a summary of our discussion with the reviewers.

Most reviewers agreed our paper was well-written and addressing an important problem. However, they primarily suggested including additional baselines and performance metrics to further substantiate our experimental results. Based on the additional results we provided,

- Reviewer La5K increased their score from 5 to 8.
- Reviewer jx4r increased their score from 5 to 8.
- Reviewer riJc increased their confidence score from 3 to 4.
- Reviewer 3tVu increased their score from 5 to 6.
- Reviewer XWnt increased their score from 5 to 6.

The discussion period has helped improve the quality, and consolidate the strengths, of this work. Many thanks to everyone!

---

### Meta-Review · Area_Chair_egPw · 2024-12-20

**Metareview:**

The reviewers recommend acceptance (8-8-3-6-6). The paper formalizes the problem of missing data in SBI, demonstrates that naive imputation can introduce bias in the posterior, and proposes a method derived from NPE to address this issue. The approach is well-motivated and the results are convincing. The author-reviewer discussion has been very constructive and has led to a number of clarifications and improvements that have convinced the reviewers. Reviewer NU6H, specifically, has raised concerns and recommended rejection. However, I believe the authors have properly answered these concerns during the author-reviewer discussion period. For these reasons, given the overall positive reviews, I recommend acceptance. I request the authors to implement the many suggestions made during the author-reviewer discussion period in the final version of the paper.

**Additional Comments On Reviewer Discussion:**

The author-reviewer discussion has been very constructive and has led to a number of clarifications and improvements that have convinced the reviewers. Reviewer NU6H, specifically, has raised concerns and recommended rejection. However, I believe the authors have properly answered these concerns during the author-reviewer discussion period.

---

### Decision · Program_Chairs · 2025-01-22

Accept (Poster)